# FRAGFM: HIERARCHICAL FRAMEWORK FOR EFFICIENT MOLECULE GENERATION VIA FRAGMENT-LEVEL DISCRETE FLOW MATCHING

**Joongwon Lee**[1,*], **Seonghwan Kim**[1,2,*], **Seokhyun Moon**[1,*], **Hyunwoo Kim**[3,†],
**Woo Youn Kim**[1,4,5,†]

[1]Department of Chemistry, KAIST,  [2]InnoCORE AI-CRED Institute, KAIST,
[3]College of Pharmacy, Dongguk University,  [4]Department of Data Science, KAIST,  [5]HITS

`{leejwon942,dmdtka00,mshmjp}@kaist.ac.kr, hwkim8906@dongguk.edu,`
`wooyoun@kaist.ac.kr`

## ABSTRACT

We introduce FragFM, a novel hierarchical framework via fragment-level discrete flow matching for efficient molecular graph generation. FragFM generates molecules at the fragment level, leveraging a coarse-to-fine autoencoder to reconstruct details at the atom level. Together with a stochastic fragment bag strategy to effectively handle a large fragment space, our framework enables more efficient, scalable molecular generation. We demonstrate that our fragment-based approach achieves better property control than the atom-based method and additional flexibility through conditioning the fragment bag. We also propose a Natural Product Generation benchmark (NPGen) to evaluate the ability of modern molecular graph generative models to generate natural product-like molecules. Since natural products are biologically prevalidated and differ from typical drug-like molecules, our benchmark provides a more challenging yet meaningful evaluation relevant to drug discovery. We conduct a comparative study of FragFM against various models on diverse molecular generation benchmarks, including NPGen, demonstrating superior performance. The results highlight the potential of fragment-based generative modeling for large-scale, property-aware molecular design, paving the way for more efficient exploration of chemical space.

## 1 INTRODUCTION

Deep generative models are achieving remarkable success in modeling complex, structured data, with graph generation being a prominent application area (Jang et al., 2023; Jo et al., 2022). Among various applications, *de novo* molecular graph generation, which has the potential to accelerate drug and material discovery, is particularly important. Recently, diffusion- and flow-based graph generative models have demonstrated the ability to generate complex molecular graphs (Vignac et al., 2023; Qin et al., 2024; Siraudin et al., 2024; Eijkelboom et al., 2024).

However, these models, which are built on atom-based representations, face significant scalability challenges, particularly when generating large and complex molecules (Qin et al., 2023). The quadratic growth of edges with increasing graph size leads to computational inefficiencies. At the same time, the inherent sparsity of chemical bonds makes accurate edge prediction more complex, often leading to unrealistic molecular structures or invalid connectivity constraints (Qin et al., 2023; Chen et al., 2023). Moreover, graph neural networks struggle to capture topological features like rings, leading to deviations from chemically valid structures. Although various methods incorporate auxiliary features (e.g., spectral, ring, and valency information) to mitigate these issues, they do not fully resolve the sparsity and scalability bottlenecks (Vignac et al., 2023).

---

[*]These authors contributed equally.
[†]Corresponding authors.
Code and data are available at `https://github.com/lee-jwon/FragFM`.

Fragment-based strategies, rooted in long-standing success in traditional drug discovery, offer an alternative (Hajduk & Greer, 2007; Joseph-McCarthy et al., 2014; Kirsch et al., 2019). By assembling molecules from chemically meaningful substructures, these approaches enable a more efficient exploration of chemical space, preserve global structural coherence, and provide finer control over molecular properties than atom-based methods (Jin et al., 2018; Seo et al., 2023; Hetzel et al., 2023; Jin et al., 2020a). Diffusion models also adopted the fragment-based approach, showing their potential in improving scalability and property control (Levy & Rector-Brooks, 2023; Chen et al., 2024). However, existing methods rely on a small, fixed-fragment vocabulary or employ automated fragmentation procedures, which severely limit coverage of chemical space and hinder integration of domain knowledge.

Here, we introduce FragFM, a novel hierarchical framework for molecular graph generation to address these challenges. FragFM first generates a fragment-level graph using discrete flow matching and then reconstructs it into an atom-level graph without information loss. To this end, we develop a novel **stochastic fragment bag strategy** that circumvents reliance on fixed fragment libraries, along with a **coarse-to-fine autoencoder** that ensures direct atom-level reconstruction from the generated fragment-level graph. Consequently, FragFM can efficiently explore the molecular space, avoiding the generation of chemically implausible molecules with an extensive fragment space at moderate computational cost.

Our main contributions are summarized as follows:

- We propose FragFM, a novel hierarchical framework that combines fragment-level discrete flow matching with a coarse-to-fine autoencoder, designed to operate effectively on large fragment libraries.

- We introduce NPGen, a new benchmark for natural product generation, designed to evaluate larger and more complex molecules.

- Extensive experiments show that FragFM not only outperforms prior molecular generative models, but also achieves substantially stronger performance on large, natural product–like molecules. Moreover, it remains robust under fewer denoising steps.

- FragFM enables more effective and flexible property-guided molecular generation through both fragment bag control and conventional guidance strategies.

## 2 RELATED WORK

### 2.1 MOLECULAR GRAPH GENERATIVE MODELS

Modern molecular graph generative models can be classified into autoregressive and one-shot generation models. Autoregressive models generate graphs sequentially based on their node, generally an atom or fragment, and edge representations (Lim et al., 2020; Mercado et al., 2021; Shi et al., 2020; Jin et al., 2018). Despite their performance, these models have an intrinsic issue in learning the permutation of nodes in the graph, which must be invariant for a given graph, often making them highly inefficient. Among one-shot models, there exists a model that directly generates molecular graphs (Kwon et al., 2020). Also, denoising models have recently become fundamental for generating molecular graphs by iteratively refining noisy graphs into structured molecular representations. Diffusion methods, which have been successful in various domains, have been extended to graph structure data (Jo et al., 2022; Niu et al., 2020), demonstrating the advantages of applying diffusion in graph generation. This approach was further extended by incorporating discrete stochastic processes (Austin et al., 2021), addressing the inherently discrete nature of molecular graphs (Vignac et al., 2023). The discrete diffusion modeling is reformulated using the continuous-time Markov chain (CTMC) (Xu et al., 2024; Siraudin et al., 2024; Kim et al., 2024), allowing for more flexible and adaptive generative processes. More recently, flow-based models have been explored for generating molecular graphs. Continuous flow matching (Lipman et al., 2022) has been applied to structured data (Eijkelboom et al., 2024), while discrete flow models (Campbell et al., 2024; Gat et al., 2024) have been extended to categorical data generation, with recent methods showing that they can also model molecular distributions as diffusion models (Qin et al., 2024; Hou et al., 2024).

## 2.2 Fragment-Based Molecule Generation

Fragment-based molecular generative models construct new molecules by assembling existing molecular substructures, known as fragments. This strategy enhances chemical validity and synthesizability, facilitating the efficient exploration of novel molecular structures compared to the atom-based approaches. Several works have employed fragment-based approaches within variational autoencoders (VAEs) by learning to assemble in a chemically meaningful way (Jin et al., 2020b; Kong et al., 2022; Maziarz et al., 2021; Podda et al., 2020). Also, Jin et al. (2018) adopts a stepwise generation approach, constructing a coarse fragment-level graph before refining it into an atom-level molecule through substructure completion. The other strategies construct molecules sequentially assembling fragments, enabling better control over molecular properties during generation (Seo et al., 2023; Jin et al., 2020b). In contrast to these assembly-based methods, Noutahi et al. (2024) proposed SAFE-GPT, a GPT-2–based transformer model that generates SAFE (Sequential Attachment-based Fragment Embedding) strings, a novel representation that expresses molecules as sequences of fragment-level tokens rather than through explicit fragment attachment.

Fragment-based approaches have also been explored in diffusion-based molecular graph generation. Levy & Rector-Brooks (2023) proposed a method that utilizes a fixed set of frequently occurring fragments to generate drug-like molecules, ensuring chemical validity but limiting exploration beyond predefined structures. Since enumerating all possible fragment types is infeasible, the method operates solely within a fixed fragment vocabulary. In contrast, Chen et al. (2024) proposed an alternative, dataset-dependent fragmentation strategy based on byte-pair encoding, which provides a more flexible molecular representation. However, this approach does not yet integrate chemically meaningful fragmentation methods (Degen et al., 2008; Liu et al., 2017), which are inspired by chemical synthesis and functionality, limiting its ability to leverage domain-specific chemical priors.

## 2.3 Hierarchical molecular generative models

Beyond these fragment-based approaches, several hierarchical generative models explicitly leverage multi-scale graph structure for molecule generation. MolGrow (Kuznetsov & Polykovskiy, 2021) and MolHF (Zhu et al., 2023) introduce hierarchical normalizing flows that generate molecular graphs in a coarse-to-fine fashion by recursively refining coarsened graph representations, enabling the generation of larger and more complex molecules but operating on non-chemical subgraphs rather than chemically defined fragments. In 3D molecule generation, Qiang et al. (2023) proposes a hierarchical diffusion model that sequentially samples coarse-grained fragments, fine-grained fragments, and atom-level conformations, improving local structural validity without relying on explicit synthesis-driven fragmentation schemes. Compared to these hierarchy-on-graphs methods, which induce fragment-like units implicitly from data and graph coarsening, fragment-based models with chemically meaningful fragment vocabularies provide more direct control over substructural composition and synthesizability, and our work follows this latter line by treating domain-informed fragments as the fundamental generation units while still benefiting from modern generative architectures.

## 3 FragFM Framework

We propose FragFM, a novel hierarchical framework that utilizes discrete flow matching at the fragment-level graph. As illustrated in fig. 1, our framework introduces two key strategies: (i) a coarse-to-fine autoencoder, and (ii) a stochastic fragment bag strategy for coarse-graph generation. The autoencoder compresses atom-level graphs $\mathbf{G}$ into fragment-level graphs $\mathcal{G}$, while preserving atomistic connectivity in a latent variable $z$. This design enables the use of discrete flow matching (DFM) at the fragment level. The stochastic fragment bag strategy ensures the model handles comprehensive fragment libraries at a manageable computational cost. To realize this strategy, we adopt a graph neural network module for fragment embedding, which enables generalization to unseen fragments. In this section, we first describe the conversion between fragment- and atom-level graphs in section 3.1, and then present the flow-matching procedure for the coarse graph, including both training (section 3.2) and generation (section 3.3) at the fragment level.

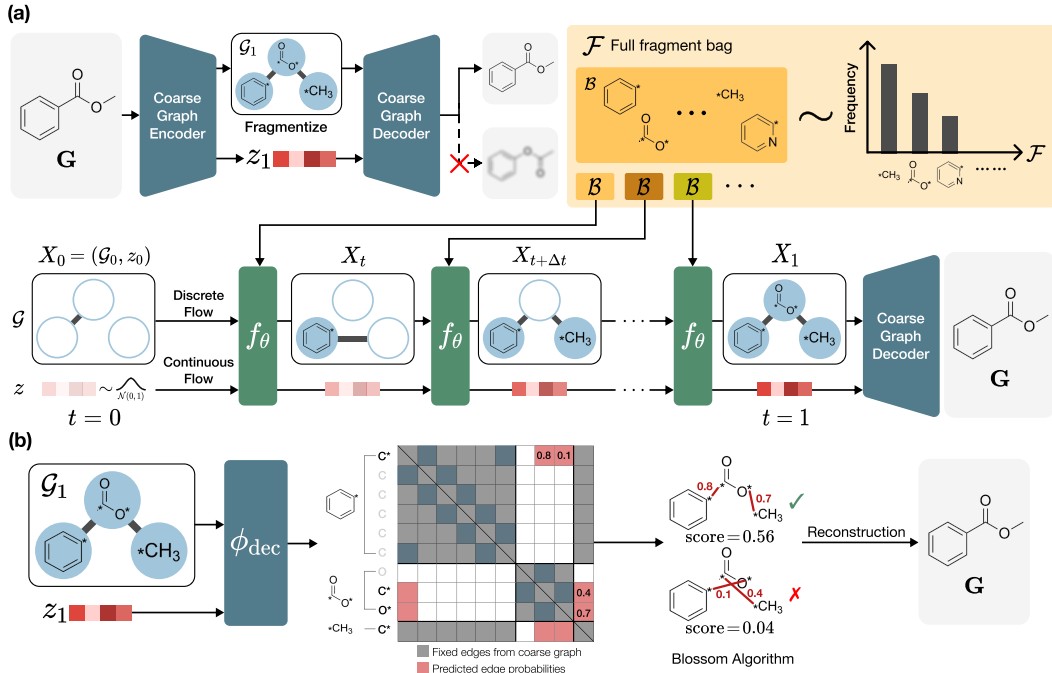

Figure 1: **Overview of FragFM**. **(a)** FragFM utilizes a hierarchical framework of coarse-to-fine autoencoder (section 3.1) and fragment-level graph flow matching (section 3.2). An input atom-level graph (**G**) is initially decomposed via the fragmentation rule. This is then processed by a coarse-to-fine encoder, which compresses it into a joint representation $X = (\mathcal{G}, z)$ comprising a fragment-level graph $\mathcal{G}$ and a latent vector $z$ designed to capture fine-grained atomistic connectivity information not explicitly present in $\mathcal{G}$. During generation (section 3.3), neural network $f_\theta$ selects the most probable fragment from a fragment bag $\mathcal{B}$, which is a stochastically sampled subset of the full fragment bag $\mathcal{F}$. FragFM then employs two flow-matching processes: (i) a discrete flow generates the target fragment-level graph $\mathcal{G}_1$ from an initial $\mathcal{G}_0$ (mask and uniform prior for node and edge, respectively), operating with fragments from $\mathcal{B}$; (ii) a continuous flow generates the target latent vector $z_1$ from a Gaussian prior $\mathcal{N}(0, 1)$ (from an initial $z_0$). **(b)** Finally, given $(\mathcal{G}_1, z_1)$, the coarse-to-fine decoder reconstructs the atom-level molecular graph by first predicting the probabilities of all possible atom-to-atom edges, and then applying the Blossom algorithm to select the edge set that maximizes the likelihood of the true graph. Further details and hyperparameters are described in section E and fig. 21.

## 3.1 MOLECULAR GRAPH COMPRESSION BY COARSE-TO-FINE AUTOENCODER

While a fragment-level graph $\mathcal{G}$ offers a higher-level abstraction of molecular structures, it also introduces ambiguity in reconstructing atomic connections. Specifically, a single fragment-level connectivity $\mathcal{E}$ can map to multiple distinct, valid atom-level configurations. To achieve accurate end-to-end molecular generation, it is therefore crucial to preserve atom-level connectivity $\mathbf{E}$ when forming the fragment-level representation. Drawing on a hierarchical generative framework (Razavi et al., 2019; Rombach et al., 2022; Qiang et al., 2023), we employ a coarse-to-fine autoencoder.

The encoder compresses an atom-level graph $\mathbf{G}$ into its fragment-level counterpart $\mathcal{G}$ and, for each input molecule, outputs a single continuous latent vector $z$ that encodes the committed connectivity details. Specifically, $\mathbf{G}$ is first converted into $\mathcal{G}$ using a predefined fragmentation rule (*e.g.* BRICS (Degen et al., 2008)), after which a neural network encodes $(\mathbf{G}, \mathcal{G})$ into $z$. Given $\mathcal{G}$ and $z$, the decoder reconstructs atom-level edges between adjacent fragments in $\mathcal{G}$. This process combines a neural network that outputs continuous edge scores with the Blossom algorithm (Edmonds, 1965), which discretizes these scores into valid atom-level connectivity. The coarse-to-fine autoencoder is simply

constructed as:

$$\textbf{Encoder:} \quad \mathbf{G} \xrightarrow{\text{Rule}} \mathcal{G}, \quad (\mathbf{G}, \mathcal{G}) \xrightarrow{\phi_{\text{enc}}} z,$$

$$\textbf{Decoder:} \quad \mathcal{G}, z \xrightarrow{\phi_{\text{dec}}} \text{score} \xrightarrow{\text{Blossom}} \mathbf{E}.$$

We verified that the autoencoder can faithfully reconstruct atom-level graphs, achieving over 99% bond-level accuracy on standard benchmarks (see section C.3 for details). Additional implementation details are provided in section A.1.

## 3.2 FLOW MATCHING FOR COARSE GRAPH

We aim to model the joint distribution over the fragment-level graph and its latent representation, $X := (\mathcal{G}, z)$, through the flow-matching after a continuous-time generative paradigm. Flow matching begins at a known prior at $t=0$ and follows a learned vector field that continuously transforms this prior into the target data distribution at $t=1$.

In our coarse graph $\mathcal{G}$, both nodes $\{x\}$ and edges $\{\varepsilon\}$ are discrete categorical variables, for which we adopt DFM realized by a continuous time Markov chain (CTMC) (Campbell et al., 2024). In this section, we focus on the generation of fragment types. For latent vector and edges, we follow the standard flow-matching for continuous (Lipman et al., 2022) and discrete (Campbell et al., 2024) features, further described in section A.2.

**DFM for fragment types and Info-NCE Loss.** Because realistic chemical spaces involve an extremely large vocabulary of fragment types $|\mathcal{F}|$, directly modeling a CTMC over the entire space is computationally prohibitive. To address this, we adopt a masked version of DFM, in which a node remains fixed once it is de-masked, and further introduce a stochastic fragment bag strategy to efficiently handle the large fragment vocabulary. Given a noisy state $X_t = (\mathcal{G}_t, z_t)$, we draw a subset $\mathcal{B} \subset \mathcal{F}$ of size $N$ from the full fragment vocabulary and then sample a node $x_1$ within this restricted subset. As a result, the model approximates the in-bag conditional posterior $p(x_1 \mid X_t, x_1 \in \mathcal{B})$ rather than the unconditional one $p_{1|t}(x_1 \mid X_t)$.

To train the model, we follow the Info-NCE formulation (Oord et al., 2018), constructing fragment bags and parameterizing the density ratio $p_{1|t}(x_1|X_t)/p_1(x_1)$ with a neural network $f_\theta$. For each training step we build a bag $\mathcal{B}$ that contains one positive fragment $x_1^+ \sim p_{1|t}(x_1|X_t)$ and $N-1$ negative fragments $x_1^-$ sampled i.i.d. from the marginal fragments library distribution $p_1(x)$. Applying the Info-NCE formulation, we can write the in-bag posterior as

$$p_{1|t}(x \mid X_t, \mathcal{B}) = \frac{\mathbf{1}_{\mathcal{B}}(x) p_{1|t}(x \mid X_t)/p_1(x)}{\sum_{y \in \mathcal{B}} p_{1|t}(y \mid X_t)/p_1(y)}, \tag{1}$$

where $\mathbf{1}_{\mathcal{B}}$ is an indicator function. We let the $f_\theta(X_t, x)$ approximate the unknown density ratio $p_{1|t}(x|X_t)/p_1(x)$, by optimizing $\theta$ with the standard Info-NCE loss:

$$\mathcal{L}(\theta) = -\mathbb{E}_{\mathcal{B}}\left[\log \frac{f_\theta(X_t, x^+)}{\sum_{y \in \mathcal{B}} f_\theta(X_t, y)}\right], \tag{2}$$

which encourages the network to assign higher scores to the positive $x^+$ within $\mathcal{B}$ while pushing down the negatives. Because the loss involves only $x \in \mathcal{B}$, its cost scales with $N$ rather than $|\mathcal{F}|$.

## 3.3 GENERATION PROCESS OF FRAGFM

During sampling, the model evolves nodes, edges, and the latent vector step by step from the prior distribution. This requires a discretized forward kernel expressed as: $p_{t+\Delta t|t}(X_{t+\Delta t} \mid X_t) = \prod_i p_{t+\Delta t|t}(x_{t+\Delta t}^{(i)} \mid X_t) \prod_{ij} p_{t+\Delta t|t}(\varepsilon_{t+\Delta t}^{(ij)} \mid X_t) \, p_{t+\Delta t|t}(z_{t+\Delta t} \mid X_t)$. Similar to Campbell et al. (2024), we modeled each transition of nodes and edges as independent. In this section, we focus on the DFM process for nodes, while details of edges and latent vectors are provided in section A.2.

**In-bag transition kernel** One-step transition kernel $p_{t+\Delta t|t}$ can be obtained by direct Euler integration of the rate matrix, which is computed by an expectation of $x_1$-conditioned rate matrix over the full fragment set $\mathcal{F}$. In standard DFM, this expectation is approximated by sampling $x_1$ from the trained model $p_{1|t}(x_1 \mid X_t)$. We instead define an in-bag transition kernel by restricting $x_1$ to a randomly selected subset $\mathcal{B} \subset \mathcal{F}$.

Following the conventional Info-NCE approaches, we construct $\mathcal{B}$ by drawing $N$ i.i.d. fragments from the marginal $p_1$, assuming that $\mathcal{B}$ is independent of the current state. Consequently the $\mathcal{B}$ conditioned $x_1$-posterior is

$$p_{1|t,\mathcal{B}}^\theta(x_1 \mid X_t, \mathcal{B}) = \frac{\mathbf{1}_\mathcal{B}(x_1) f_\theta(X_t, x_1)}{\sum_{y \in \mathcal{B}} f_\theta(X_t, y)}.$$

The bag conditioned forward kernel for a node is then simply induced by:

$$p_{t+\Delta t|t}^\theta(x_{t+\Delta t}|X_t, \mathcal{B}) := \mathbb{E}_{x_1 \sim p_{1|t,\mathcal{B}}^\theta(\cdot|X_t, \mathcal{B})} \left[ p_{t+\Delta t|t}(x_{t+\Delta t}|X_t, x_1) \right]. \tag{3}$$

Strictly speaking, eq. (3) differs from the kernel without the bag. It nevertheless serves as a practical surrogate that converges to the exact one as the bag size $N$ approaches the fragment-pool size $|\mathcal{F}|$ (Oord et al., 2018). When the Euler step size $\Delta t$ is small, and the bag size $N$ is moderately large, the discrepancy is negligible while the computational cost remains manageable.

**Conditional generation** While generating valid molecules is essential, steering them toward desired properties is crucial for practical use. Following Dhariwal & Nichol (2021); Vignac et al. (2023), we adopt classifier guidance, steering with an external property predictor.

Because our framework employs a bag-conditioned transition kernel, conditioning introduces two key requirements. First, the selection of the fragment bag must be steered by the target property $c$, so that the candidate fragments align with the desired outcome. Second, the transition kernel must incorporate both the bag and the property, effectively forming a multi-conditioned kernel. In practice, the first requirement is addressed by re-weighting fragment sampling probabilities when constructing $\mathcal{B}$, guided by a property predictor $p_{\psi_{\text{prop}}}(c \mid x)$ and a tunable fragment bag re-weighting parameter $\lambda_\mathcal{B}$. The second is handled by a guidance strength parameter $\lambda_X$ as in Vignac et al. (2023). Further details of the conditional transition kernel and the construction of property-conditioned bags $p(\mathcal{B} \mid c)$ are provided in section A.3.

## 4 NPGen: Natural Product Generation Benchmark

We introduce NPGen, a new benchmark for molecular generative models focused on natural products (NPs), which are biologically synthesized compounds produced by organisms and characterized by distinctive structural features (Feher & Schmidt, 2003; Stratton et al., 2015). NPs represent a biologically meaningful subset of chemical space, and serve as a motivation for many approved drugs (Boufridi & Quinn, 2018; Rosén et al., 2009; Atanasov et al., 2021). However, existing benchmarks such as MOSES and GuacaMol predominantly comprise small, structurally simple molecules, and their evaluation metrics—Fréchet ChemNet Distance, scaffold overlap, and KL divergence over simple properties—are nearing saturation, limiting their ability to capture NP-specific characteristics (Bechler-Speicher et al., 2025). To address this limitation, we construct NPGen by selecting molecules from the COCONUT database (Sorokina et al., 2021; Chandrasekhar et al., 2025), which occupies a distinct region of chemical space compared to existing benchmarks (fig. 2a). As a result, NPGen contains 658,566 natural product molecules with an average heavy-atom count of 35.0 (larger than those in MOSES (21.7) and GuacaMol (27.9)) and with richer structural diversity characteristic of natural products (fig. 2b). NPGen's evaluation includes not only standard metrics (Validity, Uniqueness, Novelty) but also NP-oriented measures such as KL divergence of NP-likeness score distributions (Ertl et al., 2008) and NP's biosynthetic pathways and structure classes (Kim et al., 2021), which capture molecular functionality and biological context related to its structure. We provide additional details on NPs (section B.1), dataset construction (section B.2), baseline implementations (section B.3), evaluation metrics (section B.4), and dataset statistics (section B.5).

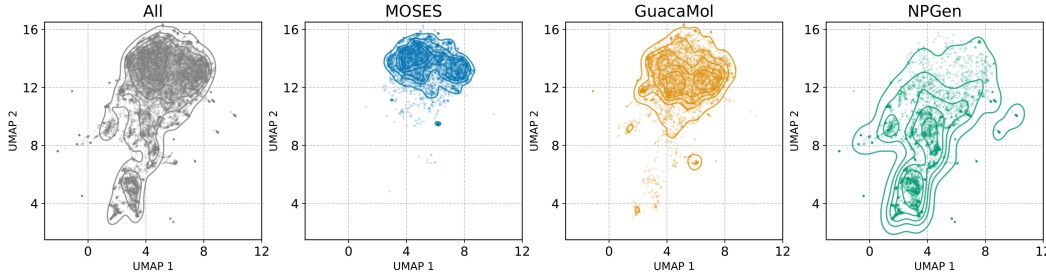

(a) UMAP plot of MOSES, GuacaMol, and NPGen datasets, each with 5,000 randomly sampled molecules.

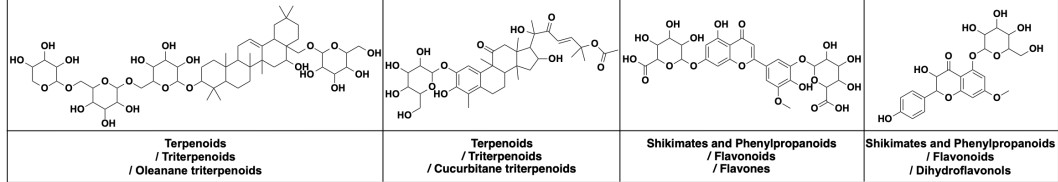

(b) Representative molecules from NPGen with NPClassifier pathway/superclass/class annotations.

Figure 2: **NPGen dataset overview. (a)** UMAP visualization comparing MOSES, GuacaMol, and NPGen datasets. **(b)** Representative molecules from NPGen with annotations from NPClassifier (pathway, superclass, and class).

## 5 RESULTS

Here, we present the main results of FragFM on molecular generation benchmarks (sections C.1, 5.1 and 5.2), conditional generation (sections C.8 and 5.3), sampling efficiency (sections C.9 and 5.4), and an ablation on fragmentation rules (section 5.5).

Extended analyses and ablation studies are provided in the Appendix, including an evaluation of retrosynthetic analysis (section C.2), an analysis of the coarse-to-fine autoencoder (section C.3), generalization to rare and novel fragment types (sections C.4 and C.5), further study of fragment bag size (section C.6), experiments for novel molecule generation (section C.7). We also provide visualizations of generated molecules for both standard benchmarks and baseline models in NPGen (sections F.3 and F.4) as well as visualizations of the latent space of fragment embeddings (section F.2).

### 5.1 STANDARD MOLECULAR GENERATION BENCHMARKS

We evaluate FragFM on the MOSES (table 1), GuacaMol (table 4), and ZINC250k (table 5) datasets, which focus on small molecule generation. We compare against a range of baseline models spanning different generation strategies and representation levels, with additional details provided in section D.

For MOSES, we follow Vignac et al. (2023) and report results on the scaffold-split test set. In this benchmark, atomistic models typically underperform the fragment-based method on validity and FCD. Notably, FragFM achieves nearly 100% validity without any explicit validity constraints—comparable to JT-VAE, which explicitly enforces molecular validity—and attains an FCD of 0.58, substantially outperforming all baselines. In addition, FragFM achieves state-of-the-art (all FCDs, MOSES Filters, SNN, ZINC NSPDK) or near-best (GuacaMol KL Div.) results on property- and structure-based scores across benchmarks.

Despite strong results on most metrics, performance on the MOSES Scaf and novelty metrics is relatively weaker. For novelty, we note that higher values do not always guarantee better molecular quality, as there exists a trade-off between fidelity and novelty (Mahmood et al., 2021; Geng et al., 2023). In this regard, we demonstrate that FragFM can trade off fidelity for novelty through a simple modification, which we describe in section C.7. For Scaf, this stems from the scaffold-split protocol of the MOSES benchmark: unseen scaffolds in the test set cannot be generated if they are absent from the fragment vocabulary, a limitation of fixed fragment vocabularies also observed in JT-VAE. Unlike prior fragment-based approaches, however, FragFM allows flexible replacement of the vocabulary due

to the GNN-based fragment embedding. To verify this, we further demonstrate that it can generalize when equipped with a test-set fragment vocabulary, as detailed in section C.4.

Table 1: **Molecule generation on MOSES**. We use 25,000 generated molecules for evaluation. The upper part comprises autoregressive methods, while the second part comprises one-shot methods, including diffusion-based and flow-based methods. Results for FragFM are averaged over three independent runs. The best performance is highlighted in **bold**, and the second-best is underlined.

| Model | Rep. Level | Valid ↑ | Unique ↑ | Novel ↑ | Filters ↑ | FCD ↓ | SNN ↑ | Scaf ↑ |
|---|---|---|---|---|---|---|---|---|
| Training set | - | 100.0 | 100.0 | - | 100.0 | 0.48 | 0.59 | 0.0 |
| GraphINVENT (Mercado et al., 2021) | Atom (Graph) | 96.4 | 99.8 | - | 95.0 | 1.22 | 0.54 | 12.7 |
| JT-VAE (Jin et al., 2018) | Fragment (Graph) | **100.0** | **100.0** | 99.9 | 97.8 | 1.00 | 0.53 | 10.0 |
| SAFE-GPT (Noutahi et al., 2024) | Fragment (Sequence) | 98.1 | **100.0** | 90.9 | 98.2 | 0.71 | 0.54 | 9.8 |
| MolHF (Zhu et al., 2023) | Atom (Graph) | 71.0 | **100.0** | **100.0** | 17.8 | 35.4 | 0.23 | 1.5 |
| MolGrow (Kuznetsov & Polykovskiy, 2021) | Fragment (Graph) | **100.0** | **100.0** | 99.8 | 82.7 | 6.39 | 0.42 | 7.7 |
| DiGress (Vignac et al., 2023) | Atom (Graph) | 85.7 | **100.0** | 95.0 | 97.1 | 1.19 | 0.52 | 14.8 |
| DisCo (Xu et al., 2024) | Atom (Graph) | 88.3 | **100.0** | 97.7 | 95.6 | 1.44 | 0.50 | 15.1 |
| Cometh (Siraudin et al., 2024) | Atom (Graph) | 90.5 | **100.0** | 96.4 | 97.2 | 1.44 | 0.51 | 15.9 |
| Cometh-PC (Siraudin et al., 2024) | Atom (Graph) | 90.5 | 99.9 | 92.6 | **99.1** | 1.27 | 0.54 | **16.0** |
| DeFoG (Qin et al., 2024) | Atom (Graph) | 92.8 | 99.9 | 92.1 | 98.9 | 1.95 | 0.55 | 14.4 |
| FragFM (ours) | Fragment (Graph) | 99.8 | **100.0** | 87.1 | **99.1** | **0.58** | **0.56** | 10.9 |

## 5.2 BENCHMARKING GENERATIVE MODELS ON NPGEN

We now present the comprehensive evaluation results for our proposed NPGen benchmark (section 4), establishing baselines across diverse modalities—including atom-fragment-level and graph-sequence-based models—alongside our FragFM. As shown in table 2, while the sequence-based SAFE-GPT achieves high distributional alignment compared to graph-based baselines, it exhibits limited novelty (73.5%). Among graph-based models, FragFM achieves superior performance on functionality-driven metrics, including NP-likeness and NP-Classifier scores, while maintaining high validity and novelty. Furthermore, it surpasses atom-based discrete flow and diffusion models (DeFoG, DiGress) by a substantial margin while offering significantly faster sampling (e.g., $5\times$ faster than DiGress; fig. 7). These results highlight the advantage of our fragment-based framework in efficiently capturing the complex structures of natural products. We provide visualizations of generated molecules for graph-based baselines in fig. 3 and section F.4.

Table 2: **Molecule generation results on NPGen**. We use 30,000 generated molecules for evaluation. The upper part comprises autoregressive methods, while the second part comprises one-shot methods, including diffusion-based and flow-based methods. The results are averaged over three runs. The best performance is highlighted in **bold**, and the second-best is underlined.

| Model | Rep. Level | Val. ↑ | Unique. ↑ | Novel ↑ | NP Score KL Div. ↓ | NP Class KL Div. ↓ Pathway | Superclass | Class | FCD ↓ |
|---|---|---|---|---|---|---|---|---|---|
| Training set | - | 100.0 | 100.0 | - | 0.0006 | 0.0002 | 0.0028 | 0.0094 | 0.01 |
| GraphAF (Shi et al., 2020) | Atom (Graph) | 79.1 | 63.6 | 95.6 | 0.8546 | 0.9713 | 3.3907 | 6.6905 | 25.11 |
| JT-VAE (Jin et al., 2018) | Fragment (Graph) | **100.0** | 97.2 | 99.5 | 0.5437 | 0.1055 | 1.2895 | 2.5645 | 4.07 |
| HierVAE (Jin et al., 2020a) | Fragment (Graph) | **100.0** | 81.5 | 97.7 | 0.3021 | 0.4230 | 0.5771 | 1.4073 | 8.95 |
| SAFE-GPT (Noutahi et al., 2024) | Fragment (Sequence) | 96.5 | 98.6 | 73.5 | **0.0024** | **0.0054** | **0.0414** | **0.1722** | **0.15** |
| MolHF (Zhu et al., 2023) | Atom (Graph) | 71.0 | 59.6 | 97.6 | 0.8831 | 1.8072 | 9.1608 | 10.3760 | 31.26 |
| DiGress (Vignac et al., 2023) | Atom (Graph) | 85.4 | **99.7** | **99.9** | 0.1957 | 0.0229 | 0.3370 | 1.0309 | 2.05 |
| DeFoG (Qin et al., 2024) | Atom (Graph) | 85.9 | 98.4 | 99.2 | 0.1550 | 0.1252 | 0.4134 | 1.3597 | 4.46 |
| FragFM (ours) | Fragment (Graph) | 98.0 | 99.0 | 95.4 | 0.0374 | 0.0196 | 0.1482 | 0.3570 | 1.34 |

## 5.3 ENHANCED CONTROLLABILITY VIA JOINT FRAGMENT AND CLASSIFIER GUIDANCE

A key requirement for molecular generative models is controllability, *i.e.*, steering generated molecules toward desired properties while retaining validity and distributional fidelity. We evaluate the conditional generation ability of FragFM against the atom-based baseline DiGress, where both models employ a classifier guidance scheme. For all conditional generation experiments, both FragFM and DiGress use property predictors trained on the full set of labeled molecules for each corresponding dataset.

To this end, we first vary the guidance strength ($\lambda_X$) and plot the resulting trade-off curves (MAE–FCD and MAE–validity). When conditioning on simple molecular properties such as QED

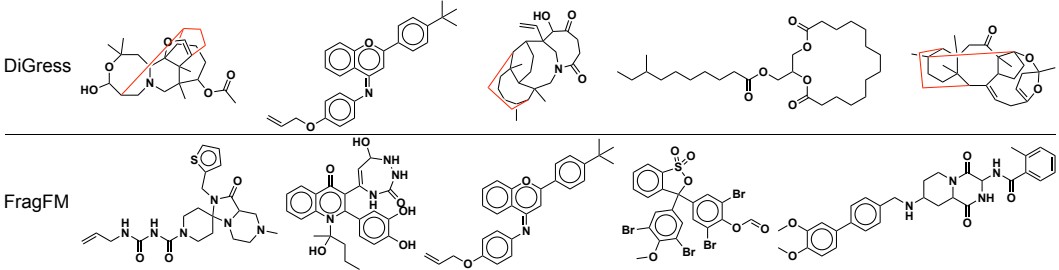

Figure 3: **Randomly selected molecules from DiGress (top) and FragFM (bottom) trained on NPGen**. We randomly sample a moderate-sized molecule containing 31 to 40 heavy atoms. Chemically implausible moieties are highlighted in red. More examples are provided in section F.4.

(figs. 4 and 14), logP (fig. 15), and ring count (fig. 16), FragFM attains lower conditional MAE with lower FCD and higher validity, while the atom-based baseline often suffers sharp validity drops under strong guidance.

We further confirm that fragment-based generation provides additional flexibility through fragment-bag conditioning by $\lambda_{\mathcal{B}}$. In the JAK2 docking score task (fig. 5), FragFM consistently outperforms the atom-based baseline even without bag guidance ($\lambda_{\mathcal{B}} = 0$). Increasing $\lambda_{\mathcal{B}}$ further shifts the generated distribution toward the target docking score while preserving nearly 100% validity, whereas DiGress not only suffers from substantial validity loss but also shows little shift in the docking score distribution from the original data. Importantly, this conditioning is applied in a particularly challenging region of ZINC250k (top 0.08%), where such low docking scores are extremely rare, highlighting the robustness of fragment-based design in realistic scenarios.

Finally, we analyze the joint effect of the two guidance terms, $\lambda_X$ and $\lambda_{\mathcal{B}}$. As shown in the MAE–FCD curves (fig. 5), each curve corresponds to varying $\lambda_X$, while adjusting $\lambda_{\mathcal{B}}$ consistently shifts these curves toward more favorable trade-off regions. This demonstrates that fragment-bag reweighting complements the standard classifier guidance, offering an additional degree of controllability that is unique to fragment-based generation and unattainable in purely atomistic approaches. Moreover, the results of using $\lambda_{\mathcal{B}}$ only (red points in fig. 5) indicate that employing a conditional fragment bag alone can already achieve effective conditioning. Together, these findings align with the long-standing success of fragment-based paradigms in medicinal chemistry (Sadybekov et al., 2022; Hajduk & Greer, 2007) and recent computational strategies (Lee et al., 2024), while emphasizing the importance of preparing property-oriented fragment candidates.

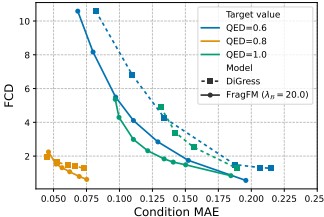
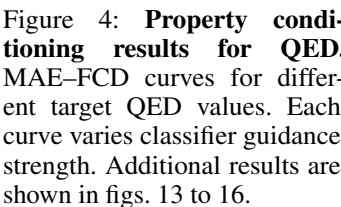
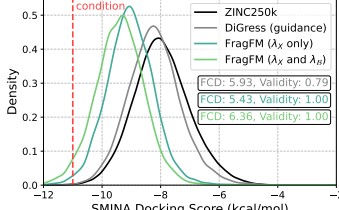
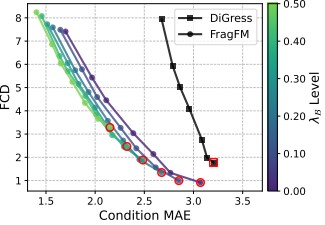

Figure 4: **Property conditioning results for QED.** MAE–FCD curves for different target QED values. Each curve varies classifier guidance strength. Additional results are shown in figs. 13 to 16.

Figure 5: **Effect of $\lambda_{\mathcal{B}}$ in conditioning**. (left) $\lambda_{\mathcal{B}} = 0.4$, $\lambda_X = 2.0$; the DiGress guidance level is set as 2,000 for comparable FCD values. (right) MAE–FCD curves on ZINC250k with JAK2 docking score conditioning at $-11.0$ kcal/mol. Red markers indicate $\lambda_X = 0.0$, *i.e.*, fragment-bag-only guidance. Each curve shows results as the classifier guidance strength is varied. Additional results are shown in figs. 17 to 20.

## 5.4 SAMPLING EFFICIENCY AND ROBUSTNESS OF FRAGFM

Iterative denoising in stochastic generative models involves a trade-off between the number of sampling steps and output quality. As shown in Figure 6, most diffusion- and flow-based models suffer declines in validity and FCD as the number of steps decreases, whereas FragFM remains robust, maintaining over 95% validity and FCD below $1.0$ even with fewer steps. This efficiency arises from operating on fragment-level structures rather than individual atoms. Moreover, FragFM achieves substantially faster sampling time at the fragment level, as illustrated in Figure 7. Additional results and full tables are provided in section C.9.

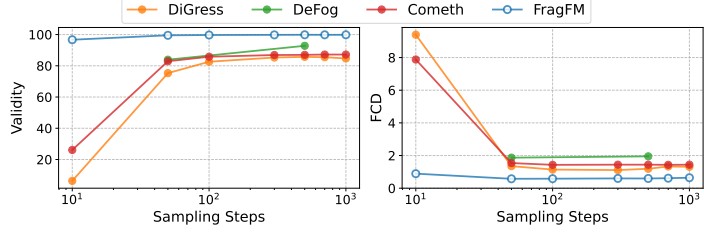
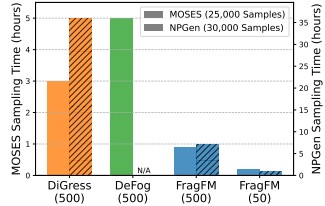

Figure 6: **Analysis of sampling steps across multiple denoising models.** FragFM maintains higher sampling quality than baseline atom-based denoising models as the number of sampling steps is reduced, exhibiting significantly less performance degradation. Additional results are provided in section C.9.

Figure 7: **Sampling time for MOSES and NPGen across different models**. The number in parentheses indicates the sampling steps.

## 5.5 ABLATION STUDY ON FRAGMENTATION RULES

We further ablate the fragmentation rule by replacing the default BRICS with RECAP (Lewell et al., 1998) and rBRICS (Zhang et al., 2024). Although BRICS is used by default due to its wide adoption and chemical interpretability, FragFM is not tied to a specific fragmentation scheme. As shown in Table 3, FragFM maintains consistently strong performance under alternative fragmentation rules, while a finer fragmentation rule (rBRICS) slightly increases novelty. Overall, these results show that FragFM is robust to different fragmentation-induced vocabularies when trained under each scheme.

Table 3: **Molecule generation on MOSES and ZINC250k with different fragmentation rules.** For rBRICS and RECAP, we used the same hyperparameter settings as FragFM (BRICS).

| Model | MOSES | | | | | | | ZINC250k | | |
| | Valid ↑ | Unique ↑ | Novel ↑ | Filters ↑ | FCD ↓ | SNN ↑ | Scaf ↑ | Valid ↑ | NSPDK ↓ | FCD ↓ |
|---|---|---|---|---|---|---|---|---|---|---|
| Training set | 100.0 | 100.0 | - | 100.0 | 0.48 | 0.59 | 0.0 | 100.0 | 0.0001 | 0.062 |
| FragFM (BRICS) | 99.8 | 100.0 | 87.1 | 99.1 | 0.58 | 0.56 | 10.9 | 99.81 | 0.0002 | 0.630 |
| FragFM (RECAP) | 99.8 | 99.9 | 83.6 | 99.3 | 0.56 | 0.57 | 13.3 | 99.66 | 0.0003 | 0.580 |
| FragFM (rBRICS) | 99.8 | 100.0 | 88.5 | 98.7 | 0.58 | 0.56 | 13.1 | 99.79 | 0.0003 | 0.563 |

## 6 CONCLUSION

In this paper, we have introduced FragFM, a novel hierarchical framework with fragment-level discrete flow matching followed by lossless reconstruction of the atom-level graph, for efficient molecular graph generation. To this end, we proposed a stochastic fragment bag strategy with a coarse-to-fine autoencoder to circumvent dependency on a limited fragment library cost-effectively. Standing on long-standing fragment-based strategies in chemistry, FragFM showed superior performance on the standard molecular generative benchmarks compared to the previous graph generative models. Additionally, applying classifier guidance at the fragment level and conditioning the fragment bag on the target property enables more precise control over diverse molecular properties. These significant improvements pave the way for a new frontier for fragment-based denoising approaches in molecular graph generation. Finally, to contribute to the growth of the molecular graph-generating domain, we developed a new benchmark for evaluating models of natural products, which is also crucial in drug discovery.

ACKNOWLEDGMENTS

We thank the anonymous reviewers for their constructive feedback and valuable comments, which helped improve this manuscript. We also thank Jun Hyeong Kim and Seonghwan Seo for helpful discussions and feedback that informed this work.

This work was supported by the Basic Science Research Program through the National Research Foundation of Korea (NRF), funded by the Ministry of Science and ICT (Grant Nos. RS-2022-NR068758, RS-2023-NR077040, RS-2023-00257479, RS-2023-00211868). S.K. was additionally supported by Grant No. N10250153. It was also supported by a grant from the Korean ARPA-H Project through the Korea Health Industry Development Institute (KHIDI), funded by the Ministry of Health and Welfare, the Republic of Korea (RS-2024-00512498).

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

# A  ADDITIONAL METHOD DETAILS

## A.1  COARSE-TO-FINE AUTOENCODER

To convert between atomistic and fragment-level representations, we introduce a coarse-to-fine autoencoder. Since the coarse graph already captures the abstract molecular structure, the primary role of the autoencoder is to restore atomistic connectivity information that is lost during the fragmentation process.

**Coarse-graph encoder**  In the encoding phase, each molecule is first decomposed into fragments using the BRICS decomposition rules (Degen et al., 2008), yielding a fragment-level graph $\mathcal{G}$. During this decomposition, connectivity information between atoms belonging to different fragments is discarded; for example, the relative orientation of an antisymmetric fragment with respect to others cannot be determined solely from the fragment graph. Unlike conventional autoencoders, our encoder network is therefore only required to encode the missing atomistic connectivity into the latent variable $z$, since the coarse graph already retains the overall molecular structure.

**Fine-graph decoder**  In the decoding phase, the goal is to restore atom-level connections between fragments that are linked at the coarse level. Each fragment is defined by junction atoms that mark the cut sites introduced during fragmentation. As a result, we only need to consider connectivity between junction atoms across neighboring fragments, rather than all possible atom pairs. From the coarse graph, we extract candidate junction-atom pairs, and the neural network predicts their connectivity as a continuous score conditioned on the latent variable $z$. These scores are then discretized into final bond assignments using the blossom algorithm, the details of which are provided below.

Formally, the encoding and decoding process is summarized as:

$$\textbf{Encoder:} \quad \mathbf{G} \xrightarrow{\text{Rule}} \mathcal{G}, \quad (\mathbf{G}, \mathcal{G}) \xrightarrow{\phi_{\text{enc}}} z,$$

$$\textbf{Decoder:} \quad (\mathcal{G}, z) \xrightarrow{\phi_{\text{dec}}} \text{score} \xrightarrow{\text{Blossom}} \mathbf{E}. \tag{4}$$

**Training details**  We train the coarse-to-fine autoencoder by minimizing reconstruction losses over atomistic connectivities. During training, each connectivity is treated as independent of the others, and the loss is formulated as a binary cross-entropy objective. In addition, we add a small KL regularization term, as in VAEs, to the training loss on the latent variable in order to enforce a well-structured and properly scaled latent space. The total loss is formulated as:

$$\mathcal{L}_{\text{AE}}(\phi) = \mathbb{E}_{\mathbf{G} \sim p_{\text{data}}} \left[ \sum_{(i,j) \in \mathcal{A}} \mathcal{L}_{\text{BCE}} \left( e_{ij}, \hat{e}_{ij} \right) + \beta D_{\text{KL}} \left( q_\phi(z|\mathbf{G}) \parallel p(z) \right) \right], \tag{5}$$

where $e_{ij}$ and $\hat{e}_{ij}$ denote the ground truth connectivity and predicted score between atoms $i$ and $j$, and $\mathcal{A}$ is the set of candidate junction-atom pairs derived from the coarse graph $\mathcal{G}$. We set a low regularization coefficient of $\beta = 0.0001$ to maintain high-fidelity reconstruction.

**Atom-level reconstruction by Blossom algorithm**  Although each connectivity is trained independently, reconstructing a valid graph from the predicted scores requires accounting for their dependencies. To this end, we employ the Blossom algorithm (Edmonds, 1965) to determine the optimal matching on the atom-level graph. More specifically, the Blossom algorithm returns a matching in which each atom is constrained to be paired with at most one partner, while maximizing the sum of connectivity scores (logit) predicted by the decoder. Within our framework, this procedure ensures accurate reconstruction of atom-level connectivity from fragment-level graphs, thereby yielding chemically valid molecular structures.

The algorithm takes as input the matching nodes $\mathbf{V}_m$, edges $\mathbf{E}_m$, and edge weights $w_{ij}$. Once the fragment-level graph and the probabilities of atom-level edges from the coarse-to-fine autoencoder are computed, we define $\mathbf{V}_m$ as the set of junction atoms in fragment graphs, which are marked as $\star$ in fig. 1, and $\mathbf{E}_m$ as the set of connections between junction atoms belonging to connected fragments.

Formally, an edge $e_{kl}$ exists in $\mathbf{E}_m$ if the corresponding atoms belong to different fragments that are connected in the fragment-level graph, expressed as:

$$e_{kl} \in \mathbf{E}_m \quad \text{if} \quad v_k \in \mathbf{V}_i, \quad v_l \in \mathbf{V}_j, \quad \text{and} \quad \varepsilon_{ij} \in \mathcal{E}, \tag{6}$$

where $\varepsilon_{ij}$ denotes the coarse level edge between $i$-th and $j$-th fragments, and $\mathbf{V}_i \subseteq \mathbf{V}_m$ denotes junction atoms that included in the $i$-th fragments. The edge weights $w_{ij}$ correspond to the predicted logit of each connection obtained from the coarse-to-fine autoencoder. The Blossom algorithm is then applied to solve the maximum weighted matching problem, formulated as

$$\mathbf{M}^* = \mathrm{argmax}_{\mathbf{M} \subseteq \mathbf{E}_m} \sum_{(i,j) \in \mathbf{M}} w_{ij} \quad \text{s.t.} \quad \deg_{\mathbf{M}}(v) \leq 1 \ \forall v \in \mathbf{V}_m. \tag{7}$$

Here, $\mathbf{M}^*$ represents the optimal set of fragment-level connections that best reconstructs atom-level connectivity, maximizing the joint probability of the autoencoder prediction. Although the algorithm has an $O(N^3)$ complexity for $N$ fragment junctions, its computational cost remains negligible in our case, as the number of fragment junctions is relatively small compared to the total number of atoms in a molecule.

## A.2 Details of Flow Modeling

In this section, we describe the details of flow modeling in the FragFM framework. The generation process jointly produces the coarse graph $\mathcal{G}$ and the latent vector $z$, where the generation of $\mathcal{G}$ can be interpreted as a joint procedure over nodes and edges. In total, three different types of variables are generated jointly: two discrete variables (node and edge states) and one continuous variable (the latent vector). Below, we detail the flow formulation and the corresponding loss functions for each type.

### A.2.1 Flow modeling for node

Following the original DFM formulation (Campbell et al., 2024), we specify the distribution of nodes (fragment types) at $t{=}1$ as $p_1(x)$ and define the $x_1$-conditioned time marginal by linear interpolation:

$$p_{t|1}(x_t \mid x_1) = t \, \delta(x_t, x_1) + (1{-}t) \, p_0(x_t), \tag{8}$$

where $p_0$ is a prior and $\delta(\cdot, \cdot)$ is the Kronecker delta. We adopt the masked version of DFM proposed by Campbell et al. (2024), in which the prior distribution assigns probability only to the mask token. With this choice, the interpolation distribution in eq. (8) becomes particularly simple: it is nonzero only when $x_t$ equals the mask token or the data token $x_1$.

The corresponding CTMC transition rate is provided by the authors,

$$R_t(x, y \mid x_1) = \frac{\mathrm{ReLU}\big(\partial_t p_{t|1}(y \mid x_1) - \partial_t p_{t|1}(x \mid x_1)\big)}{S \, p_{t|1}(x \mid x_1)}, \qquad \forall x \neq y, \tag{9}$$

with $S$ the number of states for which $p_{t|1}(x \mid x_1) > 0$. A brief algebraic manipulation of the Kolmogorov forward equation yields the $x_1$-unconditional generator

$$R_t(x, y) = \mathbb{E}_{x_1 \sim p_{1|t}(\cdot | x)}\big[R_t(x, y \mid x_1)\big]. \tag{10}$$

from which we can sample trajectories $\{x_t\}_{t \in [0,1]}$. Realizing these trajectories requires the posterior distribution $p_{1|t}(x_1 \mid X_t)$, where $X_t$ denotes the noisy version of the coarse-graph $\mathcal{G}_t$ and latent vector $z_t$. As described in the main text, we approximate this posterior by modeling the bag-conditioned distribution $p_{1|t,\mathcal{B}}(x_1 \mid X_t, \mathcal{B})$.

### A.2.2 Flow modeling for edge

Let $\varepsilon^{ij} \in \mathcal{E}$ denote the absence $(0)$ or presence $(1)$ of an edge between the $i$-th and $j$-th fragments in the coarse graph. Because the edge state is binary, we adopt a Bernoulli prior with probability density (mass) function $p_0(\varepsilon) = \frac{1}{2}$ for both states.

Following the discrete flow-matching (DFM) recipe, the $\varepsilon_1$-conditioned time marginal is

$$p_{t|1}\big(\varepsilon_t \mid \varepsilon_1\big) = t \, \delta\big(\varepsilon_t, \varepsilon_1\big) + (1 - t) \, p_0(\varepsilon_t), \qquad t \in [0, 1]. \tag{11}$$

The CTMC rate that realizes this marginal is

$$R_t\big(\varepsilon, \varepsilon' \mid \varepsilon_1\big) = \frac{\mathrm{ReLU}\big(\partial_t p_{t|1}(\varepsilon' \mid \varepsilon_1) - \partial_t p_{t|1}(\varepsilon \mid \varepsilon_1)\big)}{2 \, p_{t|1}(\varepsilon \mid \varepsilon_1)}, \qquad \forall \varepsilon \neq \varepsilon', \tag{12}$$

$$R_t(\varepsilon, \varepsilon') = \mathbb{E}_{\varepsilon_1 \sim p_{1|t}(\cdot | \varepsilon)}\big[R_t(\varepsilon, \varepsilon' \mid \varepsilon_1)\big]. \tag{13}$$

The posterior $p_{1\,|t}(\varepsilon_1^{ij} \mid X_t)$ is parameterized by a neural network $g_\theta^{\text{edge}}(X_t)_{ij}$. We trained the model by minimizing the cross-entropy loss:

$$\mathcal{L}_{\text{edge}} = \sum_{ij} \mathbb{E}_{X_1, X_t, t}\left[-\varepsilon_1^{ij} \log g_\theta^{\text{edge}}(X_t)_{ij} - (1 - \varepsilon_1^{ij}) \log\left(1 - g_\theta^{\text{edge}}(X_t)_{ij}\right)\right]. \tag{14}$$

In the sampling phase, the neural network replaces the posterior in eq. (13). Because $|\mathcal{E}| = 2$, the expectation above is computed exactly—no Monte-Carlo sampling is required. Thus, the forward kernel for $i, j$-th edge is as:

$$p_{t+\Delta t|t}^\theta(\varepsilon_{t+\Delta t}^{ij} \mid X_t) = \delta\left(\varepsilon_t^{ij}, \varepsilon_{t+\Delta t}^{ij}\right) + R_t\left(\varepsilon_t^{ij}, \varepsilon_{t+\Delta t}^{ij}\right)\Delta t. \tag{15}$$

### A.2.3 FLOW MODELING FOR LATENT VECTOR

Let $z \in \mathbb{R}^d$ be the continuous latent vector attached to the fragment-level graph. We model its evolution with conditional flow matching (CFM; Lipman et al. (2022)), which views generation as integrating an ODE whose time-dependent velocity field (VF) is learned from data. Specifically, we linearly change the mean and standard deviation $\mu_t(x) = tz_1$ and $\sigma_t(z) = 1 - t$. The corresponding conditioned target VF is

$$u_t(z_t|z_1) = \frac{z_1 - z_t}{1 - t}. \tag{16}$$

Then, the trajectory for a prior sample $z_0 \sim \mathcal{N}(\mathbf{0}, \mathbf{I})$ and a data sample $z_1 \sim p_1(z)$ under the target VF, *i.e.*, the solution to $\frac{dz_t}{dt} = u_t(z_t|z_1)$ with $z_0$ is given by:

$$z_t = (1 - t)z_0 + tz_1, \qquad t \in [0, 1]. \tag{17}$$

We fit a neural vector field $v_\theta(X_t)$ by minimizing the mean-squared error

$$\mathcal{L}_{\text{CFM}} = \mathbb{E}_{X_1, X_t, t}\left[\left\|v_\theta(X_t) - \frac{z_1 - z_t}{1 - t}\right\|_2^2\right]. \tag{18}$$

To generate a latent vector, we solve the ODE

$$\frac{\mathrm{d}\hat{z}_t}{\mathrm{d}t} = v_\theta(X_t), \tag{19}$$

forward from $t = 0$ to $t = 1$ with a deterministic solver. The resulting $\hat{z}_1$ is then fed to the coarse-to-fine decoding network to obtain an atom-level graph.

**CFM as a Limiting Case of a VE Diffusion Bridge**  Unlike diffusion models, which first define a reference process and then learn its drift, CFM directly prescribes the time-marginal distribution and optimizes the corresponding velocity fields that "point" toward a fixed data point. This raises a question: How can we treat CFM with the transition kernel $p_{t+\Delta t|t}(z_{t+\Delta t} \mid z_t)$ used in diffusion models?

A diffusion bridge is a reference diffusion process conditioned to hit a fixed endpoint. Its SDE is

$$\mathrm{d}z_t = \left[\mathbf{f}(z_t, t) + g^2(t)\,\nabla_{z_t} \log Q(z_T \mid z_t)\big|_{z_T = y}\right]\mathrm{d}t \;+\; g(t)\,\mathrm{d}\mathbf{w}_t, \tag{20}$$

where $Q(z_T \mid z_t)$ is the unconditioned transition kernel of the reference process, $\mathbf{f}$ its drift, and $g$ its diffusion coefficients.

If the reference process is the variance-exploding (VE) diffusion $\mathrm{d}z_t = g(t)\,\mathrm{d}\mathbf{w}_t$, Zhou et al. (2024) show that equation 20 reduces to

$$\mathrm{d}z_t = \frac{\mathrm{d}\sigma_t^2/\mathrm{d}t}{\sigma_T^2 - \sigma_t^2}\,(z_T - z_t)\,\mathrm{d}t \;+\; g(t)\,\mathrm{d}\mathbf{w}_t. \tag{21}$$

Setting $T = 1$ and $\sigma_t^2 = c^2 t$ (constant $c$) gives

$$\mathrm{d}z_t = \frac{z_1 - z_t}{1 - t}\,\mathrm{d}t \;+\; c\,\mathrm{d}\mathbf{w}_t. \tag{22}$$

Taking a limit of $c \to 0$ eliminates the stochastic term and leaves the deterministic drift

$$\frac{\mathrm{d}z_t}{\mathrm{d}t} = \frac{z_1 - z_t}{1 - t}, \tag{23}$$

which is exactly the velocity field optimized by CFM, *i.e.*, the VE diffusion bridge collapses to CFM.

Given a coupling $\pi(z_0, z_1)$, we can form a mixture bridge $\Pi$ by averaging the pinned-down trajectories over $\pi$. According to Proposition 2 of Shi et al. (2023), its Markov approximation $M$ satisfies

$$\mathrm{d}z_t = \mathbb{E}_{1|t}\left[\frac{z_1 - z_t}{1 - t}\right]\mathrm{d}t + c\,\mathrm{d}\mathbf{w}_t, \quad \text{with } M_t = \Pi_t \ \forall t. \tag{24}$$

When $c \to 0$, the drift term above coincides with the averaged velocity field learned by CFM eq. (18), confirming that $M$ recovers the CFM dynamics in the zero-noise limit.

### A.2.4 TRAINING DETAILS

Our objective is to learn a generative diffusion on the coarse graph state (Recall $X_t = (\mathcal{G}_t, z_t)$ with $\mathcal{G}_t = (\{x_t^i\}, \{\varepsilon_t^{ij}\})$ —the node-type vector $x_t^i$, binary edge matrix $\varepsilon_t^{ij}$, and latent vector $z_t \in \mathbb{R}^d$). By combining the node-type Info-NCE loss, the edge binary-cross-entropy loss, and the latent CFM loss into a single training objective.

**Sampling a training triple** $(X_1, X_t, t)$**.**

1. **Data endpoint.** Sample a atomistic graph $\mathbf{G}_1$ from the molecular dataset, and apply coarse-to-fine encoder to obtain $X_1 = (\mathcal{G}_1, z_1)$.

2. **Time sampling.** Sample a time $t \in [0, 1]$ from a uniform distribution.

3. **Forward noise.** Independently transform each component to its noised counterpart:
   - **Node.** For every node indexed with $i$, sample $x_t^i \sim p_{t|1}(\cdot \mid x_1^i)$ with the masked prior.
   - **Edge.** For each pair $(i, j)$, draw $\varepsilon_t^{ij} \sim p_{t|1}(\cdot \mid \varepsilon_1^{ij})$ using eq. (11).
   - **Latent.** Sample $z_0 \sim \mathcal{N}(\mathbf{0}, \mathbf{I})$ and set $z_t = (1 - t)z_0 + tz_1$ as in eq. (17).

4. **Construct $X_t$.** Collect the three noised components into $X_t = (\mathcal{G}_t, z_t)$.

**Joint loss.**

$$\mathcal{L}_{\text{node}}(\theta; \mathcal{B}) = -\log \frac{f_\theta(X_t, x_1)}{\sum_{y \in \mathcal{B}} f_\theta(X_t, y)}, \tag{2}$$

$$\mathcal{L}_{\text{edge}}(\theta) = \sum_{i<j}\left[-\varepsilon_1^{ij} \log g_\theta^{\text{edge}}(X_t)_{ij} - (1 - \varepsilon_1^{ij})\log\left(1 - g_\theta^{\text{edge}}(X_t)_{ij}\right)\right], \tag{14}$$

$$\mathcal{L}_{\text{latent}}(\theta) = \left\|v_\theta^{\text{latent}}(X_t) - (z_1 - z_0)\right\|_2^2. \tag{18}$$

We minimize the weighted sum

$$\mathcal{L}_{\text{total}}(\theta; \mathcal{B}) = \mathcal{L}_{\text{node}}(\theta; \mathcal{B}) + \alpha_{\text{edge}}\,\mathcal{L}_{\text{edge}}(\theta) + \alpha_{\text{latent}}\,\mathcal{L}_{\text{latent}}(\theta), \tag{25}$$

with $\alpha_{\text{edge}}, \alpha_{\text{latent}} > 0$. We fix $\alpha_{\text{edge}} = 5.0$ and $\alpha_{\text{latent}} = 1.0$ for all of our experiments.

### A.3 CONDITIONAL GENERATION WITH FRAGMENT BAG

While generating valid molecules is essential, steering them toward the desired property is crucial for the practical application of molecular generative models. Direct conditioning via classifier-free guidance (Ho & Salimans, 2022) offers strong control but tightly binds the generative network to specific properties and often requires retraining when new targets are introduced. Instead, we adopt classifier guidance, steering generation at sampling time with an external property predictor. This decouples the generator from any single conditioning signal and allows the predictor to be trained or updated independently (Dhariwal & Nichol, 2021; Vignac et al., 2023).

To generate molecules conditioned on external properties, we require a property-steered transition kernel, denoted $p_{t+\Delta t|t}(X_{t+\Delta t} \mid X_t, c)$. The transition kernel of $X$ factorizes into kernels for nodes, edges, and the latent vector. Among these, node types require special treatment because they are sampled from a candidate fragment bag $\mathcal{B}$.

Analogous to the unconditional case described in the main text, we define the property-conditioned forward kernel using the bag strategy as

$$p_{t+\Delta t|t}(X_{t+\Delta t} \mid X_t, c) \approx p(X_{t+\Delta t} \mid X_t, \mathcal{B}^c, c), \tag{26}$$

where $\mathcal{B}^c$ denotes the property-conditioned fragment bag. The original $\mathcal{B}$ is constructed by sampling one positive fragment from $x^+ \sim p(x \mid X_t)$ and $N-1$ negative fragments from $x^- \sim p(x)$. Analogously, the $c$-conditioned bag $\mathcal{B}^c$ is constructed by sampling one positive fragment from $x^+ \sim p(x \mid X_t, c)$ and $N-1$ negative fragments from $x^- \sim p(x \mid c)$. By Bayes' rule, the $c$-conditioned in-bag transition kernel can be factorized into the $c$-unconditional kernel and a guidance ratio:

$$p(X_{t+\Delta t} \mid X_t, \mathcal{B}^c, c) = \underbrace{p(X_{t+\Delta t} \mid X_t, \mathcal{B}^c)}_{\text{eq. (3)}} \cdot \underbrace{\frac{p(c \mid X_{t+\Delta t}, X_t, \mathcal{B}^c)}{p(c \mid X_t, \mathcal{B}^c)}}_{\text{Guidance ratio}}. \tag{27}$$

**Guidance ratio modeling**  The guidance ratio in eq. (27) can be written by:

$$\frac{p(c|X_{t+\Delta t}, X_t, \mathcal{B}^c)}{p(c|X_t, \mathcal{B}^c)} = \frac{p(c|X_{t+\Delta t}, \mathcal{B}^c)}{p(c|X_t, \mathcal{B}^c)},$$

$$= \frac{p(c|X_{t+\Delta t})p(\mathcal{B}^c|c, X_{t+\Delta t})/p(\mathcal{B}^c|X_{t+\Delta t})}{p(c|X_t)p(\mathcal{B}^c|c, X_t)/p(\mathcal{B}^c|X_t)}.$$

The ratio $\frac{p(\mathcal{B}^c|c, X_{t+\Delta t})\, p(\mathcal{B}^c|X_t)}{p(\mathcal{B}^c|X_{t+\Delta t})\, p(\mathcal{B}^c|c, X_t)}$ is intractable, yet it involves the difference between two consecutive states $X_t$ and $X_{t+\Delta t}$. Because an Euler step is very small, it can be assumed that the diffusion state evolves smoothly: $X_{t+\Delta t} = X_t + \mathcal{O}(\Delta t)$. If the bag-sampling distributions $p(\mathcal{B}^c \mid X)$ and $p(\mathcal{B}^c \mid c, X)$ vary continuously with $X$, a first-order Taylor expansion yields

$$p(\mathcal{B}^c \mid \cdot, X_{t+\Delta t}) = p(\mathcal{B}^c \mid \cdot, X_t) + \mathcal{O}(\Delta t), \tag{28}$$

So the whole ratio is approximately 1, to be

$$\frac{p(c|X_{t+\Delta t}, X_t, \mathcal{B}^c)}{p(c|X_t, \mathcal{B}^c)} \approx \frac{p(c|X_{t+\Delta t})}{p(c|X_t)}. \tag{29}$$

Following Nisonoff et al. (2024); Vignac et al. (2023), we can estimate the ratio via noisy predictor $\hat{c}(X_t)$ with 1st order Taylor expansion, yielding

$$\log \frac{p(c|X_{t+\Delta t})}{p(c|X_t)} \approx \langle \nabla_{X_t} \log p(c|X_t), X_{t+\Delta t} - X_t \rangle,$$

$$\approx \sum_i \langle \nabla_{x_t^{(i)}} \log p(c|X_t), x_{t+\Delta t}^{(i)} \rangle + \sum_{ij} \langle \nabla_{\varepsilon_t^{(ij)}} \log p(c|X_t), \varepsilon_{t+\Delta t}^{(ij)} \rangle$$

$$+ \langle \nabla_{z_t} \log p(c|X_t), z_{t+\Delta t} - z_t \rangle + C.$$

In practice, we estimate $p(c|X_t)$ by Gaussian modeling with a time-conditioned noisy classifier parameterized by $\psi$, $\mathcal{N}(c; \mu_X(X_t, t; \psi), \sigma^2)$. Thus, the guidance term is written as:

$$\frac{p(c|X_{t+\Delta t})}{p(c|X_t)} \propto \exp(-\lambda_X \sum_i \langle \nabla_{x_t^{(i)}} \|\mu_X(X_t, t) - c\|^2, x_{t+\Delta t}^{(i)} \rangle) \tag{30}$$

$$\times \exp(-\lambda_X \sum_{ij} \langle \nabla_{\varepsilon_t^{(ij)}} \|\mu_X(X_t, t) - c\|^2, \varepsilon_{t+\Delta t}^{(ij)} \rangle)$$

$$\times \exp(-\lambda_X \langle \nabla_{z_t} \|\mu_X(X_t, t) - c\|^2, z_{t+\Delta t} - z_t \rangle),$$

where $\lambda_X$ controls the strength of the guidance.

The smoothness assumption we adopt is exactly the one adopted by earlier discrete-guidance methods (Vignac et al., 2023; Nisonoff et al., 2024), our derivation remains consistent with the foundations laid out in those works.

**Conditional bag sampling** To sample $\mathcal{B}^c$, we first recall the unconditional case. When no property is specified, a bag $\mathcal{B} = \{x_1, \ldots, x_N\}$ of size $N$ is drawn without replacement from the fragments vocabulary $\mathcal{F}$ with probability

$$P(\mathcal{B}) = \frac{\prod\limits_{x \in \mathcal{B}} p(x)}{\sum\limits_{\substack{\mathcal{S} \subset \mathcal{F} \\ |\mathcal{S}| = N}} \prod\limits_{y \in \mathcal{S}} p(y)}, \tag{31}$$

*i.e.*, bags that contain high-frequency fragments under the marginal distribution $p(x)$ are sampled more often. To steer this toward a desired property $c$, we replace each $p(x)$ by its conditional counterpart $p(x|c)$. Viewing a fragment as part of a molecule $X$, the condition can be written as the expected property value over all molecules that contain that fragment: $\sum_X p(x|X, c)p(X|c)$.

The resulting bag distribution becomes

$$P(\mathcal{B}^c \mid c) = \frac{\prod\limits_{x \in \mathcal{B}^c} p(x \mid c)}{\sum\limits_{\substack{\mathcal{S} \subset \mathcal{F} \\ |\mathcal{S}| = N}} \prod\limits_{y \in \mathcal{S}} p(y \mid c)}. \tag{32}$$

Applying Bayes' rule and dropping the constant factor $p(c)$ gives

$$p(x|c) \propto p(x)p(c|x). \tag{33}$$

In practice, we estimate $p(c|x)$ as a Gaussian distribution with a light neural regressor parameterized by $\psi$,

$$p_\psi(c \mid x) = \mathcal{N}(c; \mu_\mathcal{B}(x; \psi), \sigma^2), \quad p_\psi(x \mid c) \propto p(x) \exp(-\lambda_\mathcal{B} \| \mu_\mathcal{B}(x; \psi) - c \|^2), \tag{34}$$

where $\mu_\mathcal{B}(x, \psi)$ is the predicted mean, $\sigma^2$ is a fixed variance, and $\lambda_\mathcal{B}$ controls the strength of the property-guided bag selection.

To parameterize the guidance ratio in eq. (30), we train a time-conditioned noisy property regressor $\mu_X(X_t, t)$ that predicts the target property $c$ from the corrupted molecule state $X_t$:

$$\mathcal{L}_X(\psi) = \mathbb{E}_{(X,c) \sim \mathcal{D}} \, \mathbb{E}_{t \sim \mathcal{U}(0,1)} \, \mathbb{E}_{X_t \sim q_t(\cdot|X)} \big[ \| \mu_X(X_t, t; \psi) - c \|^2 \big]. \tag{35}$$

We also train a fragment-level regressor $\mu_\mathcal{B}(x)$ that estimates the global property $c$ from each fragment, as required for the conditional fragment guidance in eq. (34):

$$\mathcal{L}_\mathcal{B}(\psi) = \mathbb{E}_{(X,c) \sim \mathcal{D}} \left[ \sum_{x \in \mathcal{F}(X)} \| \mu_\mathcal{B}(x; \psi) - c \|^2 \right]. \tag{36}$$

Both objectives are optimized jointly during training with shared parameters. Architectural details are provided in section E.2, and the corresponding hyperparameters are summarized in table 17.

### A.4 DETAILED BALANCE

The space of valid rate matrices extends beyond the original formulation of eq. (9); thus, alternative constructions can still satisfy the Kolmogorov equation. Campbell et al. (2024) show that if a matrix $R_t^{DB}$ fulfils the detailed-balance identity:

$$p_{t|1}(x_t \mid x_1) R_t^{DB}(x_t, y \mid x_1) = p_{t|1}(y \mid x_1) R_t^{DB}(y, x_t \mid x_1), \tag{37}$$

then,

$$R_t^\eta = R_t^* + \eta R_t^{DB}, \quad \eta \in \mathbb{R}^+, \tag{38}$$

remains a valid CTMC generator. A larger $\eta$ injects extra stochasticity, opening additional state-transition pathways.

Although several designs are possible, we follow Campbell et al. (2024). The only non-zero rates for fragment-type nodes are the transitions between a concrete type $x_1$ and the mask state $M$:

$$R_t^{DB}(x, y|x_1) = \delta(x, x_1)\delta(y, M) + \delta(x, M)\delta(y, x_1), \tag{39}$$

where $M$ denotes the masked type.

For edges, whose states are binary $\varepsilon^{ij} \in \{0, 1\}$, we consider a flip rate $\eta_{\text{edge}}$ and a matching backward rate that satisfies eq. (37) which leads to:

$$R_t^{DB}(\varepsilon, \varepsilon' \mid \varepsilon_1) = \delta(\varepsilon, \varepsilon_1) + \frac{1+t}{1-t}\delta(\varepsilon', \varepsilon_1). \tag{40}$$

We set $(\eta_{\text{node}}, \eta_{\text{edge}}) = (20.0, 20.0)$ for the MOSES and ZINC250k datasets, and $(2.0, 20.0)$ for the GuacaMol and NPGen datasets.

## B   DETAILS OF NPGEN BENCHMARK

Here, we provide additional details on (1) the dataset construction process from the COCONUT database, including the filtering steps, (2) the NP-specific evaluation metrics and how they are computed, (3) the statistics and distributional properties of the NPGen benchmark, and (4) the baseline models and setups used in our evaluation. These details complement the high-level description in the main paper and are intended to facilitate reproducibility and further use of NPGen by the community.

### B.1   BACKGROUND ON NATURAL PRODUCTS

Natural products (NPs) are chemical compounds biologically synthesized by organisms, *e.g.*, plants, fungi, and bacteria. They have long served as a valuable resource in drug discovery, with their unique structural features compared to typical synthetic compounds—more complex ring architectures, higher heteroatom density, and abundant oxygen-based functional groups that contribute to polarity, stereochemical diversity, and bioactivity (Feher & Schmidt, 2003; Stratton et al., 2015). In essence, NPs occupy a biologically meaningful subset of chemical space, often resembling endogenous metabolites, which makes them particularly effective as templates or direct sources for drug design (Boufridi & Quinn, 2018; Rosén et al., 2009; Atanasov et al., 2021). Moreover, unlike many synthetic compounds, NPs frequently resemble endogenous metabolites, increasing their likelihood of interacting with biological targets in meaningful ways.

A substantial portion of clinically approved small-molecule drugs are inspired by or derived from NPs, mimicking their structures or functionalities (Newman & Cragg, 2020). Multiple classes of drugs, such as antibiotics (penicillin, derived from *Penicillium* fungi; erythromycin, from *Saccharopolyspora erythraea* bacteria), anticancer agents (paclitaxel, originally isolated from the *Taxus brevifolia* (Pacific yew tree); doxorubicin, from *Streptomyces peucetius* bacteria; vincristine, isolated from *Catharanthus roseus* (periwinkle)), and immunosuppressants (cyclosporin A, from the soil fungus *Tolypocladium inflatum*) demonstrate the long-standing role of NPs in drug discovery. Consistent with the above, analyses of FDA approvals indicate that nearly one-third of all small-molecule drugs introduced since the 1980s are natural-product-derived or natural-product-inspired, underscoring their sustained therapeutic relevance (Atanasov et al., 2021). In this regard, generative models capable of reflecting NP-specific structural and biological characteristics represent an important frontier for computational drug discovery.

For the machine learning community, natural products (NPs) present a particularly valuable and challenging benchmark: their structural diversity, functional complexity, and biological relevance extend far beyond the synthetic-like molecules that dominate current datasets. While modern generative models have achieved strong performance on widely used benchmarks such as MOSES (Polykovskiy et al., 2020) and GuacaMol (Brown et al., 2019), these datasets are largely composed of small, structurally simple molecules, and their evaluation metrics—Fréchet ChemNet Distance, scaffold overlap, and KL divergence on basic descriptors—are approaching saturation. As a result, they cannot adequately assess models aiming to capture the complexity and biological meaningfulness of NPs. A benchmark centered on NPs, therefore, not only reflects real-world NP inspired drug discovery needs but also provides a domain-specific stress test for molecular generative models, probing their ability to extrapolate to richer regions of chemical space (Bechler-Speicher et al., 2025).

## B.2 DATASET CONSTRUCTION

To construct the NPGen dataset, we utilized the 2024/12/31 version of the COCONUT database (Sorokina et al., 2021; Chandrasekhar et al., 2025), which comprises 695,120 natural product-like molecules. Given that the original database contains compounds with transition metals—species that are rarely encountered in typical organic natural products—we applied a filtering procedure to retain only molecules composed exclusively of non-metal atoms: `B`, `C`, `N`, `O`, `F`, `Si`, `P`, `S`, `Cl`, `As`, `Se`, `Br`, `I`. Additionally, to exclude arbitrarily large macromolecules, we retained only those molecules whose heavy-atom counts fell between 2 and 99.

Furthermore, we consider only neutral molecules, excluding salts and molecules containing "." in their SMILES representations. After filtering, 658,566 molecules were retained. The resulting dataset was randomly partitioned into training, validation, and test subsets using an 80:5:15 split under the assumption of i.i.d. sampling, yielding 526,852, 32,928, and 98,786 molecules, respectively.

## B.3 IMPLEMENTATION DETAILS FOR BASELINES

As baselines, we selected a set of molecular graph generative models with two aspects, *i.e.*, generation strategy (autoregressive and one-shot) and representation level (atom and fragment). We provide more details on the baseline models.

**GraphAF** (Shi et al., 2020) is a flow-based autoregressive model for molecular graph generation that constructs molecules sequentially by adding atoms and their corresponding bonds. We used the authors' official implementation from (`https://github.com/DeepGraphLearning/GraphAF`) with its default settings, extending only the preprocessing and generation steps to include atom types that the original implementation does not support `B`, `As`, `Si`, `Se`. During generation, the official implementation terminates sampling once 40 atoms have been generated per molecule; we increased this limit to 99 to match the NPGen benchmark's maximum heavy-atom count.

**JT-VAE** (Jin et al., 2018) is a fragment-based autoregressive variational autoencoder that generates molecules by building a junction tree of chemically meaningful substructures and then assembling the corresponding atom-level graph (Jin et al., 2018). We used the authors' official implementation (`https://github.com/wengong-jin/icml18-jtnn`) with the acceleration module (`fast_molvae`). Because the codebase relies on Python 2 and is incompatible with newer GPU drivers, we performed training and sampling on an NVIDIA GeForce RTX 2080 Ti. We performed a random hyperparameter search over `hidden_dim` and `batch_size`, and report the best results.

**HierVAE** (Jin et al., 2020a) builds on JT-VAE by introducing a hierarchical latent space and a scaffold-aware message-passing scheme to boost structural diversity and sampling fidelity. We used the authors' official implementation (`https://github.com/wengong-jin/hgraph2graph`), extending only the preprocessing step to include the `As` atom type. By default, HierVAE employs a greedy motif-sampling strategy, which prioritizes the top-scoring fragments and may bias the output distribution. We observed that this led to artifacts, only generating single carbon chains on the NPGen benchmark. To provide a fair comparison, we report the results of the alternative stochastic-sampling mode (enabled via a single option flag in the official implementation), without modifying the core codebase.

**SAFE-GPT** (Noutahi et al., 2024) is an autoregressive sequence generative model that produces SAFE (Sequential Attachment-based Fragment Embedding) strings—a novel representation proposed in the paper, where molecules are expressed as sequences of fragments. The model is built on a GPT-2–like transformer architecture. While the original work reported only basic results on validity, uniqueness, and diversity, we trained the SAFE-GPT-20M model using its official implementation to obtain comprehensive benchmark results. We trained the model using AdamW (learning rate 5e-4) with a linear warmup schedule over the first 10k steps. The tokenizer was constructed for the NPGen benchmark, and training was conducted for 40k steps with a maximum sequence length of 275. For every step in data processing, training, and sampling, we used the original code from `https://github.com/datamol-io/safe`.

**MolHF** (Zhu et al., 2023) is a hierarchical normalizing flow model designed for molecular graph generation. It employs a coarse-to-fine strategy, generating a fragment connectivity graph and subsequently refining it at the atom level via conditional flows. We trained MolHF with its default

setting provided in its `https://github.com/violet-sto/MolHF`. The model uses conditional latent variables, $1 \times 1$ convolution, rotation of node features, and a learned prior. Since the NPGen dataset has larger molecules than the previous benchmarks, we had to decrease the batch size, resulting in the training with Adam for 200 epochs (batch size $64$, learning rate $2e - 4$).

**DiGress** (Vignac et al., 2023) is an atom-based generative model that employs discrete diffusion. We run the authors' official implementation ( `https://github.com/cvignac/DiGress` ) with all default hyperparameters, adding the atom types 'B', 'As' and their corresponding charges.

**DeFoG** (Qin et al., 2024) is an atom-based generative model that employs discrete flow matching. We run the authors' official implementation ( `https://github.com/manuelmlmadeira/DeFoG` ) with all default hyperparameters, adding the atom types 'B', 'As' and their corresponding charges.

### B.4 METRICS

As mentioned in the main text, we utilize two methods for distributional metrics: NP-likeness score (Ertl et al., 2008) and NP Classifier (Kim et al., 2021). Both strategies are developed by domain experts to effectively analyze the molecule through the lens of a natural product.

**NP-likeness score** is developed to quantify the similarity of a given molecule to the structural space typically occupied by natural products. Since one of the major differences between NPs and synthetic molecules is structural features such as the number of aromatic rings, stereocenters, and distribution of nitrogen and oxygen atoms, the NP-likeness of a molecule is calculated as the sum of the contributions of its constituent fragments, where each fragment's contribution is based on its frequency in natural product versus synthetic molecule databases. We show the distributions of NP-likeness scores for existing benchmarks and NPGen in fig. 9. Since the source database of NPGen, COCONUT, aggregates compounds from diverse sources, it naturally contains molecules across a wide range of NP-likeness values. Importantly, the goal of a generative model is to faithfully reproduce the dataset's distribution rather than simply maximizing NP-likeness. To this end, we measure the Kullback–Leibler (KL) divergence between the distributions of NP-likeness scores for generated and reference molecules.

Since NP-likeness scores are continuous, we adopt a non-parametric estimation procedure from the prior benchmarks, such as MOSES and GuacaMol. Specifically, we calculate NP-likeness scores for all molecules in both sets, then estimate their probability density functions (PDFs) using Gaussian kernel density estimation (KDE). We evaluate both PDFs on a common range discretized into 1,000 points spanning the observed minimum and maximum scores, adding a small constant ($10^{-10}$) for numerical stability. The resulting discrete distributions are compared via KL divergence using the `scipy.stats.entropy` function, providing a robust measure of how closely the generated distribution aligns with that of natural products in the reference set.

**NPClassifier** is a deep learning-based tool specifically designed to classify NPs. It categorizes molecules at three hierarchical levels—Pathway (7 categories; *e.g.*, Polyketides, Terpenoids), Superclass (70 categories, *e.g.*, Macrolides, Diterpenoids), and Class (672 categories; *e.g.*, Erythromycins, Kaurane diterpenoids)—reflecting the biosynthetic origins, broader chemical and chemotaxonomic properties, and specific structural families recognized by the NP research community. This multi-level system, built on an NP-specific ontology and trained on over 73,000 NPs using counted Morgan fingerprints, provides a classification based on knowledge of natural products, including their biosynthetic relationships and structural diversity.

We employed Kullback-Leibler (KL) divergence as a metric for both methods. We compute the KL divergence for the NP-likeness score and the NPClassifier differently, as they are continuous and discrete values, respectively. It is worth noting that NPClassifier often predicts 'Unclassified', which indicates a molecule is not included in any classes, along with multiple class results (*e.g.*, 'Peptide alkaloids, Tetramate alkaloids' in Class). We treat all prediction results as another unique class, since molecules can have multiple structural features.

## B.5 DATASET STATISTICS

We analyzed the distributions of several molecular properties to highlight NPGen's distinctions from standard molecular generative benchmarks (MOSES and GuacaMol). These properties fell into two categories: (1) simple molecular descriptors, such as the number of atoms, molecular weight, and number of hydrogen bond acceptors and donors (fig. 8), and (2) functionality-related properties, including NP-likeness scores and NPClassifier prediction results (fig. 9). Consistent with the nature of NPs, which are generally larger and more complex than typical synthetic drug-like molecules, NPGen molecules are, on average, larger in terms of the number of atoms and molecular weight compared to those in MOSES and GuacaMol (figs. 8a and 8b). Furthermore, molecules in NPGen exhibit higher numbers of hydrogen bond acceptors and donors (see figs. 8c and 8d), reflecting another characteristic of NPs.

The difference between benchmarks becomes more significant when examining functionality-related properties. NPClassifier predictions for Pathway (fig. 9a) indicate that NPGen molecules span a diverse range of NP categories. In contrast, molecules from MOSES and GuacaMol mostly fall into `Alkaloids`, which are non-peptidic nitrogenous organic compounds, or remain unclassified. Focusing on four selected Superclass categories for which NPClassifier had demonstrated high predictive performance (F1 score higher than 0.95 for categories with more than 500 compounds (Kim et al., 2021)), NPGen shows higher proportions of molecules in these specific categories. Conversely, molecules from the other benchmarks mostly fall into `Unclassified`, implying that they are dissimilar to NPs. The NP-likeness score further emphasizes this divergence (fig. 9c). In particular, NPGen's distribution is largely shifted toward higher scores (average: 1.14) compared with MOSES (average: -1.67) and GuacaMol (average: -0.90), where higher scores indicate greater similarity to NPs.

Additionally, we visualize the chemical space of existing benchmarks (MOSES, GuacaMol) and NPGen using UMAP (McInnes et al., 2018) in Figure 2a. While MOSES and GuacaMol largely overlap, NPGen extends into distinct regions, indicating coverage of different chemical subspaces.

These statistical analyses demonstrate that NPGen has distinct features compared with existing molecular generative benchmarks, establishing its suitability as a unique molecular graph generative benchmark targeting NP-like chemical space.

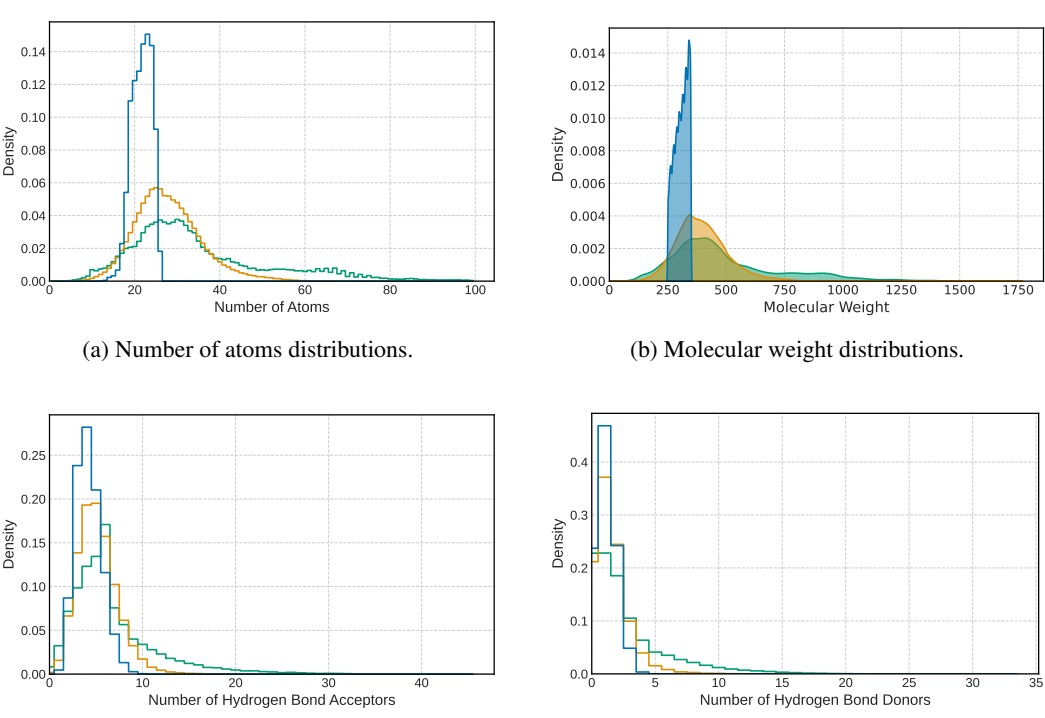

(a) Number of atoms distributions.

(b) Molecular weight distributions.

(c) Number of hydrogen bond acceptors distributions.

(d) Number of hydrogen bond donors distributions.

Figure 8: **Comparison of simple molecular property distributions among three benchmarks: MOSES, GuacaMol, and NPGen**. The number of molecules in each dataset is 1,936,962, 1,591,378, and 658,566, respectively.

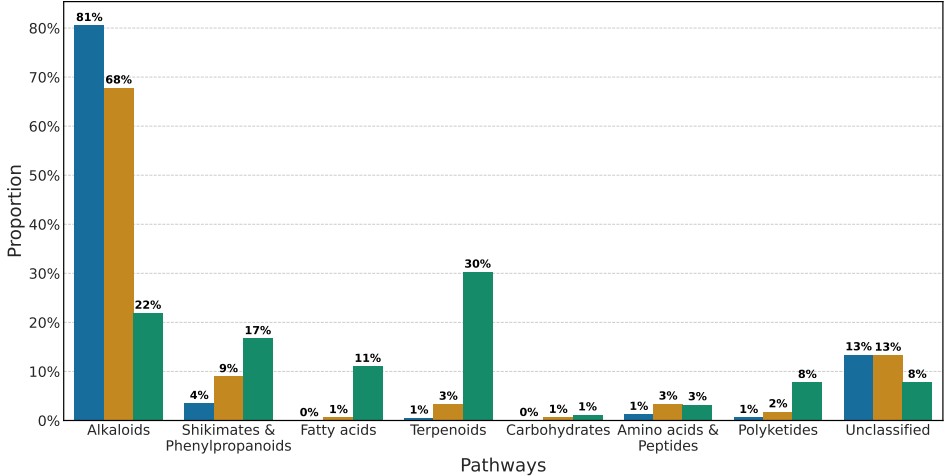

(a) Proportion of all Pathway, along with `Unclassified`. Note that we excluded predicted results with multiple categories, for clarity. Specifically, number of predictions decreased as 1,936,962 to 1,873,287, 1,591,378 to 1,542,118, and 658,566 to 628,840 for MOSES, GuacaMol, and NPGen, respectively.

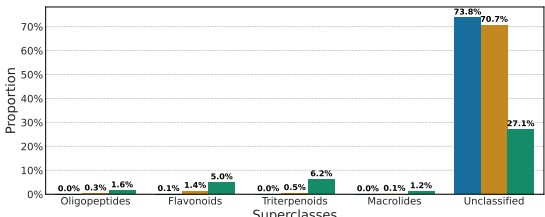

(b) Proportion of four Superclass, along with `Unclassified`. Note that we excluded predicted results with multiple categories, for clarity. Specifically, number of predictions decreased as 1,936,962 to 1,933,854, 1,591,378 to 1,588,486, and 658,566 to 650,013 for MOSES, GuacaMol, and NPGen, respectively.

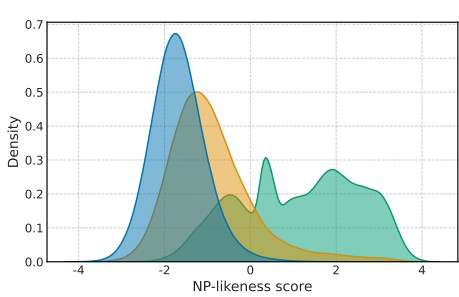

(c) NP-likeness score distribution.

Figure 9: **Comparison of NP-likeness score and NPClassifier prediction results among three benchmarks: MOSES, GuacaMol, and NPGen**. The number of molecules in each dataset is 1,936,962, 1,591,378, and 658,566, respectively. Note that we also report the ratio of unclassified entities in the datasets in figs. 9a and 9b. Class prediction statistics are not included since there are 687 classes and the ratio of each class is too small compared to that of the `Unclassified` class.

## C   ADDITIONAL RESULTS AND ANALYSES

### C.1   ADDITIONAL BENCHMARKS

In the GuacaMol (table 4) and ZINC250k benchmarks, similar to MOSES, FragFM achieved the best performance among diffusion- and flow-based baselines in terms of Validity and V.U. metrics, obtained a state-of-the-art FCD score, and ranked as a close second in the KL Div. score on GuacaMol. On ZINC250k, FragFM achieved the best performance across all reported metrics, surpassing the strongest atom-based baseline with a five-fold improvement in NSPDK and a two-fold reduction in FCD. These results underscore the effectiveness of the fragment-based approach of FragFM in generating valid and chemically meaningful molecules. Visualization results for GuacaMol are provided in section F.3.

Table 4: **Molecule generation results on the GuacaMol benchmark**. We use a total of 10,000 generated molecules for evaluation. All baselines except MCTS in this table are one-shot methods. The results for FragFM are averaged over three independent runs. The best performance is shown in **bold**, and the second-best is underlined.

| Model | Rep. Level | Val. ↑ | V.U. ↑ | V.U.N. ↑ | KL Div. ↑ | FCD ↑ |
|---|---|---|---|---|---|---|
| Training set | - | 100.0 | 100.0 | - | 99.9 | 92.8 |
| MCTS (Jensen, 2019) | Atom | **100.0** | **100.0** | 95.4 | 82.2 | 1.5 |
| NAGVAE (Kwon et al., 2020) | Atom | 92.9 | 88.7 | 88.7 | 38.4 | 0.9 |
| DiGress (Vignac et al., 2023) | Atom | 85.2 | 85.2 | 85.1 | 92.9 | 68.0 |
| DisCo (Xu et al., 2024) | Atom | 86.6 | 86.6 | 86.5 | 92.6 | 59.7 |
| Cometh (Siraudin et al., 2024) | Atom | 94.4 | 94.4 | 93.5 | 94.1 | 67.4 |
| Cometh-PC (Siraudin et al., 2024) | Atom | 98.9 | 98.9 | 97.6 | 96.7 | 72.7 |
| DeFoG (Qin et al., 2024) | Atom | 99.0 | 99.0 | **97.9** | **97.7** | 73.8 |
| FragFM (ours) | Fragment | 99.7 | 99.3 | 95.0 | 97.4 | **85.8** |

Table 5: **Molecule generation on ZINC250k benchmark**. We use 25,000 generated molecules for evaluation. The upper part comprises autoregressive methods, while the second part comprises one-shot methods, including diffusion-based and flow-based methods. The best performance is highlighted in **bold**, and the second-best is underlined.

| Model | Rep. Level | Valid ↑ | NSPDK ↓ | FCD ↓ |
|---|---|---|---|---|
| Training set | - | - | 0.0001 | 0.062 |
| GraphAF (Shi et al., 2020) | Atom (Graph) | 67.92 | 0.0432 | 16.128 |
| GraphDF (Luo et al., 2021) | Atom (Graph) | 89.72 | 0.1737 | 33.899 |
| MolHF (Zhu et al., 2023) | Atom (Graph) | 94.75 | 0.0709 | 22.230 |
| GDSS (Jo et al., 2022) | Atom (Graph) | 97.12 | 0.0192 | 14.032 |
| GSDM (Luo et al., 2023) | Atom (Graph) | 92.57 | 0.0168 | 12.435 |
| GruM (Jo et al., 2023) | Atom (Graph) | 98.32 | 0.0023 | 2.235 |
| SwinGNN (Yan et al., 2023) | Atom (Graph) | 86.16 | 0.0047 | 4.398 |
| DiGress (Vignac et al., 2023) | Atom (Graph) | 94.98 | 0.0021 | 3.482 |
| GGFlow (Hou et al., 2024) | Atom (Graph) | 99.63 | 0.0010 | 1.455 |
| FragFM (ours) | Fragment (Graph) | **99.81** | **0.0002** | **0.630** |

We also report the mean and standard deviation of the newly proposed NPGen benchmark across three runs for all baselines and FragFM in table 6.

Table 6: Molecule generation results on NPGen with error ranges. We use a total of 30,000 molecules for evaluation. The upper part comprises autoregressive methods, while the second part comprises one-shot methods, including diffusion-based and flow-based methods. The results are averaged over three runs. The numbers with ± indicates the standard deviation against each runs.

| Model | Val. ↑ | Unique. ↑ | Novel ↑ | NP Score KL Div. ↓ | NP Class KL Div. ↓ | | | FCD ↓ |
| | | | | | Pathway | Superclass | Class | |
| --- | --- | --- | --- | --- | --- | --- | --- | --- |
| Training set | 100.0 | 100.0 | - | 0.0006 | 0.0002 | 0.0028 | 0.0094 | 0.01 |
| GraphAF (Shi et al., 2020) | $79.1_{\pm 0.1}$ | $63.6_{\pm 0.2}$ | $95.6_{\pm 0.0}$ | $0.8546_{\pm 0.0095}$ | $0.9713_{\pm 0.0055}$ | $3.3907_{\pm 0.0730}$ | $6.6905_{\pm 0.0905}$ | $25.11_{\pm 0.08}$ |
| JT-VAE (Jin et al., 2018) | $100.0_{\pm 0.0}$ | $97.2_{\pm 0.1}$ | $99.5_{\pm 0.0}$ | $0.5437_{\pm 0.0188}$ | $0.1055_{\pm 0.0019}$ | $1.2895_{\pm 0.1243}$ | $2.5645_{\pm 0.4557}$ | $4.07_{\pm 0.02}$ |
| HierVAE (Jin et al., 2020a) | $100.0_{\pm 0.0}$ | $81.5_{\pm 1.1}$ | $97.7_{\pm 0.0}$ | $0.3021_{\pm 0.0063}$ | $0.4230_{\pm 0.0051}$ | $0.5771_{\pm 0.0121}$ | $1.4073_{\pm 0.0630}$ | $8.95_{\pm 0.06}$ |
| SAFE-GPT (Noutahi et al., 2024) | $96.5_{\pm 0.1}$ | $98.6_{\pm 1.4}$ | $73.5_{\pm 0.2}$ | $0.0024_{\pm 0.0004}$ | $0.0054_{\pm 0.0012}$ | $0.0414_{\pm 0.0022}$ | $0.1722_{\pm 0.0056}$ | $0.15_{\pm 0.00}$ |
| MolHF (Zhu et al., 2023) | $71.0_{\pm 0.3}$ | $59.6_{\pm 0.4}$ | $97.6_{\pm 0.05}$ | $0.8831_{\pm 0.0198}$ | $1.8072_{\pm 0.0285}$ | $9.1608_{\pm 0.6722}$ | $10.3760_{\pm 0.0875}$ | $31.26_{\pm 0.10}$ |
| DiGress (Vignac et al., 2023) | $85.4_{\pm 0.0}$ | $99.7_{\pm 0.0}$ | $99.9_{\pm 0.0}$ | $0.1957_{\pm 0.0028}$ | $0.0229_{\pm 0.0001}$ | $0.3370_{\pm 0.0042}$ | $1.0309_{\pm 0.0182}$ | $2.05_{\pm 0.01}$ |
| DeFoG (Qin et al., 2024) | $85.9_{\pm 0.15}$ | $98.4_{\pm 0.09}$ | $99.2_{\pm 0.04}$ | $0.1550_{\pm 0.0070}$ | $0.1252_{\pm 0.0088}$ | $0.4134_{\pm 0.0061}$ | $1.3597_{\pm 0.0575}$ | $4.46_{\pm 0.08}$ |
| FragFM (ours) | $98.0_{\pm 0.0}$ | $99.0_{\pm 0.0}$ | $95.4_{\pm 0.1}$ | $0.0374_{\pm 0.0001}$ | $0.0196_{\pm 0.0008}$ | $0.1482_{\pm 0.0026}$ | $0.3570_{\pm 0.0006}$ | $1.34_{\pm 0.01}$ |

## C.2 RETROSYNTHETIC ANALYSIS

To assess whether FragFM can generate structures that are not only chemically valid but also synthetically accessible, we performed a retrosynthetic analysis using AIZynthFinder (Genheden et al. (2020)), a widely used template-based retrosynthetic planning tool. We evaluated baseline models and FragFM in the MOSES benchmark by generating 25,000 molecules and recording (i) the fraction of molecules for which AiZynthFinder succeeded in identifying a synthetic route ('Solved') and (ii) the number of reaction steps required for successfully solved routes. As shown in fig. 10, FragFM achieves the lowest unsolved ratio among all learned generative models, and furthermore attains the highest proportion of solved routes requiring only a single step. While the MOSES dataset itself contains 80% of solved molecules under the same setup, FragFM's solved rate remains close (77%), indicating that the model successfully learns fragment compositions that map to synthetically feasible structures. These results suggest that BRICS-based fragment-level generation introduces chemically meaningful inductive biases that steer the model toward generating molecules that are more often experimentally accessible, even though no explicit synthetic constraints were used during training.

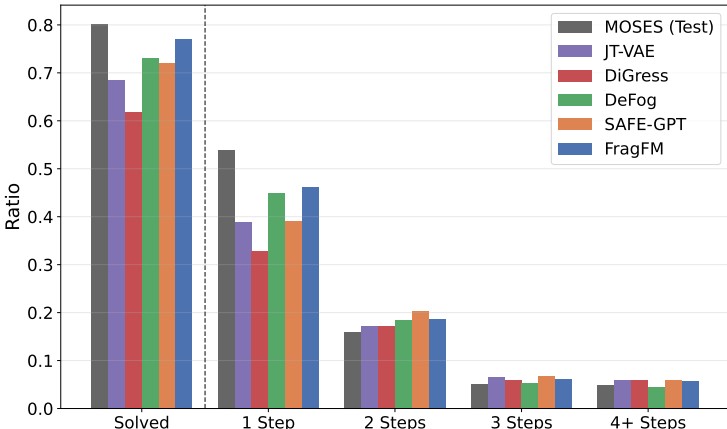

Figure 10: **AIzynthFinder synthesis-step distributions for the MOSES dataset and baseline models trained on MOSES.**

## C.3 COARSE-TO-FINE AUTOENCODER

Similar to the latent diffusion model (Rombach et al., 2022), we first train the coarse-to-fine autoencoder, which is then kept frozen during flow model training. While the main experimental results imply that the autoencoder works effectively in conjunction with the flow model, we further assess whether the autoencoder alone can accurately reconstruct the original molecule from its coarse graph and latent representation. We measure reconstruction accuracy at both the bond and whole-graph

levels. As reported in table 7, bond-level accuracy exceeds 99% on MOSES and GuacaMol, indicating nearly perfect recovery of individual chemical bonds. Graph-level accuracy is similarly high, confirming that overall connectivity patterns are faithfully preserved. Even on the structurally diverse and larger NPGen dataset, the autoencoder maintains strong performance with only a slight drop in accuracy, underscoring its robustness in handling complex molecular topologies.

We next examine the role of the latent representation $z$ in enabling accurate reconstruction. While the coarse fragment graph captures higher-level structural motifs, it alone is insufficient to recover atom-level connectivity. This limitation underscores the importance of our coarse-to-fine design, where the latent representation complements the fragment graph by encoding fine-grained connectivity details. To make this explicit, we conduct an ablation in which reconstruction is attempted with a random latent vector $z \sim \mathcal{N}(0, I)$ instead of the encoded $z$. As reported in table 8, reconstruction accuracy collapses in this case, particularly on the larger and more complex NPGen benchmark. In contrast, decoding with the encoded $z$ consistently yields near-perfect reconstruction, empirically demonstrating that $z$ encodes nearly complete atom-level connectivity and highlighting the novelty of integrating fragment-level and atom-level information through our coarse-to-fine framework.

Table 7: **Coarse-to-fine autoencoder accuracy**. "Bond" denotes the accuracy of individual atom-to-atom bonds, while "Graph" denotes the percentage of graphs in which all bonds are predicted correctly.

| Dataset | Train set reconstruction (%) | | Test set reconstruction (%) | |
|---|---|---|---|---|
| | Bond | Graph | Bond | Graph |
| MOSES | 99.99 | 99.96 | 99.99 | 99.93 |
| GuacaMol | 99.99 | 99.43 | 99.98 | 99.42 |
| ZINC250k | 100.0 | 99.96 | 99.64 | 98.71 |
| NPGen | 99.98 | 97.62 | 99.71 | 97.43 |

Table 8: **Coarse-to-fine autoencoder accuracy under ablation on $z$.** For each test dataset, reconstruction was attempted by decoding the fragment-level graph with a randomly sampled $z \sim \mathcal{N}(0, I)$.

| Graph-level accuracy on test set (%) | MOSES | GuacaMol | NPGen |
|---|---|---|---|
| Random $z$ | 55.2 | 46.7 | 34.4 |
| Encoded $z$ | 99.9 | 99.4 | 97.4 |

FragFM uses a pre-trained decoder from the coarse-to-fine autoencoder. Along the latent variable $z$, which carries connectivity information across multiple fragments, the decoder predicts the probability of junctions among fragments based on the coarse graph and $z$, recovering the intended molecule. To isolate the role of this pre-trained decoder, we conduct an ablation in which its parameters are randomly re-initialized during inference. As shown in table 9, basic molecular properties such as validity, uniqueness, and novelty remain mostly unchanged, while distributional and structural metrics substantially degrade. We attribute this degradation to the incorrect generation of molecules from the coarse graph, which pushes the distribution away from the original dataset. This underscores the practical importance of our decoder and hierarchical design.

Table 9: **Molecule generation on MOSES with ablation on coarse-to-fine autoencoder**. We use 25,000 generated molecules for evaluation. Results for FragFM are averaged over three independent runs.

| Model | Valid ↑ | Unique ↑ | Novel ↑ | Filters ↑ | FCD ↓ | SNN ↑ | Scaf ↑ |
|---|---|---|---|---|---|---|---|
| FragFM | 99.8 | 100.0 | 87.1 | 99.1 | 0.58 | 0.56 | 10.9 |
| FragFM (random decoder) | 99.8 | 99.9 | 87.1 | 98.5 | 0.9 | 0.52 | 9.8 |

## C.4 FRAGMENT BAG GENERALIZATION

Tables 10 and 11 report FragFM's performance when sampling with fragment bags derived from the test-set molecules on both MOSES and GuacaMol. Recall that MOSES uses a scaffold-split evaluation—test scaffolds are deliberately excluded from training—so sampling with only training-set fragments limits the model's ability to recover those unseen scaffolds, resulting in a depressed Scaf score. We re-ran the generation using fragment bags drawn from the test split to validate this. In MOSES, the training split contains 39,247 fragments, while the test split contains 9,789 fragments, of which 2,588 are unseen during training. In GuacaMol, the training split contains 192,751 fragments, while the test split contains 63,608 fragments, with 22,962 unseen. Using only test-set fragments for generation, FragFM's Scaf score increases dramatically, while all other metrics on MOSES and GuacaMol remain unchanged. This demonstrates that, leveraging its fragment embedding module, FragFM can generalize to novel fragments without compromising validity, uniqueness, or other quality measures.

Table 10: **Molecule generation with unseen fragment bag on MOSES dataset**. We use a total of 25,000 generated molecules for evaluation. The results are averaged over three independent runs.

| Model | Rep. Level | Valid ↑ | Unique ↑ | Novel ↑ | Filters ↑ | FCD ↓ | SNN ↑ | Scaf ↑ |
|---|---|---|---|---|---|---|---|---|
| Training set | - | 100.0 | 100.0 | - | 100.0 | 0.48 | 0.59 | 0.0 |
| GraphINVENT (Mercado et al., 2021) | Atom | 96.4 | 99.8 | - | 95.0 | 1.22 | 0.54 | 12.7 |
| JT-VAE (Jin et al., 2018) | Fragment | 100.0 | 100.0 | 99.9 | 97.8 | 1.00 | 0.53 | 10.0 |
| DiGress (Vignac et al., 2023) | Atom | 85.7 | 100.0 | 95.0 | 97.1 | 1.19 | 0.52 | 14.8 |
| DisCo (Xu et al., 2024) | Atom | 88.3 | 100.0 | 97.7 | 95.6 | 1.44 | 0.50 | 15.1 |
| Cometh (Siraudin et al., 2024) | Atom | 90.5 | 99.9 | 92.6 | 99.1 | 1.27 | 0.54 | 16.0 |
| DeFoG (Qin et al., 2024) | Atom | 92.8 | 99.9 | 92.1 | 98.9 | 1.95 | 0.55 | 14.4 |
| FragFM (train fragments) | Fragment | 99.8 | 100.0 | 87.1 | 99.1 | 0.58 | 0.56 | 10.9 |
| FragFM (test fragments) | Fragment | 99.8 | 100.0 | 88.2 | 98.9 | 0.44 | 0.57 | 24.5 |

Table 11: **Molecule generation with unseen fragment bag on GuacaMol dataset**. We use a total of 10,000 generated molecules for evaluation. The results are averaged over three independent runs.

| Model | Rep. Level | Val. ↑ | V.U. ↑ | V.U.N. ↑ | KL Div. ↑ | FCD ↑ |
|---|---|---|---|---|---|---|
| Training set | - | 100.0 | 100.0 | - | 99.9 | 92.8 |
| MCTS (Jensen, 2019) | Atom | 100.0 | 100.0 | 95.4 | 82.2 | 1.5 |
| NAGVAE (Kwon et al., 2020) | Atom | 92.9 | 88.7 | 88.7 | 38.4 | 0.9 |
| DiGress (Vignac et al., 2023) | Atom | 85.2 | 85.2 | 85.1 | 92.9 | 68.0 |
| DisCo (Xu et al., 2024) | Atom | 86.6 | 86.6 | 86.5 | 92.6 | 59.7 |
| Cometh (Siraudin et al., 2024) | Atom | 98.9 | 98.9 | 97.6 | 96.7 | 72.7 |
| DeFoG (Qin et al., 2024) | Atom | 99.0 | 99.0 | 97.9 | 97.7 | 73.8 |
| FragFM (train fragments) | Fragment | 99.7 | 99.4 | 95.0 | 97.4 | 85.7 |
| FragFM (test fragments) | Fragment | 99.8 | 99.4 | 97.4 | 97.6 | 85.7 |

## C.5 LONG TAIL FRAGMENT RECOVERY OF FRAGFM

Molecular fragment distributions in real-world datasets are highly imbalanced, with a small number of frequent fragments and a long tail of rare ones. When we generate molecules at a fragment level, it is important to consider whether generative models can effectively recover such rare fragments during sampling, rather than being biased to the frequent head of the distribution. This challenge closely aligns with the long-tail problem in language models, where models tend to underpredict or ignore infrequent categories despite their importance (Zhao et al., 2021; Kang & Choi, 2023).

To this end, we group fragments in the training set according to their occurrence count $k$, and refer to them as $k$-rare fragments. We then compare the occurrence ratios across groups and quantify how often these fragments reappear in generated molecules, thereby assessing the model's ability to recover the long tail.

Figure 11 shows the results. Across all datasets, the recovery ratios of $k$-rare fragments in generated molecules closely follow the distributions observed in the training sets. Even for fragments that occur fewer than five times, FragFM is able to regenerate them at comparable frequencies. Unlike language models, which often underestimate rare tokens, we suppose that our fragment-to-vector module helps generalize across fragments, contributing to this effect. These findings suggest that the fragment-based representation, together with our modeling strategy, provides effective coverage of the long-tail space.

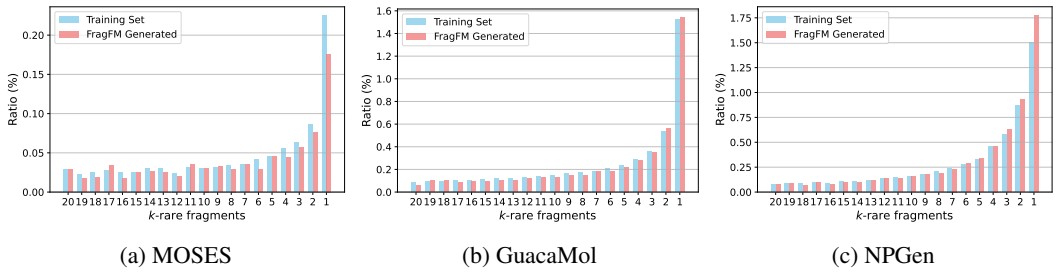

(a) MOSES         (b) GuacaMol         (c) NPGen

Figure 11: **Long-tail fragment recovery results.** Occurrence ratios of $k$-rare fragments in (a) MOSES, (b) GuacaMol, and (c) NPGen. Generated sets consist of 25,000 molecules for MOSES, 10,000 for GuacaMol, and 30,000 for NPGen.

## C.6   EFFECT OF FRAGMENT BAG SIZE

The size of the fragment bag $\mathcal{B}$ in eqs. (1) to (3) controls the number of fragments sampled at each step during training and inference. The posterior in eq. (1) corresponds to a self-normalized importance sampling (SNIS) estimator of the true posterior, whose bias and variance vanish as $N$ increases.

During training (eqs. (1) and (2)), the bag size $N_{\text{train}}$ determines how many negatives are included in the Info-NCE loss. Larger $N_{\text{train}}$ generally improves stability, although the optimality of the density-ratio estimator $f_\theta$ itself does not directly depend on $N_{\text{train}}$. By contrast, at inference time the bag size $N_{\text{inference}}$ directly controls the variance of the estimator: larger $N_{\text{inference}}$ yields better results, as it more closely approximates the full fragment library $\mathcal{F}$. In the default setting of FragFM, we set both $N_{\text{train}}$ and $N_{\text{inference}}$ to 384. To analyze the effect of fragment bag size, we ablate $N_{\text{train}}$ and $N_{\text{inference}}$ separately to quantify the trade-off between computational efficiency and fidelity.

As shown in fig. 12(a), increasing the inference-time bag size $N_{\text{inference}}$ consistently improves validity, filter scores, and FCD by reducing estimator variance. In contrast, varying the training-time bag size $N_{\text{train}}$ produces only minor differences once it is moderately large, with a slight upward trend observed in filter scores, as shown in fig. 12(b). We speculate that this stability may partly stem from training regularization techniques such as EMA, which help smooth optimization dynamics.

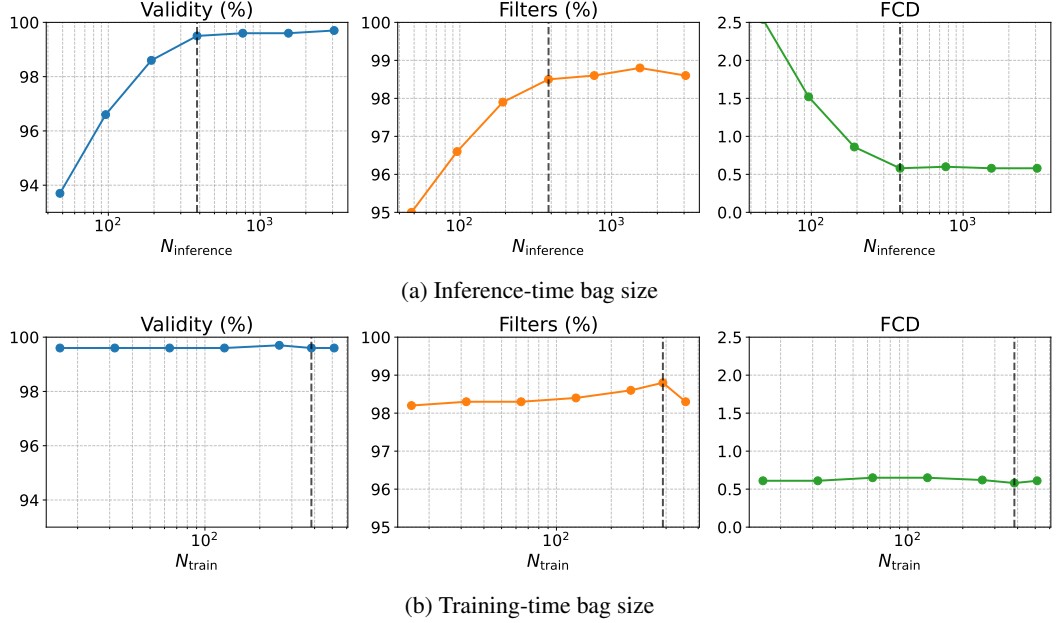

(a) Inference-time bag size

(b) Training-time bag size

Figure 12: **Effect of fragment bag size.** Comparison across (a) inference-time bag size (with $N_{\text{train}} = 384$), (b) training-time bag size (with $N_{\text{inference}} = 384$). To isolate the effect of bag size, the detailed-balance term was excluded during sampling. The horizontal axis (fragment bag size) is log-scaled in all subplots, and the default FragFM configuration is indicated by a black dashed line.

## C.7 STEERING FRAGFM TOWARDS NOVEL MOLECULES

While FragFM achieves state-of-the-art fidelity on most distributional metrics, its novelty on small-molecule benchmarks (tables 1 and 4) is slightly lower than some baselines. We note that this difference is modest and reflects the trade-off between novelty and fidelity in molecular generation (Mahmood et al., 2021; Geng et al., 2023). In fact, higher novelty scores can sometimes arise from atom-based models that generate valid (in terms of atomic valency) yet chemically implausible motifs that cannot stably exist in the real world, which does not necessarily correspond to meaningful chemical space exploration. Nevertheless, users may sometimes prefer sampling molecules from broader and previously underexplored regions of chemical space. To this end, we found that it is possible to modulate the sampling algorithm to generate more unseen molecules, fully utilizing the flexibility of the stochastic bag strategy.

Specifically, we consider two types of temperature scaling: (1) applying a temperature factor $\mathcal{T}_{\text{pred}}$ to reweight fragment logits within the in-bag transition kernel (eq. (3)), and (2) applying a temperature factor $\mathcal{T}_{\text{bag}}$ when reweighting fragments during bag construction at each Euler step (eq. (31)).

Empirically, as shown in table 12, increasing either $\mathcal{T}_{\text{pred}}$ or $\mathcal{T}_{\text{bag}}$ improves novelty with only a slight trade-off in fidelity metrics such as validity and FCD. Notably, the combined setting ($\mathcal{T}_{\text{pred}} = 1.5, \mathcal{T}_{\text{bag}} = 1.5$) achieves over $94\%$ novelty while still maintaining high validity ($99.2\%$) and filters ($98.3\%$) metrics, indicating that the generated molecules remain well within the MOSES filter constraints. This demonstrates that temperature scaling provides a simple and effective mechanism to balance fidelity and exploration, offering users a practical control knob for tuning generation behavior.

Table 12: **Effect of temperature scaling on MOSES.** Benchmark results under different settings of prediction temperature $\mathcal{T}_{\text{pred}}$ and fragment-bag temperature $\mathcal{T}_{\text{bag}}$.

| $\mathcal{T}_{\text{pred}}$ | $\mathcal{T}_{\text{bag}}$ | Valid ↑ | Unique ↑ | Novel ↑ | Filters ↑ | FCD ↓ | SNN ↑ | Scaf ↑ |
|---|---|---|---|---|---|---|---|---|
| 1.0 | 1.0 | 99.8 | 100.0 | 87.1 | 99.1 | 0.58 | 0.56 | 10.9 |
| 1.0 | 1.5 | 99.2 | 100.0 | 94.2 | 98.3 | 0.90 | 0.52 | 13.5 |
| 1.5 | 1.0 | 99.7 | 100.0 | 88.6 | 98.8 | 0.88 | 0.54 | 11.0 |
| 1.5 | 1.5 | 99.2 | 100.0 | 94.5 | 98.3 | 0.91 | 0.51 | 13.1 |

## C.8 MORE RESULTS ON CONDITIONAL GENERATION

For simple molecular properties (logP, QED, and number of rings), we perform conditional generation on the MOSES dataset using regressors trained on its training split. The detailed illustration of property distributions and corresponding targets is depicted in fig. 13. For protein-target conditioning, we perform conditioning on the ZINC250k dataset. The targets were selected from the DUD-E$^+$ virtual screening benchmark for our docking score experiments, following Yang et al. (2021). The established reliability of Smina (Koes et al., 2013), which is a forked version of AutoDock Vina (Trott & Olson, 2010), evidenced by its high AUROC for discriminating hits from decoys on DUD-E$^+$, led us to use it as an oracle.

Figure 17 depicts the Smina docking score distributions for ZINC250K molecules against three targets (fa7, jak2, parp1). Since lower scores correspond to stronger predicted binding, we selected the conditioning value for each protein at the extreme left tail of its distribution (FA7: –10.0 kcal/mol, 0.01%; JAK2: –11.0 kcal/mol, 0.08%; PARP1: –12.0 kcal/mol, 0.09%), indicated by the vertical dashed lines in fig. 17 to focus generation on the most tightly binding candidates.

With the perspective of chemistry, the high FCD and low validity with conditions of DiGress shown in figs. 14 and 15 highlights a critical challenge for atom-based approaches: satisfying targeted property constraints while ensuring chemical correctness, *simultaneously*. From a chemical perspective, this distinction can be attributed to the nature of the search space; atom-based approaches explore a vastly larger space, far exceeding the molecular space, where many cases can lead to chemically invalid structures, especially when generation is heavily biased by property objectives. Conversely, FragFM's fragment-based construction inherently operates within a more chemically sound and constrained subspace by assembling pre-validated chemical motifs. These findings collectively emphasize the intrinsic advantages of employing fragments as semantically rich and structurally robust building blocks, particularly for reliable, property-focused molecular generation.

Moreover, the importance of the fragment bag's composition, which is shown in the main text (fig. 5), is intuitive: it defines the accessible chemical space and, consequently, the possible range of achievable molecular properties (*e.g.*, generating acyclic molecules is impossible if the fragment bag exclusively contains ring-based structures, among other structural constraints). Based on $\lambda_{\mathcal{B}}$, FragFM automatically modulates fragment selection probabilities, inducing a drift in the fragment space to generate the chemically valid molecules satisfying the given objective. It enables the model to construct molecules with desired properties even if the initial general-purpose fragment bag is not perfectly tailored to a specific task, making our strategy a powerful and practically manageable tool for fine-grained control.

We also analyze how conditioning strength affects the diversity (see fig. 20). Specifically, we observe a consistent trend that stronger conditioning leads to lower diversity: when we increase $\lambda_X$ alone, the diversity score decreases monotonically; a similar monotonic decrease is observed when we increase $\lambda_{\mathcal{B}}$ alone; and in the joint setting, where both $\lambda_X$ and $\lambda_{\mathcal{B}}$ are varied, diversity again exhibits an overall decreasing trend as the combined guidance strength grows. Notably, the absolute reduction in diversity remains modest (from approximately 0.880 in the unconditional setting to approximately 0.855 under the strongest guidance), indicating that FragFM continues to produce structurally rich and varied molecules even under strong conditioning. Overall, fig. 20 illustrates this trade-off between conditioning strength and diversity and shows that conditional generation in FragFM introduces only a small diversity cost in exchange for improved alignment with fragment-bag and property constraints.

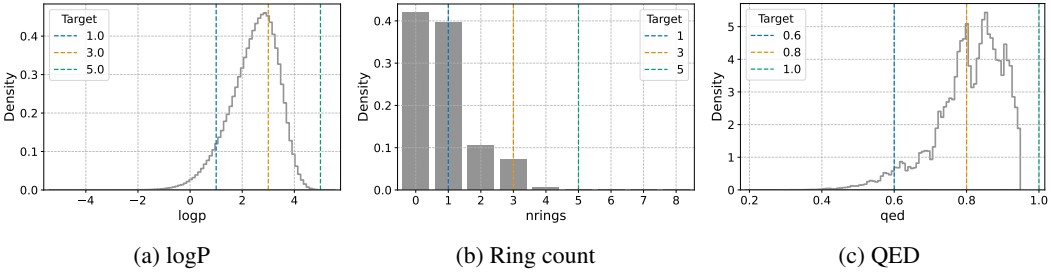

(a) logP         (b) Ring count         (c) QED

Figure 13: **Distribution of molecular properties (logP, ring count, and QED) for the MOSES dataset**. Colored vertical lines denote the conditioning scores applied for each target property value.

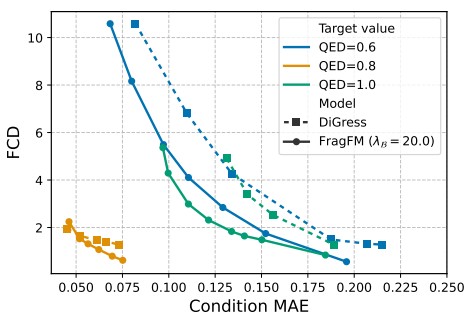

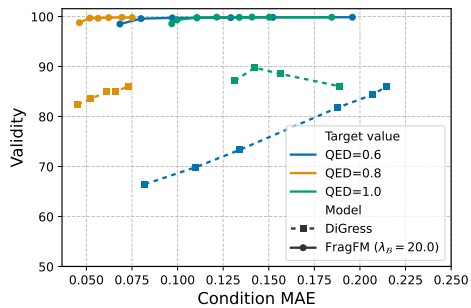

(a) MAE-FCD curves for QED conditioning on the MOSES dataset.

(b) MAE-validity curves for QED conditioning on the MOSES dataset.

Figure 14: **Conditioning results on QED**. MAE-FCD and MAE-validity curves for FragFM and DiGress under QED conditioning on the MOSES dataset. Different conditioning values are color-coded.

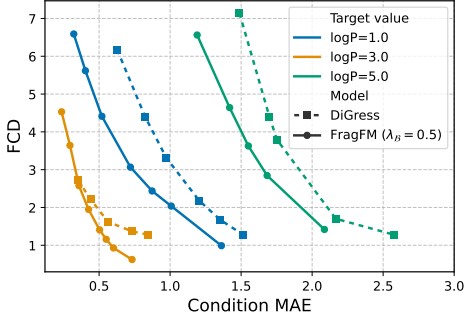

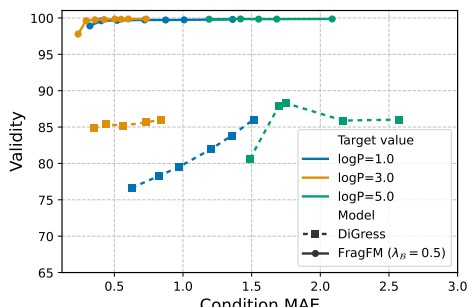

(a) MAE-FCD curves for logP conditioning on the MOSES dataset.

(b) MAE-validity curves for logP conditioning on the MOSES dataset.

Figure 15: **Conditioning results on logP**. MAE-FCD and MAE-validity curves for FragFM and DiGress under logP conditioning on the MOSES dataset. Different conditioning values are color-coded.

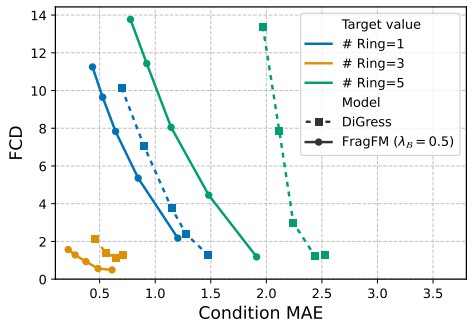 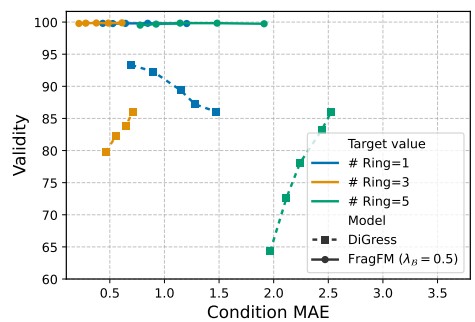

(a) MAE-FCD curves for the number of rings conditioning on the MOSES dataset.

(b) MAE-validity curves for the number of rings conditioning on the MOSES dataset.

Figure 16: **Conditioning results on number of rings**. MAE-FCD and MAE-validity curves for FragFM and DiGress under a number of rings conditioning on the MOSES dataset. Different conditioning values are color-coded.

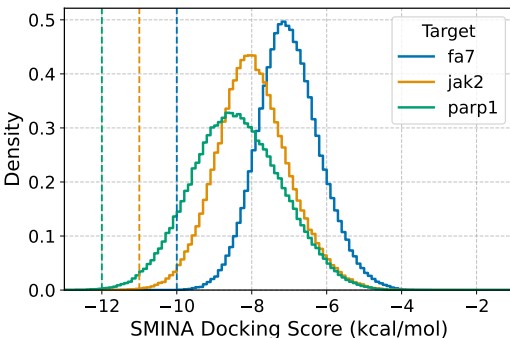

Figure 17: **Distribution of SMINA docking scores for the ZINC250k dataset across different target proteins**. Vertical lines denote the conditioning scores applied for each target protein.

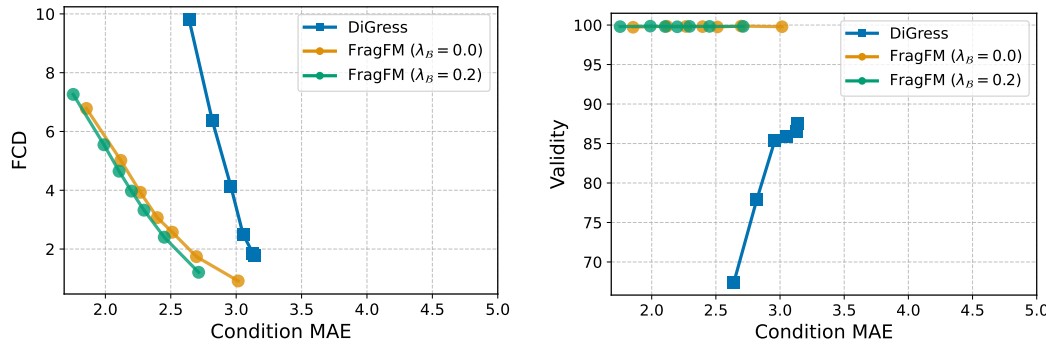

(a) MAE-FCD curves for FA7 docking score conditioning.

(b) MAE-Validity curves for FA7 docking score conditioning.

Figure 18: **Conditioning results on FA7 (target: -10.0).** Each point represents 10,000 generated molecules. Results are shown for DiGress and FragFM with $\lambda_{\mathcal{B}} = 0.0$ and $0.2$.

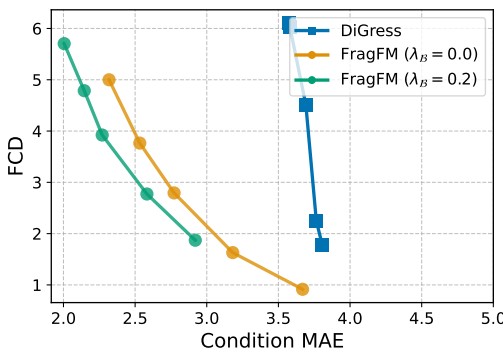

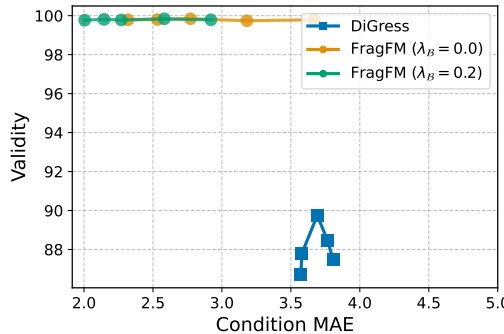

(a) MAE-FCD curves for PARP1 docking score conditioning.

(b) MAE-Validity curves for PARP1 docking score conditioning.

Figure 19: **Conditioning results on PARP1 (target: -12.0).** Each point represents 10,000 generated molecules. Results are shown for DiGress and FragFM with $\lambda_{\mathcal{B}} = 0.0$ and 0.2.

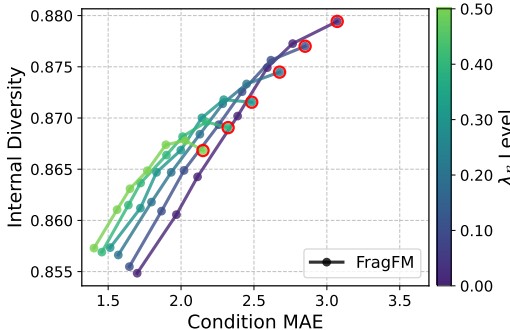

Figure 20: **Effect of guidance strength on diversity for JAK2-conditioned generation.**

## C.9 SAMPLING EFFICIENCY: SAMPLING STEPS AND TIME

Diffusion- and flow-based models typically require multiple denoising iterations, which slows sampling. Table 13 shows the performance of MOSES benchmark metrics of FragFM and baseline denoising based models with different denoising steps. For small sampling steps, FragFM outperforms the baseline models with minimal metric degradation, particularly at low step counts, by a wide margin. With only 10 sampling steps, FragFM achieves higher validity and lower FCD than competing models that run 500 steps.

We also compare sampling time across different models in table 14. By operating both node- and edge-probability predictions, edge computations scale quadratically with graph size, making them the fastest approach among the compared models. Coupled with its robust performance at far fewer steps, FragFM could be further optimized with substantial speedups over atom-level methods with high generative quality.

Table 13: **Performance of denoising-based graph generative models on the MOSES dataset across different sampling step counts**. All the models are one-shot models. Results for DeFoG and Cometh are taken from their original publications; DiGress (excluding the 500-step setting) were obtained by retraining the model with the differing sampling steps from the official implementation. The best performance is shown in **bold** for each sampling step.

| Sampling steps | Model | Rep. Level | Val. ↑ | V.U. ↑ | V.U.N. ↑ | Filters ↑ | FCD ↓ | SNN ↑ | Scaf ↑ |
|---|---|---|---|---|---|---|---|---|---|
| - | Training set | - | 100.0 | 100.0 | - | 100.0 | 0.48 | 0.59 | 0.0 |
| 10 | DiGress | Atom | 6.3 | 6.3 | 6.3 | 66.4 | 9.40 | 0.38 | 7.4 |
| | Cometh | Atom | 26.1 | 26.1 | 26.0 | 59.9 | 7.88 | 0.36 | 8.9 |
| | DeFoG | Atom | - | - | - | - | - | - | - |
| | FragFM | Fragment | **96.8** | **96.6** | **89.4** | **96.8** | **0.92** | **0.52** | **15.3** |
| 50 | DiGress | Atom | 75.3 | 75.3 | 72.3 | 94.0 | 1.35 | 0.51 | 16.1 |
| | Cometh | Atom | 82.9 | 82.9 | 80.5 | 94.6 | 1.54 | 0.49 | **18.4** |
| | DeFoG | Atom | 83.9 | 83.8 | 81.2 | 96.5 | 1.87 | **0.59** | 14.4 |
| | FragFM | Fragment | **99.5** | **99.5** | **89.1** | **98.5** | **0.65** | 0.54 | 11.2 |
| 100 | DiGress | Atom | 82.6 | 82.6 | 79.2 | 95.2 | 1.14 | 0.51 | 15.4 |
| | Cometh | Atom | 85.8 | 85.7 | 82.9 | 96.5 | 1.43 | 0.50 | **17.2** |
| | DeFoG | Atom | - | - | - | - | - | - | - |
| | FragFM | Fragment | **99.7** | **99.7** | **88.5** | **98.8** | **0.62** | **0.55** | 11.6 |
| 300 | DiGress | Atom | 85.3 | 85.3 | 81.1 | 96.5 | 1.11 | 0.52 | 13.5 |
| | Cometh | Atom | 86.9 | 86.9 | 83.8 | 97.1 | 1.44 | 0.51 | **17.8** |
| | DeFoG | Atom | - | - | - | - | - | - | - |
| | FragFM | Fragment | **99.8** | **99.8** | **87.2** | **98.9** | **0.58** | **0.55** | 11.6 |
| 500 | DiGress | Atom | 85.7 | 85.7 | 81.4 | 97.1 | 1.19 | 0.52 | 14.8 |
| | Cometh | Atom | 87.0 | 86.9 | 83.8 | 97.2 | 1.44 | 0.51 | **15.9** |
| | DeFoG | Atom | 92.8 | 92.7 | 85.4 | 98.9 | 1.95 | 0.55 | 14.4 |
| | FragFM | Fragment | **99.8** | **99.8** | **86.9** | **99.1** | **0.58** | **0.56** | 10.9 |
| 700 | DiGress | Atom | 85.5 | 85.5 | 82.6 | 95.0 | 1.33 | 0.50 | 15.3 |
| | Cometh | Atom | 87.2 | 87.1 | 83.9 | 97.2 | 1.43 | 0.51 | **15.9** |
| | DeFoG | Atom | - | - | - | - | - | - | - |
| | FragFM | Fragment | **99.9** | **99.9** | **86.9** | **99.1** | **0.61** | **0.56** | 10.8 |
| 1000 | DiGress | Atom | 84.7 | 84.7 | 81.3 | 96.1 | 1.31 | 0.51 | 14.5 |
| | Cometh | Atom | 87.2 | 87.2 | 84.0 | 97.2 | 1.44 | 0.51 | **17.3** |
| | DeFoG | Atom | - | - | - | - | - | - | - |
| | FragFM | Fragment | **99.8** | **99.8** | **86.6** | **99.1** | **0.62** | **0.56** | 12.9 |

Table 14: **Comparison of sampling time across different datasets and methods**. Experiments were conducted on a single NVIDIA GeForce RTX 3090 GPU and an Intel Xeon Gold 6234 CPU @ 3.30GHz. *Results for DeFoG are taken from the original paper, where experiments were conducted on an NVIDIA A100 GPU. Note that DeFoG shares most of its backbone architecture with DiGress, with only minor adjustments.

| | | Sampling steps | MOSES | GuacaMol | NPGen |
|---|---|---|---|---|---|
| Property | Min. nodes | - | 8 | 2 | 2 |
| | Max. nodes | - | 27 | 88 | 99 |
| | # Samples | - | 25000 | 10000 | 30000 |
| Sampling Time (hour) | DiGress | 500 | 3.0 | - | 36.0 |
| | DeFoG* | 500 | 5.0 | 7.0 | - |
| | FragFM | 500 | 0.9 | 1.3 | 7.0 |
| | FragFM | 50 | 0.2 | 0.2 | 0.9 |

## D  Experimental Setup and Details

### D.1  Datasets

We provide details of the datasets used in our experiments, including MOSES (Polykovskiy et al., 2020), GuacaMol (Brown et al., 2019), and ZINC250k (Irwin & Shoichet, 2005).

**MOSES.** The MOSES benchmark is constructed from a subset of the ZINC Clean Leads dataset, containing approximately 1.9 million drug-like molecules. The molecules are curated and filtered for training and evaluation of molecular generative models, with a predefined data split of training, test, and scaffold split test sets. All molecules in the MOSES dataset meet drug-likeness criteria, including molecular weight and logP ranges.

**GuacaMol.** GuacaMol is based on the ChEMBL database and provides a large-scale benchmark for both distribution-learning and goal-directed molecular generation tasks. The training dataset comprises around 1.6 million molecules extracted from ChEMBL v24.

**ZINC250k.** The ZINC250k dataset comprises 249,456 molecules selected from the larger ZINC database.

### D.2  Fragmentation

We use the BRICS (Degen et al., 2008) decomposition scheme to construct our fragment library, with all inter-fragment connections restricted to single bonds. The fragment library is built by fragmenting every molecule in each dataset and collecting the resulting unique fragments into a fragment bag. The final fragment counts in the training, validation, and test splits are: MOSES (39,247 / 12,212 / 9,789), GuacaMol (192,751 / 30,418 / 63,608), NPGen (117,998 / 18,063 / 40,214), and ZINC250k (28,235 / 4,231 / 2,683). In the test sets, the number of fragments unseen during training is 2,588 for MOSES, 22,962 for GuacaMol, 12,116 for NPGen, and 367 for ZINC250k.

### D.3  Metrics

We provide details of common metrics in both MOSES (Polykovskiy et al., 2020) and GuacaMol (Brown et al., 2019) benchmarks.

**Common Metrics.** These metrics are fundamental for assessing the basic performance of molecular generative models. Note that **V.U.** and **V.U.N.** metrics are multiplied values of each metric, *i.e.*, V.U.N. is computed by multiplying validity, uniqueness and novelty.

- **Validity (Valid):** This metric measures the proportion of generated molecules that are chemically valid according to a set of rules, typically checked using tools like RDKit. A SMILES string is considered valid if it can be successfully parsed and represents a chemically sensible molecule (*e.g.*, correct atom valencies, no impossible structures).

- **Uniqueness (Unique):** This indicates the percentage of unique molecules among valid generated molecules. A high uniqueness score suggests the model is generating diverse structures rather than repeatedly producing the same few molecules.

- **Novelty (Novel):** This metric quantifies the fraction of unique and valid generated molecules that are not present in the training dataset. It assesses the model's ability to generate novel chemical molecules.

**MOSES Metrics.** The MOSES benchmark focuses on distribution learning. Key metrics beyond the foundational ones include:

- **Filters:** This refers to the percentage of valid, unique, and novel molecules that pass a set of medicinal chemistry filters (PAINS, MCF) and custom rules defined by the MOSES benchmark (*e.g.*, specific ring sizes, element types), which are used in curating a dataset of MOSES. This evaluates the drug-likeness or suitability of generated molecules according to predefined structural criteria.

- **Fréchet ChemNet Distance (FCD):** FCD (Preuer et al., 2018) measures the similarity between the distribution of generated molecules and a reference (**test**) dataset based on the

latent representation of molecules using a pre-trained neural network (ChemNet). A lower FCD indicates that the generated distribution is closer to the reference distribution.

- **Similarity to Nearest Neighbor (SNN):** This metric calculates the average Tanimoto similarity using Morgan fingerprints (Rogers & Hahn, 2010) between each generated molecule and its nearest neighbor in the reference **(test)** dataset. A higher SNN suggests that the generated molecules are similar in structure to known molecules in the target chemical space.

- **Scaffold Similarity (Scaf):** This metric specifically assesses the diversity of molecular scaffolds. It calculates a cosine similarity between the vectors of the occurrence of Bemis–Murcko scaffolds (Bemis & Murcko, 1996) of the molecules in the reference **(test)** dataset and the generated ones. A higher score suggests that the generated scaffolds are similar to reference scaffolds.

**GuacaMol Metrics.** GuacaMol provides benchmarks for distribution-learning and goal-directed generation. For its distribution-learning benchmark, which is utilized for our main results, the primary aggregated metrics are:

- **Kullback-Leibler Divergence (KL Div.) Score:** This metric computes the KL divergence between the distributions of several physicochemical and topological properties of the generated molecules and the training set. These individual KL divergences ($D_{KL}$) are then combined into a single score by averaging the negative exponentials of them (*i.e.*, $\exp(-D_{KL})$) to reflect how well the model reproduces the overall property distributions. Due to the nature of the calculation method, a score closer to 1 indicates better similarity.

- **Fréchet ChemNet Distance (FCD) Score:** Similar to the MOSES FCD, GuacaMol also uses an FCD metric to compare the distributions of generated molecules and the **training** set. The only difference is that in GuacaMol, the raw FCD value (where lower is better) is transformed into a score where higher is better.

**ZINC250k Metrics.** ZINC250k is a collection of 250k molecules from ZINC (Irwin & Shoichet, 2005). Multiple generative models evaluate their performance with the following metrics:

- **NSPDK:** The Neighborhood Subgraph Pairwise Distance Kernel (NSPDK) is a graph kernel that measures structural similarity between molecular graphs by considering neighborhoods of atoms and their pairwise distances. For generative evaluation, the NSPDK distance is computed between the generated set and the reference dataset, capturing differences in both local and global graph structures. A lower NSPDK value indicates that the generated molecules are structurally closer to the reference distribution. We used the official implementation from Jo et al. (2022) to compute the NSPDK.

- **Fréchet ChemNet Distance (FCD):** The FCD used in ZINC250k follows the same definition as in the MOSES benchmark.

## D.4 BASELINES

Next, we briefly introduce baseline strategies that we compared FragFM in the main results. We focused on molecular *graph* generative models, which are categorized by autoregressive and one-shot generation models. Each model uses either an atom- or a fragment-level representation.

**GraphInvent** (Mercado et al., 2021) employs a graph neural network (GNN) approach for *de novo* molecular design. It first computes the trajectory of graph decomposition based on atom-level representation, and then trains a GNN to learn the action of atom and bond addition on a given subgraph of a molecule. During the inference stage, it builds molecules in an atom-wise manner.

**JT-VAE** (Jin et al., 2018) or Junction Tree Variational Autoencoder, generates molecular graphs in a two-step process. It first decodes a latent vector into a tree-structured scaffold representing molecular components (like rings and motifs) and then assembles these components into a complete molecular graph, ensuring chemical validity. Since it iteratively decides whether to add a node during the sampling process, we consider it as an autoregressive model despite its use of VAE.

**SAFE-GPT** (Noutahi et al., 2024) is an autoregressive sequence generative model that produces SAFE (Sequential Attachment-based Fragment Embedding) strings—a novel representation proposed

in the paper, where molecules are expressed as sequences of fragments. The model is built on a GPT-2–like transformer architecture. While the original work reported only basic results on validity, uniqueness, and diversity, we retrained the SAFE-GPT-20M model using its official implementation to obtain comprehensive benchmark results.

**MCTS** (Jensen, 2019) is a non-deep-learning-based strategy that uses Monte Carlo Tree Search for molecular graph generation. Using atom insertion or addition as an action, it sequentially builds molecules from a starting molecule.

**MolHF** (Zhu et al., 2023) is a hierarchical normalizing flow model designed for molecular graph generation. It employs a coarse-to-fine strategy, generating a fragment connectivity graph and subsequently refining it at the atom level via conditional flows. As MolHF refers to itself as a one-shot flow-based model, we treat it as a one-shot model.

**MolGrow** (Kuznetsov & Polykovskiy, 2021) is a hierarchical normalizing flow model that constructs molecular graphs via a recursive node-splitting process. Starting from a single node, it iteratively divides nodes to generate complex structures, enabling multi-level control over molecular properties through its hierarchical latent space. Although MolGrow is a hierarchical normalizing flow model, we follow (Zhu et al., 2023) and categorize it as a one-shot model for consistency with prior taxonomy.

**NAGVAE** (Kwon et al., 2020), non-autoregressive Graph Variational Autoencoder, is a VAE-based one-shot graph generation model utilizing compressed graph representation. It reconstructs the molecular graphs from latent vectors, aiming for scalability and capturing global graph structures.

**DiGress** (Vignac et al., 2023) is the first discrete diffusion model designed for graph generation. It operates by iteratively removing noise from both graph edges and node types, learning a reverse diffusion process to construct whole graphs from a noise distribution.

**DisCo** (Xu et al., 2024) is a graph generation model that defines a forward diffusion process with a continuous-time Markov chain (CTMC). The model learns a reverse generative process to denoise both the graph structure and its attributes simultaneously.

**Cometh** (Siraudin et al., 2024) is a continuous-time discrete-state graph diffusion model. Similar to DisCo, it formulates graph generation as reversing a CTMC defined on graphs, in which the model learns the chain's transition rates to generate new graph structures.

**DeFoG** (Qin et al., 2024) is a generative framework that applies the principles of flow matching directly to discrete graph structures. After training with a flow-matching strategy, it uses a CTMC for denoising to generate graphs.

# E PARAMETERIZATION AND HYPERPARAMETERS

## E.1 COARSE-TO-FINE AUTOENCODER

Our coarse-to-fine autoencoder (eq. (4)) compresses the atom-level graph into a single latent vector $z$ and then uses it, together with the fragment-level graph $\mathcal{G}$, to reconstruct all atom–atom connections. The encoder, an MPNN (Gilmer et al., 2017), takes $G$ and pools its node features into $z$. The decoder conditions on $\mathcal{G}$ and $z$ to predict a distribution over every possible atom–atom edge between them for each pair of linked fragments. Internally, it propagates messages along original intra-fragment bonds and across all candidate inter-fragment edges, enabling the recovery of the complete atomistic structure from the coarse abstraction.

## E.2 FRAGMENT EMBEDDER, PREDICTION MODEL, AND PROPERTY DISCRIMINATOR

To parameterize the neural network $f_\theta(X_t, x)$ in eq. (2), we jointly train two components: a fragment embedder and a Graph Transformer (GT). Figure 21 provides an overview.

The fragment embedder, built on an MPNN backbone Gilmer et al. (2017), maps each fragment to a fixed-dimensional embedding vector. Given a fragment-level graph $X_t$, we apply this shared embedder to every fragment node, producing a set of local structure embeddings. Multiple GT layers, then process these embeddings to capture inter-fragment interactions and global context. The GT layers were designed with the sample architecture and hyperparameters as prior atom-based

diffusion- and flow-based molecule generative models (Vignac et al., 2023; Qin et al., 2024; Siraudin et al., 2024; Xu et al., 2024). We directly predict the discrete fragment-graph edges $\mathcal{E}_1$ and the continuous latent vector $z_1$ from the final GT output embeddings. To predict fragment types, we compute the inner product between each candidate fragment type embedding and its corresponding GT embedding to infer the scores of different fragments. We reuse the flow model's architecture for property discrimination: We aggregate both the fragment-level embeddings and the GT's global readout to produce fragment—and molecule-level property predictions.

We train our flow model with AdamW optimizer, using $(\beta_1, \beta_2) = (0.9, 0.999)$, a learning rate of $5 \times 10^{-4}$, and gradient-norm clipping at $4.0$. We employ the exponential moving average (EMA) scheme of Karras et al. (2024) to stabilize training. Training is performed on a single NVIDIA A100 GPU for 96 h on MOSES and GuacaMol and 144 h on NPGen, and we select the checkpoint with the lowest validation loss.

### E.3 AUXILIARY FEATURES

Although graph neural networks exhibit inherent expressivity limitations (Xu et al., 2018), augmenting them with auxiliary features has proven effective at mitigating these shortcomings. For example, Vignac et al. (2023) augments each noisy graph with cycle counts, spectral descriptors, and basic molecular properties (e.g., molecular weight, atom valence). More recently, relative random walk probabilities (RRWP) have emerged as a highly expressive yet efficient encoding (Ma et al., 2023): by stacking the first $K$ powers of the normalized adjacency matrix $M = D^{-1}A$, RRWP constructs a $k$-step transition probabilities that capture rich topological information. Accordingly, we integrate RDKit-derived molecular descriptors into the fragment embedder, and RRWP features into the graph transformer, enriching the model's ability to capture complex molecular semantics.

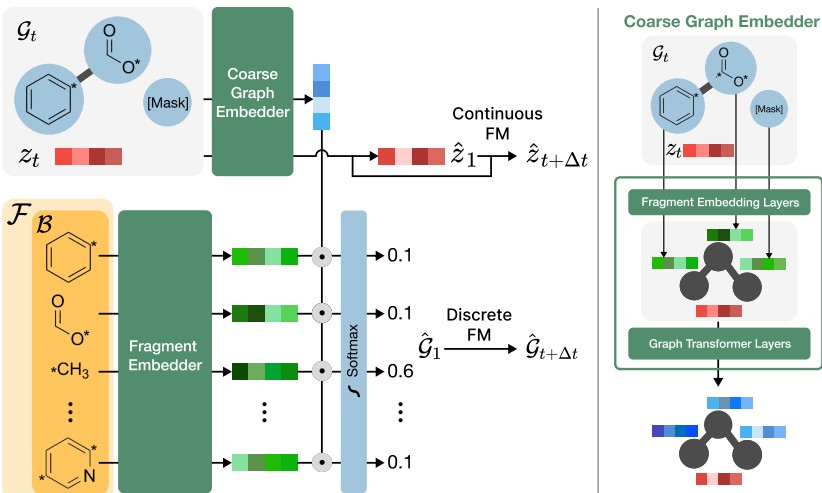

Figure 21: **Overview of the parameterization of the fragment embedder and prediction model, *i.e.*, $f_\theta$ in fig. 1**. (left) Each fragment in the fragment bag $\mathcal{B}$ is embedded by the fragment embedder, while each node in the coarse graph is mapped to a fixed-size vector. We compute $f_\theta(X_t, x)$ by taking the inner product of the two embeddings. (right) The coarse-graph embedder first maps every node to an embedding, producing a coarse graph whose nodes are single vectors; the resulting graph is then passed through the graph-transformer layer.

### E.4 NOISE SCHEDULE

Selecting an appropriate noise schedule is crucial for the performance of diffusion- and flow-based models (Vignac et al., 2023; Siraudin et al., 2024; Qin et al., 2024). Following Qin et al. (2024), we adopt the polynomially decreasing (polydec) time distortion, which skews the initially uniform time distribution so that more steps are allocated near the data manifold. Specifically, a uniformly sampled $u \sim \mathcal{U}[0, 1]$ is warped by $f(t) = 2t - t^2$, which preserves the endpoints $(f(0) = 0, f(1) = 1)$

while increasing the density toward large $t$. Under Euler discretisation, this concentrates integration steps where fine-grained denoising is most critical.

## E.5 HYPERPARAMETERS

For reproducibility, we report the full hyperparameter settings of FragFM, including the coarse-to-fine autoencoder (table 15), flow matching module (table 16), and discriminator module for guidance (table 17). Our primary contribution is the development of a fragment-based framework for molecular generation, rather than architectural novelty; however, since no prior models have adopted this design, we implemented the necessary modules accordingly. The hyperparameters were selected through preliminary exploratory experiments and kept fixed across all datasets and benchmarks for consistency. For the MPNN, we adopt the implementation from Gilmer et al. (2017); Battaglia et al. (2018), and for the graph transformer, we follow Vignac et al. (2023) but reduce the number of layers from 8 to 5, as the fragment-to-vector module already precedes it. For the property discriminator, we followed prior classifier-guidance studies, which typically used smaller networks than the corresponding diffusion models, and set the number of parameters accordingly.

Table 15: **Hyperparameters for the coarse-to-fine autoencoder**.

| Name | Value |
|---|---|
| **Model architecture** | |
| Number of parameters | $6,648,793$ |
| Backbone type | Sparse MPNN (Gilmer et al., 2017) |
| Encoder/decoder layers | 4 |
| Node embedding dimension | 256 |
| Edge embedding dimension | 128 |
| Hidden dimension | 256 |
| Latent dimension | 32 |
| Activation function | SiLU |
| Layer normalization | Yes |
| Initialization | Xavier |
| **Training setup** | |
| Batch size | 256 |
| Optimizer | AdamW ($\beta_1 = 0.9, \beta_2 = 0.999$) |
| Learning rate | $1.0 \times 10^{-4}$ |
| Learning rate warm-up | Linear, 2000 iterations |
| Weight decay | $1.0 \times 10^{-12}$ |
| Gradient clipping | 1.0 |
| KL divergence coefficient | $1.0 \times 10^{-4}$ |

Table 16: **Hyperparameters for the fragment-to-vector encoder and coarse graph propagation module.**

| Name | Value |
|---|---|
| **Fragment-to-vector encoder** | |
| Number of parameters | $4,257,290$ |
| Backbone type | Sparse MPNN (Gilmer et al., 2017) |
| Number of layers | 5 |
| Node embedding dimension | 256 |
| Edge embedding dimension | 128 |
| Hidden dimension | 256 |
| Activation function | SiLU |
| Dropout | 0.1 |
| Layer normalization | Yes |
| Initialization | Xavier |
| **Coarse graph propagation module** | |
| Number of parameters | $17,418,274$ |
| Backbone type | Graph Transformer (Vignac et al., 2023) |
| Number of layers | 5 |
| Attention heads | 8 |
| Node embedding dimension | 256 |
| Edge embedding dimension | 128 |
| RRWP walk length | 6 |
| Dropout | 0.1 |
| Layer normalization | Yes |
| Initialization | Xavier |
| **Training setup** | |
| Batch size | 256 |
| Fragment bag size | 384 |
| Optimizer | AdamW ($\beta_1 = 0.9, \beta_2 = 0.999$) |
| Learning rate | $5.0 \times 10^{-4}$ |
| Learning rate warm-up | Linear, 10,000 iterations |
| Weight decay | 0.0 |
| Gradient clipping | 4.0 |
| Exponential moving average (EMA) | Yes (0.999) |
| Loss coefficients | Fragment type (1.0), Fragment edge (5.0), Latent (1.0) |

Table 17: **Hyperparameters for the property discriminator module.**

| Name | Value |
|------|-------|
| **Fragment-to-vector encoder** | |
| Number of parameters | $923, 402$ |
| Backbone type | Sparse MPNN (Gilmer et al., 2017) |
| Number of layers | 4 |
| Node embedding dimension | 128 |
| Edge embedding dimension | 64 |
| Hidden dimension | 128 |
| Number of property readout layers | 3 |
| Activation function | SiLU |
| Dropout | 0.1 |
| Layer normalization | Yes |
| Initialization | Xavier |
| **Coarse graph propagation module** | |
| Number of parameters | $5, 649, 572$ |
| Backbone type | Graph Transformer (Vignac et al., 2023) |
| Number of layers | 4 |
| Attention heads | 8 |
| Node embedding dimension | 128 |
| Edge embedding dimension | 64 |
| Number of property readout layers | 3 |
| Dropout | 0.1 |
| Layer normalization | Yes |
| Initialization | Xavier |
| **Training setup** | |
| Batch size | 256 |
| Optimizer | AdamW ($\beta_1 = 0.9, \beta_2 = 0.999$) |
| Learning rate | $5.0 \times 10^{-4}$ |
| Learning rate warm-up | Linear, 10,000 iterations |
| Weight decay | 0.0 |
| Gradient clipping | 4.0 |
| Exponential moving average (EMA) | Yes (0.999) |

# F VISUALIZATION

## F.1 VISUALIZATION OF FRAGMENTS

We visualize the top-50 frequent fragments from each dataset (MOSES, GuacaMol, and NPGen).

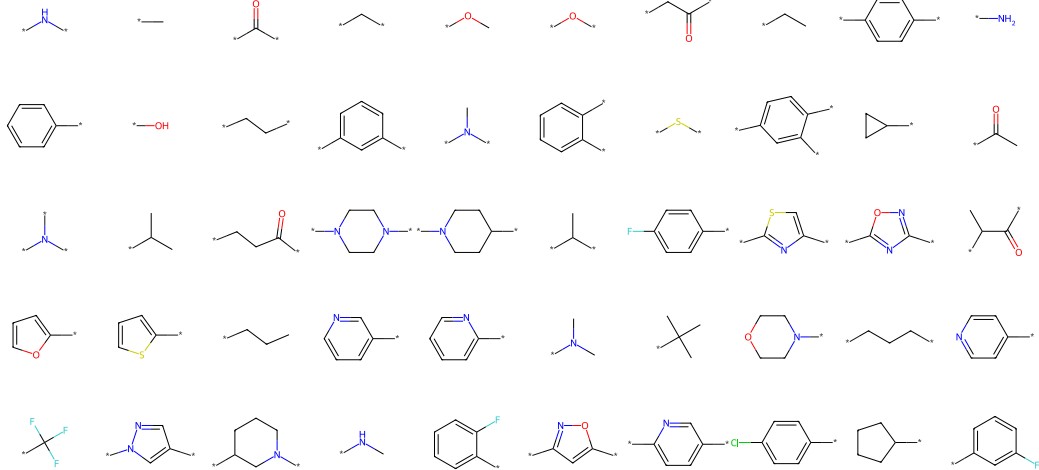

Figure 22: **Top 50 most common fragments extracted from the MOSES dataset**. More frequently occurring fragments are positioned toward the top left.

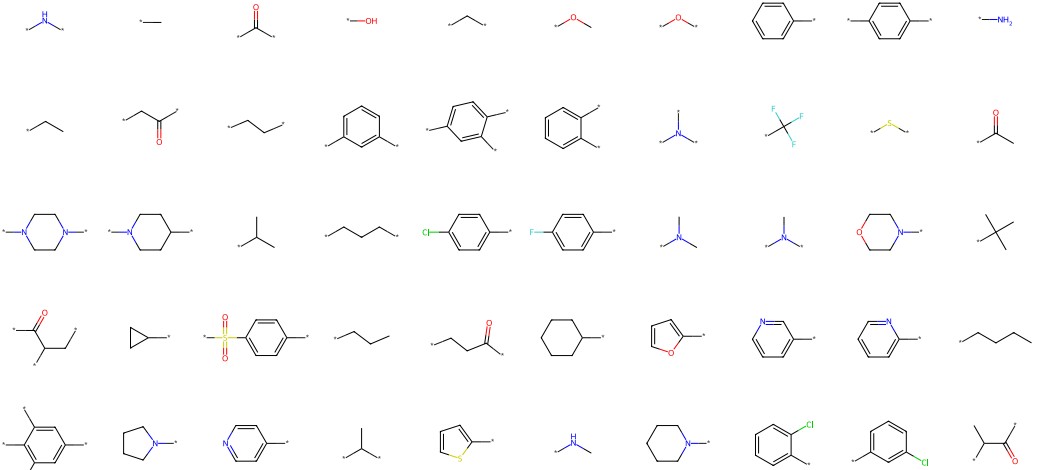

Figure 23: **Top 50 most common fragments extracted from the GuacaMol dataset**. More frequently occurring fragments are positioned toward the top left.

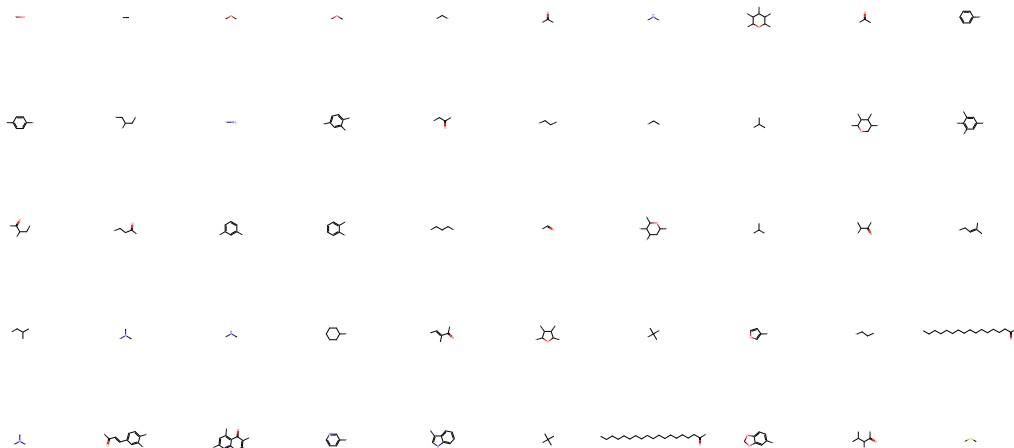

Figure 24: **Top 50 most common fragments extracted from the NPGen dataset**. More frequently occurring fragments are positioned toward the top left.

## F.2 VISUALIZATION OF FRAGMENT EMBEDDINGS

To examine the structure of the fragment embedding space learned by the fusion of the fragment-to-vector encoder module, we projected the fragment-level embeddings into two dimensions using UMAP and visualized them with respect to several physicochemical properties that can be defined at the fragment level. As shown in fig. 25, fragments with similar structures tend to occupy close regions in the latent space. We additionally visualized property-specific projections in figs. 26 and 27, including LogP, TPSA, MolMR, the number of rings, and the number of hydrogen-bond donors and acceptors. These results demonstrate that fragment-level properties align smoothly with the geometry of the learned embedding space, indicating that the model captures chemically meaningful organization at the fragment level.

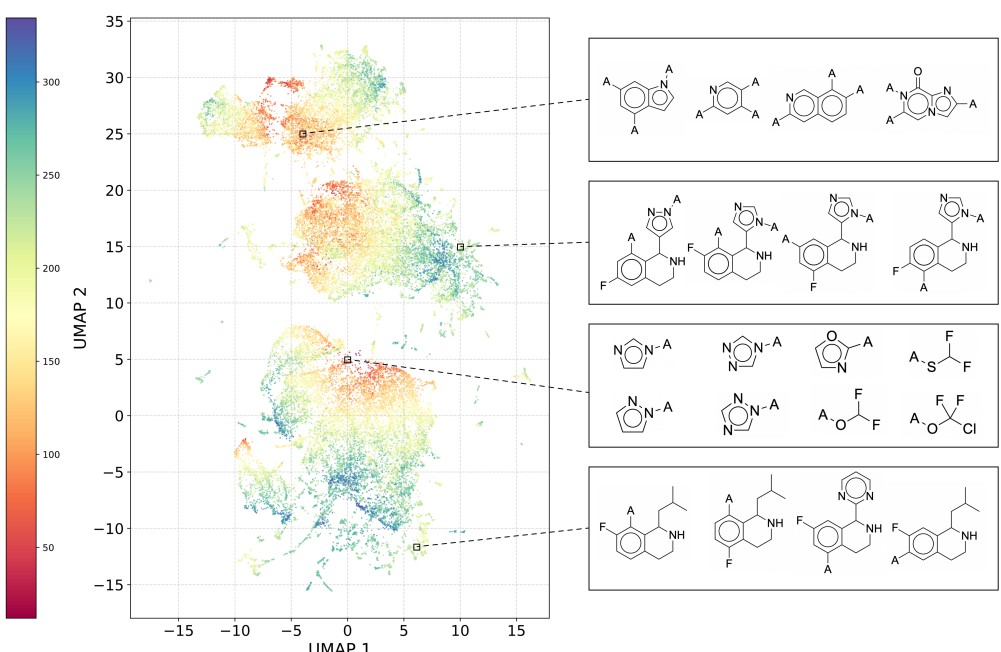

Figure 25: **UMAP visualization of MOSES fragments using fragment embeddings, color-coded by molecular weight with example molecules.** Structurally similar fragments are positioned close to each other in the embedding space.

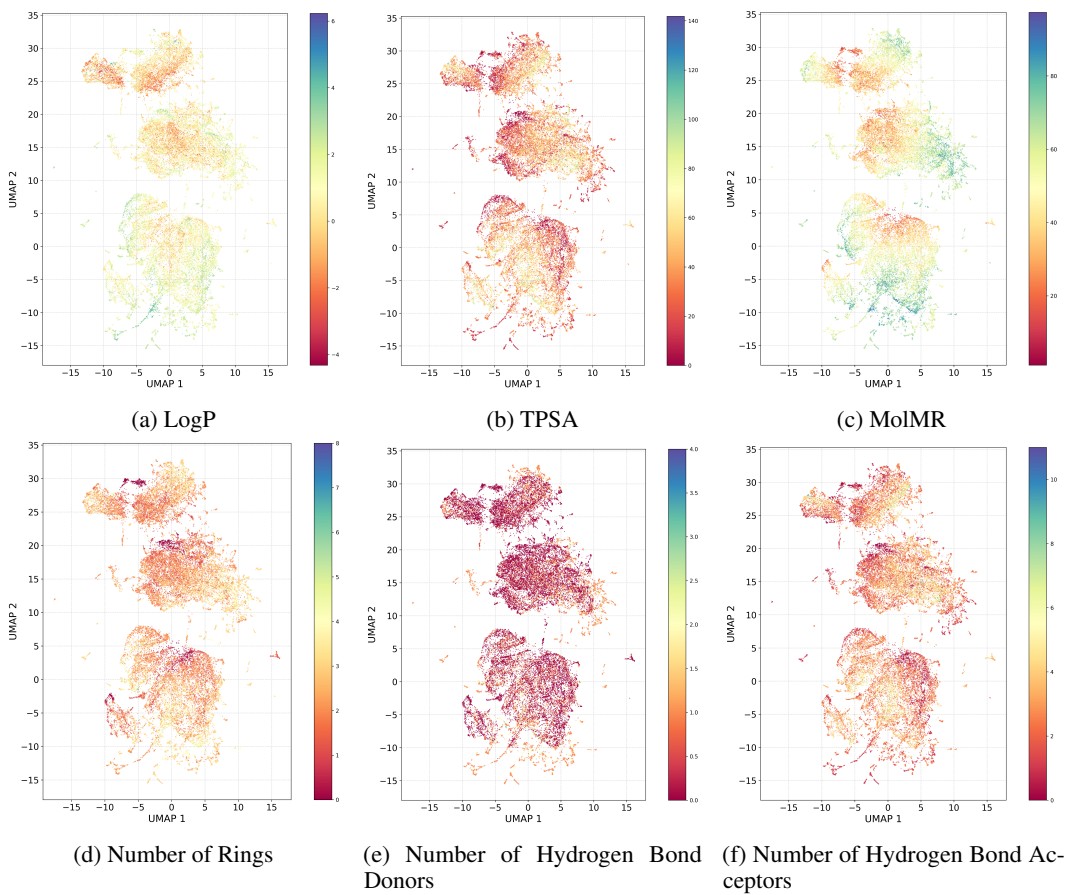

(a) LogP

(b) TPSA

(c) MolMR

(d) Number of Rings

(e) Number of Hydrogen Bond Donors

(f) Number of Hydrogen Bond Acceptors

Figure 26: **UMAP visualization of MOSES fragments using fragment embeddings, color-coded by fragment level properties.**

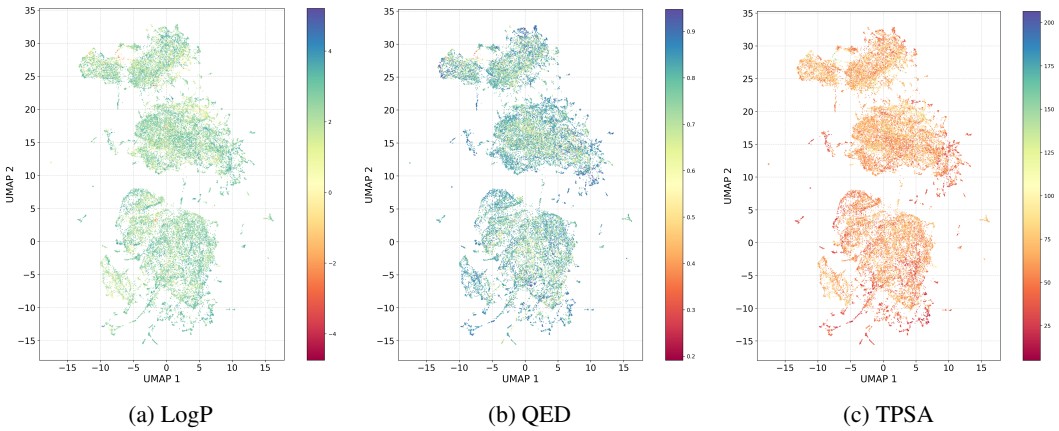

(a) LogP

(b) QED

(c) TPSA

Figure 27: **UMAP visualization of MOSES fragments using fragment embeddings, color-coded by molecule-level properties.**

## F.3 Visualization of Generated Molecules from MOSES and GuacaMol

We visualize samples generated by FragFM on the MOSES and GuacaMol datasets in figs. 28 and 29.

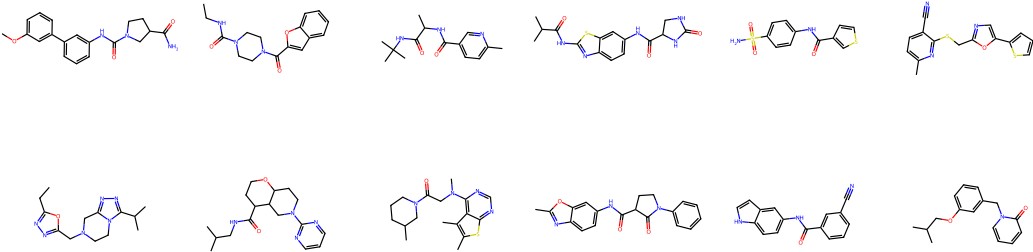

Figure 28: **Molecules generated by FragFM on the MOSES benchmark**. Molecules were randomly selected for visualization.

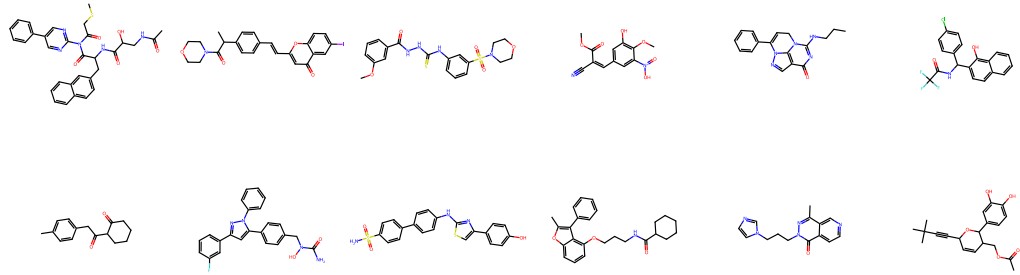

Figure 29: **Molecules generated by FragFM on the GuacaMol benchmark**. We randomly select molecules for visualization.

## F.4 Visualization and Analysis of Generated Molecules on NPGen

For the NPGen task, we show generated molecules from FragFM alongside baseline models (GraphAF, JT-VAE, HierVAE, and Digress) in figs. 30 to 34. Although all visualized molecules are formally valid in terms of valency (which rdkit can process), atom-based generative models often introduce chemically implausible motifs, such as aziridine or epoxide rings fused directly to aromatic systems, inducing severe angle strain Sweeney (2002); anti-aromatic rings with $4n$ $\pi$-electrons (violating Hückel's rule), resulting in high electronic instability Carpenter (1983); and bonds between nonadjacent atoms in a ring system, causing extreme geometric distortion Nishiyama et al. (1980). Fragment-based autoregressive models largely avoid these issues, yet they, too, exhibit limitations: JT-VAE tends to generate only small, homogeneous ring systems, while HierVAE is strongly biased toward long aliphatic chains and simple linear moieties. Consequently, these approaches show a distinct distribution of molecules from the trained dataset, matching the benchmark results in table 2.

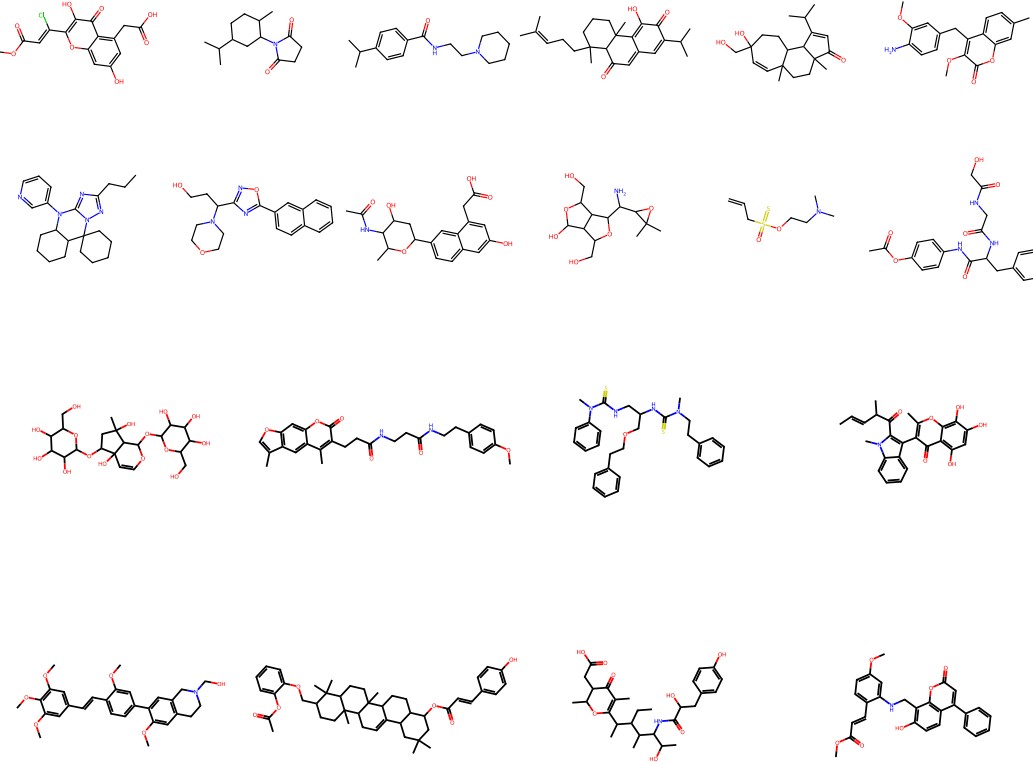

Figure 30: **Valid molecules generated by FragFM** on NPGen. The top two rows show molecules with up to 30 heavy atoms, while the bottom two rows show molecules with 31-60 heavy atoms. Molecules were randomly selected for visualization.

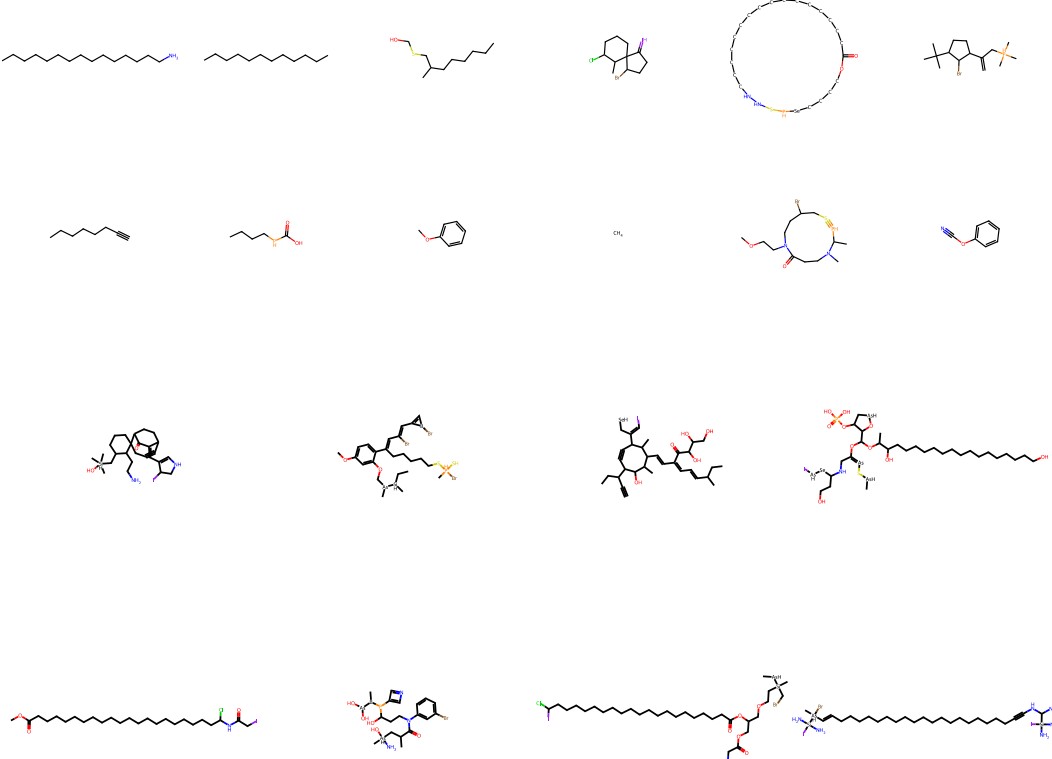

Figure 31: **Valid molecules generated by GraphAF on NPGen**. The top two rows show molecules with up to 30 heavy atoms, while the bottom two rows show molecules with 31-60 heavy atoms. Molecules were randomly selected for visualization.

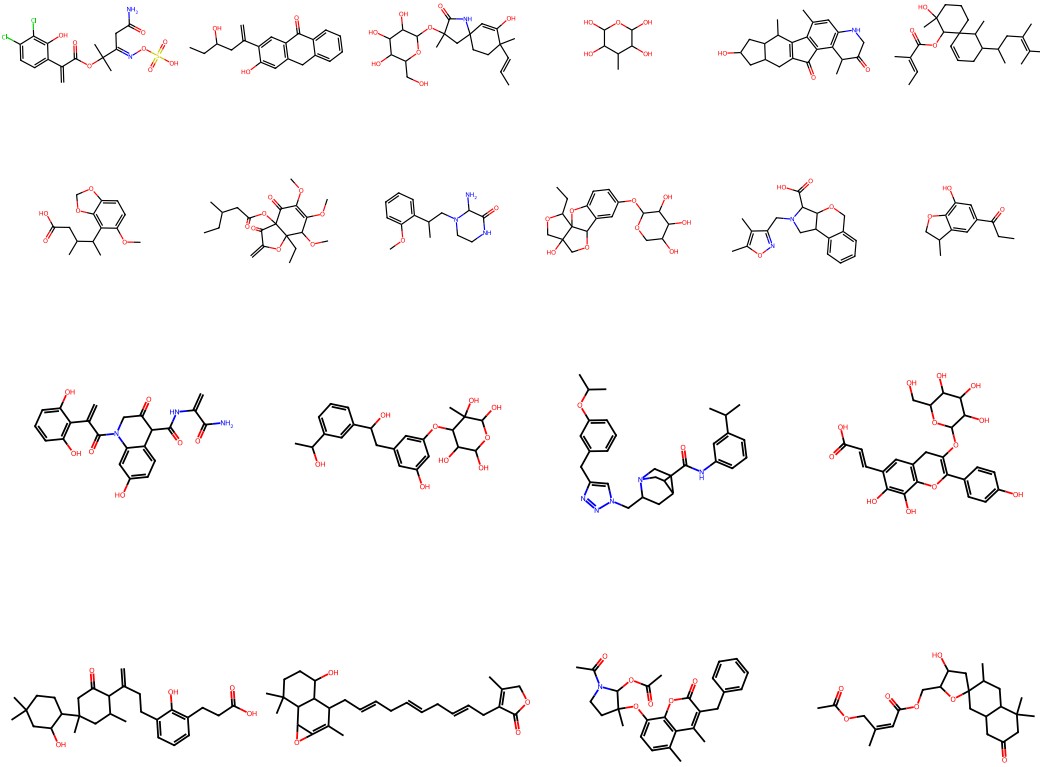

Figure 32: **Valid molecules generated by JT-VAE on NPGen**. The top two rows show molecules with up to 30 heavy atoms, while the bottom two rows show molecules with 31-60 heavy atoms. Molecules were randomly selected for visualization.

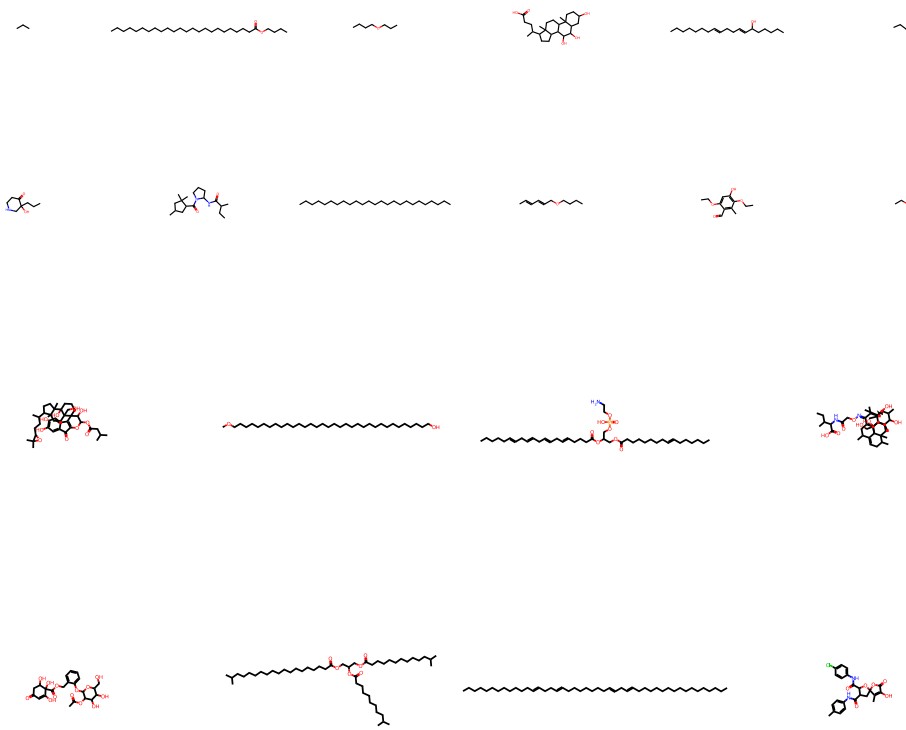

Figure 33: **Valid molecules generated by HierVAE on NPGen**. The top two rows show molecules with up to 30 heavy atoms, while the bottom two rows show molecules with 31-60 heavy atoms. Molecules were randomly selected for visualization.

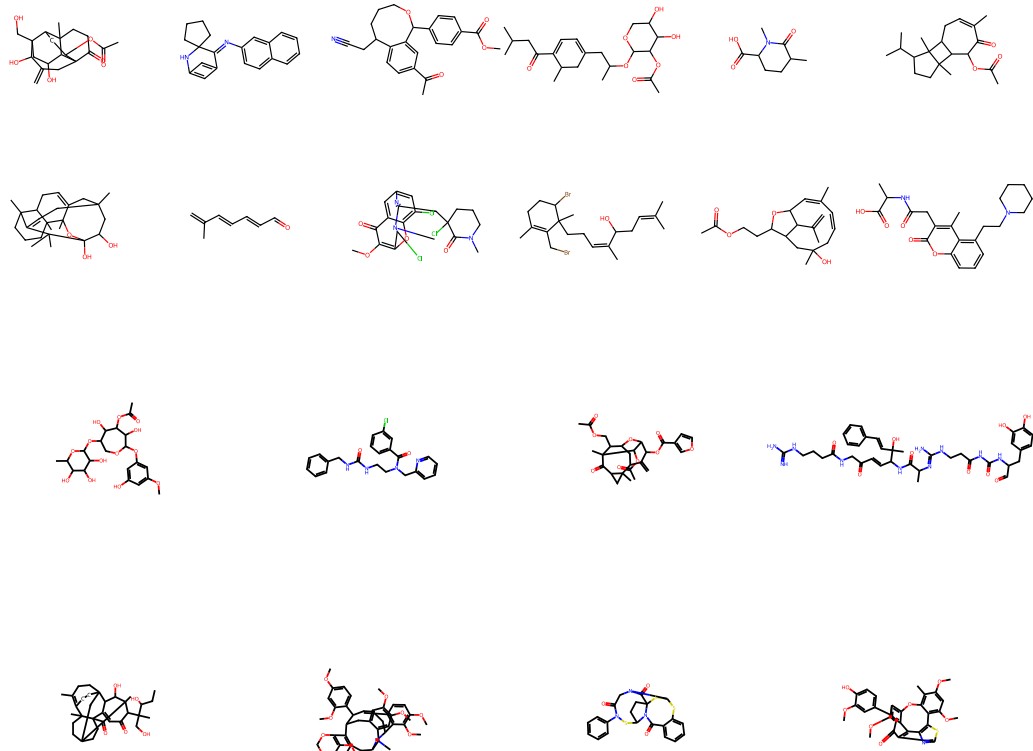

Figure 34: **Valid molecules generated by DiGress on NPGen**. The top two rows show molecules with up to 30 heavy atoms, while the bottom two rows show molecules with 31-60 heavy atoms. Molecules were randomly selected for visualization.

