# OpenReview forum: "FragFM: Hierarchical Framework for Efficient Molecule Generation via Fragment-Level Discrete Flow Matching"
_ICLR.cc/2026/Conference — ICLR 2026 Poster_

### Official Review · Reviewer_FxE9 · 2025-10-27

**Soundness:** 3
**Presentation:** 3
**Contribution:** 2
**Rating:** 4
**Confidence:** 4

**Summary:**

The paper introduces FragFM, a hierarchical framework for molecular graph generation that operates at the fragment level by combining a discrete and continuous flow matching approach with a coarse-to-fine autoencoder. The design enables efficient and scalable molecular generation by modeling chemically meaningful fragments, rather than individual atoms, while ensuring all-atom validity during molecular reconstruction. The authors also propose NPGen, a new benchmark focusing on natural products, which challenges existing approaches to generate larger, more complex, and more diverse biologically relevant compounds. Experimentally, FragFM outperforms atom- and fragment-based models on both standard molecule benchmarks and NPGen, and also demonstrates the ability to conditionally generate molecules based on desirable properties while maintaining validity.

**Strengths:**

1. The fixed-length graph embedding for fragments and fragment bag strategy allows scalable training for large fragment libraries without sacrificing chemical expressivity. By introducing the conditional fragment bag sampling term $\lambda_B$, the fragment bag restriction does not hinder conditional generation.
2. For both conditional and unconditional molecule generation, FragFM maintains high RDKit validity, demonstrating that the hierarchical approach and Blossom algorithm are effective methods to ensure chemical reasonableness.
3. FragFM outperforms atom-based and fragment-based methods in most metrics while also demonstrating a high sampling efficiency, even compared to an existing flow matching model DeFoG.

**Weaknesses:**

1. Despite the inclusion of chemical validity as a benchmark, the authors omit retrosynthetic analyses (i.e. Syntheseus) for generated molecules. This is especially important considering the reliance of the model on connecting building blocks via junction atoms, which does not necessarily guarantee a feasible experimental synthesis route.
2. Although many relevant natural products are macrocycles, the ground-truth graphs are derived directly from the results of BRICS decomposition, which results in acyclic graphs. As the Blossom algorithm employed only considers pairs of junction atoms with edges decided by this fragmentation, the model is likely unable to generate macrocycles.
3. While the hierarchical combination of fragment-level flow matching and coarse-to-fine reconstruction is well-executed, the scope of the work is somewhat incremental. Flow matching and autoencoder-based 2D molecular generation at the fragment level are individually present in the literature, as is the architecture used during training.

Minor:
1. The work omits training details for the conditional property regressor.
2. Though it may exceed the scope of the work, the model does not explicitly consider information relevant to some conditional generation tasks, such as the local geometry of the protein pocket for binding affinity.

**Questions:**

1. It is somewhat unclear how the regressor, which decides conditional fragment bag sampling weights, is trained. Does the model predict the property values of each fragment directly? If not, is it instead trained to predict the average property value of all molecules in the training dataset containing the fragment?
2. Does the NPGen dataset contain macrocycles?
3. Does conditional generation with higher $\lambda_B$ result in lower diversity? It would be interesting to see the trade-off between the strength of conditioning signals like $\lambda_B$ and $\lambda_X$ and the diversity of the results.
4. Is there any merit to changing the relative schedules of the latent vector and molecular graph during sampling? For instance, would it be possible to construct an analog design experiment by conditioning on a fully denoised latent vector of a difficult-to-synthesize molecule, then allowing the discrete graph flow matching module to run and generate analogs with a chosen fragment library?

---

> ### Author Response · Authors · 2025-11-20
>
> We are deeply grateful to the reviewer for the time dedicated to assessing our work and offering constructive and insightful feedback, including suggestions for additional evaluations of FragFM and thoughtful comments on the broader direction and scope of the paper.
>
> **[W1] On retrosynthetic analysis**
>
> FragFM is designed as a distribution-learning framework that operates on molecular graphs; as such, it does not encode retrosynthetic transformations within the generative process, similar to most contemporary graph-based generative models.
>
> That said, we fully agree that evaluating synthesizability is important for practical use. In response to the reviewer’s suggestion, we have added a detailed retrosynthetic analysis using AiZynthFinder. Across 25,000 generation trials per model, FragFM achieves the highest solved rate among all baselines and shows a distribution of solved routes that closely mirrors that of the training data. This indicates that FragFM produce systematically synthesizable assemblies, despite not leveraging reaction-level inductive biases as training objective. We believe this robustness arises from the use of BRICS fragmentation, which encodes chemically meaningful fragmentation rules and provides an implicit alignment with common retrosynthetic patterns. The complete results and discussion are now provided in **Appendix C.2** of the revised manuscript.
>
> | AIZynthFinder | Solved | 1 step | 2 steps | 3 steps | 4+ steps |
> | --- | --- | --- | --- | --- | --- |
> | MOSES (test set) | 80.1 | 53.9 | 16.0 | 5.17 | 5.0 |
> | JTVAE | 68.5 | 38.9 | 17.1 | 6.5 | 6.0 |
> | DiGress | 61.9 | 32.9 | 17.1 | 6.0 | 5.9 |
> | DeFoG | 73.1 | 44.8 | 18.4 | 5.3 | 4.6 |
> | SAFE-GPT | 72.1 | 39.0 | 20.3 | 6.8 | 6.0 |
> | FragFM | 77.0 | 46.3 | 18.7 | 6.2 | 5.8 |
>
> As an additional analysis on fragment–property relationships and partially addressing Q4, we further leverage FragFM’s fragment-bag mechanism to examine whether property-aware fragment selection can systematically propagate to molecule-level synthesizability. Inspired by the conditional guidance experiment in Figure 5, and leveraging the fact that FragFM allows fragment bags to be swapped at inference without retraining, we construct *synthesizability-aware fragment bags* and use them to steer generation without any retraining. Concretely, we select 10,000 MOSES training molecules that AiZynthFinder successfully solves, extract fragment bags from these molecules, and use these set of fragments directly during inference with the MOSES-trained FragFM. We then evaluate the synthesizability of generated molecules again using AiZynthFinder.
>
> As shown in the table below, using fragment bags derived from solvable molecules substantially improves the synthesizability profile of generated molecules: the solved rate increases 77.0 to 85.0%. Taken together with the fragment-bag conditioning experiments, these results demonstrate that property-aware fragment selection reliably propagates through FragFM’s generative process.
>
> | AIZynthFinder | Solved | 1 step | 2 steps | 3 steps | 4+ steps |
> | --- | --- | --- | --- | --- | --- |
> | MOSES (test set) | 80.1 | 53.9 | 16.0 | 5.17 | 5.0 |
> | FragFM | 77.0 | 46.3 | 18.7 | 6.2 | 5.8 |
> | FragFM (with fragments from solved molecules) | 85.0 | 52.3 | 20.7 | 6.6 | 5.4 |

---

> ### Author Response · Authors · 2025-11-20
>
> **[W2, Q2] Macrocycles in NPGen**
>
> For the macrocycle-related questions, we first report the ratio of molecules with ring sizes in the table below.
>
> | **Largest ring size** | **Training (%)** | **Validation (%)** | **Test (%)** |
> | --- | --- | --- | --- |
> | No ring | 10.55 | 10.39 | 10.54 |
> | 1 - 7 | 79.76 | 79.88 | 79.8 |
> | 8 - 11 | 3.87 | 3.84 | 3.87 |
> | 12 - 15 | 2.53 | 2.61 | 2.55 |
> | 16 - 19 | 1.87 | 1.85 | 1.86 |
> | 20 - | 1.42 | 1.43 | 1.42 |
>
> Regarding the modeling of macrocyclic structure, the reviewer is indeed correct that BRICS does not decompose bonds within ring systems, including macrocycles. As a result, each macrocyclic ring is treated as a single fragment and is directly included in the fragment vocabulary rather than being split into smaller substructures.
>
> However, this does not imply that FragFM fails to generate macrocyclic molecules. Because macrocycles are included as fragment units in the vocabulary, FragFM can technically reconstruct macrocyclic compounds whenever these fragments are sampled. Nevertheless, most macrocyclic fragments are rare in the vocabulary, which might raise concerns about under-sampling, potentially limiting their occurrence during generation. Empirically, however, we observed no such issue: as discussed in **Appendix C.5 (Appendix C.4 for old version)**, FragFM effectively recovers even infrequent fragments, including large macrocycles. We have also added representative examples of generated macrocyclic molecules in Figure 3 of the revised manuscript.
>
> In the case of generating a molecule with novel macrocycle backbone, we agree with the reviewer’s point that this cannot occur under BRICS. This phenomenon stems from the fragmentation rule rather than the FragFM modeling framework. If a cyclic-cleavage fragmentation rule were adopted (e.g. rBRICS), the flow model would be trained to generate cyclic coarse-graphs, resulting in novel macrocycle backbone.
>
> Thus, we have conducted an additional analysis about fragmentation rule and macrocycle generation. First, we trained FragFM on NPGen using rBRICS fragmentation rules and generated 10,000 molecules from the resulting model. To evaluate whether novel macrocycles emerge, we applied standard BRICS decomposition to both the generated molecules and the NPGen training set, extracting all macrocyclic fragments (ring size $\ge$ 12) from each. From this analysis, we identified several macrocyclic ring structures that do not appear in the training data. Below, we show representative pairs of newly generated macrocycles and their NPGen training-set analogs, illustrating that the generated structures are coherent with known NP-like motifs yet constitute genuinely novel macrocyclic frameworks. We also include the visualization of the macrocycles below in supplementary file **"npgen_novel_macrocycle.pdf"**.
>
> **[W3] On “incremental” advancement**
>
> While fragment-level autoencoders and flow matching have indeed been studied separately, our work goes beyond a naive integration of known components. The central contribution is the development of a fragment-level embedding space that serves as a *shared representation* capable of covering extensive chemical space and enabling high-quality molecule generation without relying on lossless reconstruction. This unified hierarchical framework allows fragment-level flow matching and atom-level reconstruction to operate jointly in a manner that, to our knowledge, is not supported by prior models.
>
> Concretely, this fragment-level embedding space yields emergent capabilities that enable FragFM to (i) operate on a higher semantic granularity than atom-level methods, improving both sampling efficiency and performance **(Figures 6, 7)**; (ii) incorporate conditioning on fragment bags and molecular properties in a simple and scalable way, while generalizing robustly to large and diverse fragment sets **(Figures 4, 5)**; and (iii) decode back to chemically faithful atom-level structures, even for large, NP-like molecules **(Table 2, Figure 30)**. These design characteristics are not present in existing fragment-level autoencoders or flow-matching approaches.

---

> ### Author Response · Authors · 2025-11-20
>
> **[Minor W1,Q1] On details regarding property predictor**
>
> We appreciate the reviewer’s attention to this point and provide additional clarification on how the fragment property predictor is trained. During revision, we have clarified the notation and loss terms in **Appendix A.3**. In summary, both the noisy molecular property predictor and the fragment-level property predictor are trained in a supervised manner. For the fragment-level predictor, fragment embeddings are first obtained from the fragment embedder (the fragment-to-vector encoder described in **Table 16** and **Figure 21**), followed by a small three-layer prediction head. Each fragment is trained to regress the property value of the molecules in which it appears (Eq. 36). We also note that the fragment-level predictor and the molecule-level predictor (Eq. 35) are optimized jointly during training.
>
> Regarding the property-conditioning experiments—LogP, QED, and ring-count conditioning on MOSES, as well as docking-score conditioning on ZINC250k—we trained the property predictors for both DiGress (using the official implementation) and FragFM using all molecules in each dataset with their corresponding property labels. We have also clarified this information in the main manuscript **Section 5.3.**
>
> **[Minor W2] On the explicit consideration of information (e.g. geometry of protein pocket)**
>
> Our work is positioned within the domain of molecular *graph* generation, where the objective is to model molecular connectivity rather than 3D conformational or pocket-specific interactions. Introducing explicit 3D pocket geometry would require a fundamentally different class of models (such as equivariant 3D diffusion or structure-conditioned frameworks) whose representations, training objectives and benchmarks are not comparable to fragment level 2D generators. Incorporating protein structure/embeddings for conditioning tasks would not constitute a fair comparison to vanilla implementation of DiGress, which relies solely on scalar molecular properties for guidance. However, we agree that incorporating 3D structural context is an interesting future direction especially for protein binding affinity task. We note that the hierarchical formulation of FragFM may offer a promising basis for such extensions.
>
> **[Q3] On molecular diversity with respect to guidance strength**
>
> We have added a quantitative analysis of how guidance strength affects diversity in **Appendix C.8** and **Figure 20**. Empirically, we observe a consistent trend that stronger conditioning leads to lower diversity: when we increase the property-guidance weight $\lambda_X$ alone, diversity decreases monotonically; the same monotonic decrease is observed when we increase the fragment-bag guidance weight $\lambda_\mathcal{B}$ alone; and in the joint setting, where both $\lambda_X$ and $\lambda_\mathcal{B}$ are varied, diversity again shows an overall decreasing trend as the combined guidance strength grows.
>
> Notably, the resulting reduction in diversity remains modest (~0.880 in the unconditional generation, ~0.855 with the strongest guidance level) indicating that the conditional generation of FragFM continues to generate structurally rich molecules even in the highly conditioned. Overall, **Figure 20** illustrates this trade-off between conditioning strength and diversity, addressing the question directly.
>
> **[Q4] On changing the relative schedules**
>
> In our design, the fragment-level graph and chosen fragment library carry most of the high-level structural and property-relevant information (scaffolds and functional groups). As a result, analog design and property control are naturally expressed by steering the fragment node-level flow and its conditioning, rather than steering $z$.
>
> From a modeling standpoint, assigning different time schedules to $\mathcal{G}_t$ and $z_t$ is in principle feasible. However, FragFM is trained under a shared distortion schedule, so changing their relative schedules only at inference time would negatively affect the performance. Extending the framework to explicitly learn or parameterize separate schedules for $(\mathcal{G}_t, z_t)$ is an interesting direction.

---

> > ### Comment · Reviewer_FxE9 · 2025-11-27
> >
> > Thank you for the clarifications.
> >
> > 1. The reviewer acknowledges that the novel shared representation yields impressive results on sampling efficiency and few-timestep generation results.
> > 2. Retrosynthetic analyses support the fact that the hierarchical reconstruction yields molecules that are highly synthetically accessible and often solvable.
> > 3. The additional experiment using rBRICS decomposition and resulting novel macrocycle generation result indicates that FragFM is flexible to macrocycles not present in the training dataset.
> >
> > The additional results provided, justification of the flexibility of FragFM, and clarification regarding the regressor are, in my opinion, sufficient to merit a score increase. However, one thing remains unclear: table 8 appears to show that re-initializing the decoder from random weights during inference yields a negligible performance decrease across evaluation metrics, despite it playing a key role in molecule reconstruction. Can you provide some insight into why this may be occurring?

---

> ### Author Response · Authors · 2025-11-27
>
> Thank you for your detailed review and for thoughtfully acknowledging the additional results in the revised manuscript.
>
> **[Regarding ablation on autoencoder]**
>
> In FragFM, the coarse-to-fine autoencoder reconstructs the atom-level graph from the coarse graph and latent vector $z$. The fragment-level graph, comprising fragment composition and their coarse-level connectivity, is determined by the discrete flow matching part, and the decoder operates on fragment junction-to-junction of given fragments with $z$ from continuous flow matching (figure 1b) (e.g. a fragment-level graph “A*”–“\*BC\*”–“*D” may be decoded either as A–BC–D or as A–CB–D).
>
> Under this design, randomly initializing the decoder affects only these reconstruction choices while leaving the fragment-level graph unchanged. The learned decoder still provides important information, as it uses data-driven patterns to determine fine-grained junction placement and atomic structure. However, because the fragment-level graph already encodes the dominant structural information (i.e., the fragment composition of each molecule), the overall performance drop remains small.

---

### Official Review · Reviewer_hbdo · 2025-10-28

**Soundness:** 3
**Presentation:** 3
**Contribution:** 2
**Rating:** 4
**Confidence:** 3

**Summary:**

This paper studies the problem of molecular graph generation. The authors propose FragFM, a novel hierarchical framework via fragment-level discrete flow matching. The authors also propose the Natural Product Generation benchmark for comprehensive evaluation. FragFM enables effective distribution learning and property-guided molecular generation.

**Strengths:**

- FragFM includes a coarse-to-fine autoencoder to deal with the ambiguity in reconstructing atomic connections.
- FragFM introduces a dynamic fragment vocabulary.

**Weaknesses:**

- In related work, hierarchical generation models such as MolGrow[1], MolHF [2], and Coarse-to-fine [3] should be included. And their comparison with fragment-based methods should be discussed.
- The coarse-to-fine autoencoder is not fully data-driven, which depends on predefined fragmentation rule and Blossom algorithm.
- Several important baselines are missing in the proposed benchmark NPGen. For example, SAFE-GPT, a fragment-based method that performs well in MOSES.
- Lacking ablation. The effectiveness of the autoencoder and the dynamic fragment vocabulary should be discussed.

[1] MolGrow: A graph normalizing flow for hierarchical molecular generation

[2] Molhf: A hierarchical normalizing flow for molecular graph generation

[3] Coarse-to-fine: a hierarchical diffusion model for molecule generation in 3d

**Questions:**

See weaknesses

---

> ### Author Response · Authors · 2025-11-20
>
> We sincerely thank the reviewer for the time and effort invested in carefully reading our manuscript and providing such a balanced and detailed evaluation. The thoughtful comments, which highlight both the strengths of our approach and the areas needing clarification, have been invaluable in helping us refine the presentation and broaden our fragment-level analyses.
>
> **[W1] Comparison with hierarchical generation models (MolGrow, MolHF, Coarse-to-fine)**
>
> We fully agree that the suggested approaches (MolGrow, MolHF, and HierDiff) share conceptual similarities with our fragment-based formulation, and we have accordingly revised the manuscript to include such baselines as related works **(Section 2.3)**. Specifically, we incorporated MolHF into our experimental comparison by training its official implementation on the MOSES and ZINC250k benchmarks. For MolGrow, whose official implementation is no longer public available, we report the MOSES results from the original paper with the best performing option (fragment-oriented). Meanwhile, Coarse-to-fine framework targets 3D molecular generation in junction with 2D, which differs from our graph-based setting (MOSES, ZINC, GuacaMol, NPGen). For this reason, we discuss Coarse-to-Fine in the related work section while focusing empirical comparisons on models that operate within the same 2D generative paradigm.
>
> |ZINC250k metrics|Valid|NSPDK|FCD|
> |---|---|---|---|
> |**MolHF**|94.75|0.0709|22.230|
> |GraphAF|67.92|0.0432|16.128|
> |GraphDF|89.72|0.1737|33.899|
> |GDSS|97.12|0.0192|14.032|
> |GSDM|92.57|0.0168|12.435|
> |GruM|98.32|0.0023|2.235|
> |SwinGNN|86.16|0.0047|4.398|
> |DiGress|94.98|0.0021|3.482|
> |GGFlow|99.63|0.0010|1.455|
> |FragFM|99.81|0.0002|0.630|
>
> |MOSES metrics|Valid|Unique|Novel|Filters|FCD|SNN|Scaf|
> |---|---|---|---|---|---|---|---|
> |**MolHF**|71.0|100.0|100.0|17.8|35.4|0.23|1.5|
> |**MolGrow (fragment-oriented)**|100.0|100.0|99.8|82.7|6.39|0.42|7.7|
> |GraphINVENT|96.4|99.8|-|95.0|1.22|0.54|12.7|
> |JT-VAE|100.0|100.0|99.9|97.8|1.00|0.53|10.0|
> |SAFE-GPT|98.1|100.0|90.9|98.2|0.71|0.54|9.8|
> |DiGress|85.7|100.0|95.0|97.1|1.19|0.52|14.8|
> |DisCo|88.3|100.0|97.7|95.6|1.44|0.50|15.1|
> |Cometh-PC|90.5|99.9|92.1|98.9|1.95|0.55|14.4|
> |DeFoG|92.8|99.9|92.1|98.9|1.95|0.55|14.4|
> |FragFM|99.8|100.0|87.1|99.1|0.58|0.56|10.9|
>
> **[W2] Dependence on the Blossom algorithm**
>
> The predefined fragmentation rule is used only to define the fragment vocabulary, analogous to specifying a tokenization scheme in language models. All downstream parameters—fragment embeddings and junction likelihoods—are entirely learned from data without imposing handcrafted connection rules.
>
> Regarding the Blossom algorithm, it is important to emphasize that this component does not introduce domain heuristics or fixed structural assumptions. Our decoder predicts, for every possible pair of fragment boundary atoms, a learned likelihood for forming a junction. Reconstructing a chemically valid atomistic graph then requires selecting a one-to-one pairing across all boundary atoms, which naturally forms a maximum-weight matching problem. We use the Blossom algorithm solely as an inference-time solver to find the matching that maximizes the total predicted log-likelihood over all candidate junctions. Thus, the algorithm is not a manually encoded rule, but rather a principled optimization procedure that yields the globally most likely reconstruction implied by the learned model.
>
> We would also like to highlight that existing fragment-based generative models typically rely on sequential or autoregressive assembly strategies, where the model commits to junction decisions one at a time based on a partially completed coarse graph. For example, JT-VAE and HierVAE attach fragments one neighborhood at a time, or autoregressive graph models such as GraphAF/GraphDF and fragment-sequence models like SAFE-GPT commit to attachment decisions based on a partially completed structure. Because each decision constrains subsequent ones, the final structure can depend heavily on the arbitrary generation order, and the training of such sequences often requires additional heuristics to linearize a graph into a decoding trajectory. These methods therefore make locally optimal (greedy) attachment choices, which may not reflect the optimal global configuration of the fragment graph.
>
> In contrast, our approach considers all junction candidates simultaneously and selects the attachment configuration that maximizes the global likelihood according to the data-driven decoder. To the best of our knowledge, FragFM is the first framework to integrate a fully learned junction-likelihood model with an exact global optimization step, enabling lossless and globally consistent atom-level reconstruction from fragment-level samples.

---

> ### Author Response · Authors · 2025-11-20
>
> **[W3] Missing baseline in NPGen (SAFE-GPT)**
>
> To further strengthen the evaluation, we extended these comparisons to NPGen in our revision, newly adding DeFoG (an atom-based discrete flow model), SAFE-GPT (a transformer-based model trained on SFAE strings, a fragment-oriented text representation of molecuels), and MolHF (a hierarchical normalizing flow) to this benchmark as well (the official implementation of MolGrow was unavailable). This update ensures that NPGen now includes baselines spanning atom-level, fragment-level, graph-based, and sequence-based generative models, thereby enhancing the completeness and practical value of NPGen as a benchmark in a way that is meaningful and useful to the research community.
>
> On NPGen, all models (DeFoG, SAFE-GPT, and MolHF) are trained and sampled using their default molecular-generation settings. Among graph-based models, FragFM shows strong performance across the evaluated metrics. SAFE-GPT, as a text-based method, attains higher distributional alignment than the graph-based baselines, while exhibiting lower novelty (~73.5%). These complementary results suggest that our method performs competitively within graph-based frameworks while maintaining a balanced trade-off between structural diversity and distributional fidelity. The complete results for all models on NPGen are provided below.
>
> | NPGen | Rep. Level | Val. | Unique. | Novel | NP Score | Pathway | Superclass | Class | FCD |
> | --- | --- | --- | --- | --- | --- | --- | --- | --- | --- |
> | GraphAF | Atom (Graph) | 79.1 | 63.6 | 95.6 | 0.8546 | 0.9713 | 3.3907 | 6.6905 | 25.11 |
> | JT-VAE | Fragment (Graph) | 100.0 | 97.2 | 99.5 | 0.5437 | 0.1055 | 1.2895 | 2.5645 | 4.07 |
> | HierVAE | Fragment (Graph) | 100.0 | 81.5 | 97.7 | 0.3021 | 0.4230 | 0.5771 | 1.4073 | 8.95 |
> | **MolHF** | Fragment (Graph) | 71.0 | 59.6 | 97.6 | 0.8831 | 1.8072 | 9.1608 | 10.3760 | 31.26 |
> | **SAFE-GPT** | Fragment (Sequence) | 96.5 | 98.6 | 73.5 | 0.0024 | 0.0054 | 0.0414 | 0.1722 | 0.15 |
> | DiGress | Atom (Graph) | 85.4 | 99.7 | 99.9 | 0.1957 | 0.0229 | 0.3770 | 1.0309 | 2.05 |
> | **DeFoG** | Atom (Graph) | 85.9 | 98.4 | 99.2 | 0.1550 | 0.1252 | 0.4134 | 1.3597 | 4.46 |
> | FragFM | Fragment (Graph) | 98.0 | 99.0 | 95.4 | 0.0374 | 0.0196 | 0.1482 | 0.3570 | 1.34 |
>
> **[W4] Ablation on autoencoder and dynamic fragment vocabulary**
>
> The autoencoder and the dynamic fragment vocabulary indeed constitute two core components of FragFM, along with the fragment embedding module.
>
> The standalone quality of the autoencoder is evaluated in **Appendix C.3 (C.2 in the initial submission)**, which shows the contribution of the latent variable $z$. To directly assess its role within the full pipeline, we also add an ablation where the flow module is paired with a randomly initialized decoder, so that the difference isolates the benefit of learned atom-level reconstruction **(Table 8, Appnedix C.3)**.
>
> Regarding the dynamic fragment vocabulary, our formulation trains a density-ratio estimator with an InfoNCE loss (Eqs. (1)–(2)), yielding a bag-conditioned posterior and transition kernel (Eq. (3)). Because $p_{\text{data}}$ is unconditioned with respect to the fragment bag $\mathcal{B}$, recovering the correct marginal requires taking the expectation of the bag-conditioned kernel over $\mathcal{B}$ at each step. In practice, we approximate this expectation by dynamically resampling $\mathcal{B}$ during sampling. Therefore, the dynamic bag strategy is not an optional design choice but a theoretical necessity for correctly modeling the unconditioned data distribution.
>
> Rather, the main factor that governs the fidelity of this approximation is the fragment bag size. Specifically, $(N_{\text{train}})$ (the number of negatives in the InfoNCE objective) primarily influences training stability, while $(N_{\text{inference}})$ directly determines the variance–bias trade-off in approximating the in-bag expectation used by the transition kernel (Eq. 3). We have analyzed this effect in **Appendix C.5**.

---

> ### Author Response · Authors · 2025-11-27
>
> Dear Reviewer hbdo,
>
> We would like to kindly follow up on our response to your review. In our revision, we incorporated additional baselines, expanded the related-work discussion (including MolGrow and Coarse-to-Fine), and added ablations on key components, which helped improve the clarity and positioning of the paper. If you have any remaining questions or further suggestions, we would greatly appreciate the opportunity to clarify them. We sincerely appreciate your time and effort in reviewing our submission.

---

### Official Review · Reviewer_RJhm · 2025-10-28

**Soundness:** 3
**Presentation:** 3
**Contribution:** 3
**Rating:** 6
**Confidence:** 4

**Summary:**

This paper introduces FragFM, a hierarchical framework for molecular graph generation that combines fragment-level discrete flow matching with a coarse-to-fine autoencoder for efficient and scalable drug-like molecule generation. The authors also propose NPGen, a new benchmark designed to evaluate the generation of complex, natural product–like molecules.

**Strengths:**

1. Interesting ideas that construct flows on latent variables and coarse graphs.
2. The “in-bag” InfoNCE formulation is computationally efficient, and provides a surrogate for different computation budget, with the bag size increasing, $x_1$ posterior becomes unconditional.
3. Strong performance on both standard benchmarks and NPGen.

**Weaknesses:**

1. The paper only evaluates 2D molecular graphs. It would strengthen the contribution to include 3D molecular generation.
2. While fragment-level generation is inherently interpretable, the paper does not analyze fragment–property correlations or latent space structure.

**Questions:**

1. How sensitive is FragFM to the choice of fragment decomposition rules (e.g., BRICS vs. other methods)?
2. Coarse graphs are generated by fixed chemical tools, can this process be learned by another graph autoencoder instead?
3. How is the fragment property predictor trained?

**Details Of Ethics Concerns:**

None.

---

> ### Author Response · Authors · 2025-11-20
>
> We thank the reviewer for the thorough evaluation and for highlighting both the strengths and the key points requiring clarification. We appreciate the constructive feedback, which helped us improve the presentation and expand our analyses regarding analyses on fragments.
>
> **[W1] On the 3D molecular generation**
>
> We appreciate the reviewer’s insightful suggestion regarding 3D molecular generation. In this work, we focused on 2D molecular graphs to clearly isolate and validate the effect of the proposed hierarchical fragment representation, while keeping the evaluation setting comparable to existing graph-based generative models. We also note that our main baselines operate on 2D graphs and are not directly aligned with 3D generation objectives or metrics, which would require a substantially different experimental setup.
>
> Importantly, FragFM was designed with 3D extension in mind. The coarse representation is complemented by the latent variable $z$, which retains atomistic junction information that may be lost during coarse abstraction. This coupling between fragment-level and atom-level representations captures precisely the information required for 3D spatial arrangement and conformational flexibility.
>
> In our ongoing implementation, we are extending FragFM to jointly generate fragment types (as if in FragFM) and their local 3D conformers within a single diffusion/flow framework. While prior fragment-based methods have sequentially assembled 3D structures by alternating between fragment selection and conformer generation, to the best of our knowledge, there is no existing approach that performs both molecular-species (2D) design and 3D structure generation in a one-shot, end-to-end manner. Due to this design choice and ongoing extension, FragFM provides a strong basis for unified fragment-level 2D–3D generative modeling, which we plan to explore further in our future work.
>
> **[W2] On the lack of fragment–property analysis**
>
> We agree with the reviewer’s point that explicitly analyzing the relationship between fragments and molecular properties would provide valuable insight. This analysis is particularly relevant for FragFM because the model operates directly at the fragment level and supports fragment-bag manipulation at inference time, making the interpretation of fragment representations an important aspect of understanding how fragment-level choices influence molecular outcomes.
>
> While our initial submission does not contain analysis on fragment’s embedding space, we note that the results on proposed fragment-bag conditioning mechanism implicitly reflects such correlations. By adjusting the fragment-bag guidance strength \lambda_{\mathcal{B}}, the model increases the occurrence probabilities of fragments that contribute to the desired property level, resulting the molecule to match the desired property (**red points in Figure 5**).  This demonstrates that fragment-bag guidance alone can steer generation toward conditioned property regions, suggesting that FragFM has learned meaningful fragment-property relationships.
>
> To make these relationships more straightforward, we conduct additional analyses during revision. In **Figure 25-27,** **Appendix F.2**  we visualize and the correlation between fragment embeddings and several fragment-level and molecular properties. Across these analyses, we observe that the learned fragment embedding space is well structured: fragments with similar physicochemical characteristics occupy coherent regions, indicating that the model organizes fragment representations in a property-aware manner.
>
> Moreover, inspired by a question raised by another reviewer (FxE9) regarding synthesizability, we examined whether fragment-property relationship also extends to this property. To this end, we revisit the core idea of fragment-bag guidance and extend it in a more direct manner by constructing synthesizability-aware fragment bags to test whether enhancing synthesizability is achievable through fragment-level control.
>
> Leveraging FragFM’s ability to replace fragment bags at inference without retraining, we constructed a fragment set enriched for synthesizable motifs derived from fragments extracted from 10,000  AiZynthFinder-solvable molecules in the MOSES training set and used this set for assembling new molecules with the MOSES-trained FragFM. The resulting molecules were re-evaluated using AiZynthFinder, and the solved rate improved from 77.0% to 85.0%.
>
> | AIZynthFinder | Solved | 1 step | 2 steps | 3 steps | 4+ steps |
> | --- | --- | --- | --- | --- | --- |
> | MOSES (test set) | 80.1 | 53.9 | 16.0 | 5.17 | 5.0 |
> | FragFM | 77.0 | 46.3 | 18.7 | 6.2 | 5.8 |
> | FragFM (with fragments from solved molecules) | 85.0 | 52.3 | 20.7 | 6.6 | 5.4 |

---

> ### Author Response · Authors · 2025-11-20
>
> **[Q1,Q2] Sensitivity to fragment decomposition rules & learnable fragmentation rule**
>
> FragFM is not tied to specific fragmentation scheme. While BRICS was used in the initial submission because it is a widely adopted and chemically interpretable rule for decomposing molecules into synthesis-inspired building blocks, the framework itself is agnostic to the source of fragments. To assess robustness, we additionally evaluated two alternative decomposition schemes; RECAP [1], a synthesis oriented fragmentation method commonly used in medicinal chemistry, and rBRICS [2], a variant of BRICS with additional ring cutting rules.
>
> As shown in the table below, FragFM exhibits consistent performance under other fragmentation rule, with no degradation in the metrics. These results confirm that the modeling strategy generalizes well across different fragmentation definitions and that the learned fragment-level flow is stable to changes in vocabulary size and composition.
>
> | MOSES metrics | valid | unique | novel | filters | FCD | SNN | scaf |
> | --- | --- | --- | --- | --- | --- | --- | --- |
> | FragFM (BRICS) | 99.8 | 100.0 | 87.1 | 99.1 | 0.58 | 0.56 | 10.9 |
> | **FragFM (RECAP)** | 99.8 | 99.9 | 83.6 | 99.3 | 0.56 | 0.57 | 13.3 |
> | **FragFM (rBRICS)** | 99.8 | 100.0 | 88.5 | 98.7 | 0.58 | 0.56 | 13.1 |
>
> | ZINC250k metrics | valid | NSPDK | FCD |
> | --- | --- | --- | --- |
> | FragFM (BRICS) | 99.81 | 0.0002 | 0.630 |
> | **FragFM (RECAP)** | 99.66 | 0.0003 | 0.580 |
> | **FragFM (rBRICS)** | 99.79 | 0.0003 | 0.563 |
>
> For adopting neural network-based learned fragmentation rules, we agree that it is actually an exciting direction. Our current design follows prior fragment-based generative models by using chemically meaningful, tool-driven decompositions (e.g., BRICS, RECAP) to ensure interpretability and compatibility with synthesis rules. Importantly, FragFM already supports flexible fragment vocabularies through its GNN-based fragment embedding and fragment-bag sampling strategy. Thus, a neural fragmentation module, such as a learnable graph autoencoder that proposes subgraphs or cut sites, can in principle be easily incorporated into FragFM. Exploring jointly learned decomposition rules is a promising avenue for future work.
>
> [1] Lewell, Xiao Qing, et al. "Recap retrosynthetic combinatorial analysis procedure: a powerful new technique for identifying privileged molecular fragments with useful applications in combinatorial chemistry." *Journal of chemical information and computer sciences* 38.3 (1998): 511-522.
>
> [2] Zhang, Leili, Vasumitra Rao, and Wendy Cornell. "r‐BRICS–A Revised BRICS Module That Breaks Ring Structures and Carbon Chains." *ChemMedChem* 19.4 (2024): e202300202.
>
> **[Q3] How is the fragment property predictor trained?**
>
> We appreciate the reviewer’s attention to this point and provide additional clarification on how the fragment property predictor is trained. During revision, we have clarified the notation and loss terms in **Appendix A.3**. In summary, both the noisy molecular property predictor and the fragment-level property predictor are trained in a supervised manner. For the fragment-level predictor, fragment embeddings are first obtained from the fragment embedder (the fragment-to-vector encoder described in **Table 15** and **Figure 21**), followed by a small three-layer prediction head. Each fragment is trained to regress the property value of the molecules in which it appears (Eq. 36). We also note that the fragment-level predictor and the molecule-level predictor (Eq. 35) are optimized jointly during training.

---

> ### Author Response · Authors · 2025-11-27
>
> Dear Reviewer RJhm,
>
> We would like to kindly follow up on our response to your review. In the revised manuscript, we added analyses of fragment–property relationships and evaluated alternative fragmentation schemes, thereby strengthening the empirical support for our approach. We have also clarified the training of the property predictors and expanded the discussion on fragmentation rules and potential 3D extensions. If you have any remaining questions or further suggestions, we would greatly appreciate the opportunity to clarify them. We sincerely appreciate your time and effort in reviewing our submission.

---

### Official Review · Reviewer_PJ7B · 2025-10-31

**Soundness:** 3
**Presentation:** 3
**Contribution:** 3
**Rating:** 6
**Confidence:** 3

**Summary:**

The paper proposes a fragment-based approach for molecular generation, with a particular focus on scalability to larger molecules (natural products; NPs).

The focus on NPs makes it difficult for me to judge the impact (of both the method and the dataset); I suspect it's somehow niche compared to the general small molecule generation, but at the same time it offers a comprehensive contribution (with both a benchmark and a new method).

**Strengths:**

- The paper is clearly written and easy to follow.
- The approach focuses on scaling to the generation of large molecules (namely natural products), which is an interesting application area.
- The approach seems novel, and has domain-specific contributions (e.g. coarse encoding)
- A new benchmark dataset is introduced, which is an additional contribution.
- Besides the comment below, the evaluation is in my opinion sound and fairly typical for 3D generation papers; it supports the claims made.

**Weaknesses:**

- The authors rely on fairly old baselines in experiments past unconditional generation. This raises questions about relative performance of the method compared to SotA. It’s also not clear why only an atom-based baseline is used in conditional generation.

**Questions:**

- The authors provide a figure for sampling times; how does the model compare in terms of training time?

---

> ### Author Response · Authors · 2025-11-20
>
> We thank the reviewer for the thoughtful and constructive assessment of our work. We appreciate the recognition of the paper’s comprehensive and domain-specific contributions, as well as the points raised, which helped us improve the completeness and clarity of the manuscript.
>
> **[Summary] On the niche comment about natural products**
>
> We would like to carefully clarify that focusing on natural products (NPs) does not make the work niche. First, our methodology is not restricted to NPs. The fragment-based hierarchical encoding and generation framework we propose can be applied broadly in molecular design beyond NPs. We selected NPs in our evaluation because they present a rigorous scalability and complexity challenge (large size, rich scaffold diversity), which aligns with the key goals of our work. In addition, our model achieves strong performance on widely used small-molecule benchmarks such as MOSES, GuacaMol, and ZINC250k, demonstrating that its effectiveness is not limited to NP-like chemical space.
>
> Second, we believe that the design and discovery of natural products (NPs) are far from niche within drug discovery; NPs represent a core and enduring challenge in modern drug discovery. NPs have historically provided a major source of bioactive compounds and approved therapeutics, owing to their rich structural complexity and pharmacologically privileged scaffolds [1]. Recent surveys highlight that nearly one-third of FDA-approved drugs since the 1980s are derived from or inspired by NPs [2]. With the growing integration of AI into medicinal chemistry, several recent works emphasize that extending deep generative models to NP-like chemical space is both timely and essential [3, 4]. These trends demonstrate that enabling AI systems to model NP-like molecules is not a niche direction but rather a critical step toward scalable and general molecular generation frameworks.
>
> Lastly, from an ML perspective, NP-like molecules serve as a valuable and challenging benchmark because their structural diversity, functional complexity, and biological relevance extend far beyond the synthetic-like molecules that dominate existing datasets. Current benchmarks such as MOSES and GuacaMol largely contain small and structurally simple molecules, and their evaluation metrics are approaching saturation for graph generative models, making them insufficient for assessing existing models. Evaluating on NPs therefore provides a meaningful stress test for a model’s ability to generalize to richer and more realistic regions of chemical space.
>
> We have additionally expanded and clarified the motivation for focusing on NPs in the revised manuscript (**Appendix B.1**), highlighting their central role in drug discovery and their value as a challenging and general benchmark for molecular generation.
>
> [1] Atanasov, Atanas G., et al. "Natural products in drug discovery: advances and opportunities." *Nature reviews Drug discovery* 20.3 (2021): 200-216.
>
> [2] Liu, Chuan-Su, et al. "Bridging chemical space and biological efficacy: advances and challenges in applying generative models in structural modification of natural products." *Natural Products and Bioprospecting* 15.1 (2025): 37.
>
> [3] Mullowney, Michael W., et al. "Artificial intelligence for natural product drug discovery." *Nature Reviews Drug Discovery* 22.11 (2023): 895-916.
>
> [4] Tay, Dillon WP, et al. "67 million natural product-like compound database generated via molecular language processing." Scientific Data 10.1 (2023): 296.

---

> ### Author Response · Authors · 2025-11-20
>
> **[W1] On the conditional generation comparison**
>
> The conditional generation experiment was designed to examine the specific contribution of fragment-level representation to property controllability, while keeping other factors constant.
>
> To ensure a fair and interpretable comparison, we controlled for both the model architecture and the sampling strategy (classifier-guided conditional sampling). For this reason, we selected DiGress as the atom-level baseline. It is one of the first diffusion / flow models to introduce classifier guidance for molecular property control and shares a highly similar backbone architecture with our model.
>
> To the best of our knowledge, no comparable fragment-level conditional generation model adopts a diffusion- or flow-based one-shot generation framework. Given this context, we believe that our chosen comparison provides the most direct and meaningful assessment of the effect of the representation itself.
>
> We sincerely welcome any suggestions of fragment-based diffusion or flow models that the reviewer considers relevant for conditional generation comparison.
>
> **[W1] On the claim about “old baselines” in unconditional generation**
>
> Each chosen baseline reflects a representative method spanning different molecular representations and model architectures. While our core comparisons emphasize graph-based molecular generative models, which are most directly aligned with our formulation, we also include broader baselines to provide an evaluation that is meaningful and useful to the research community. On the MOSES benchmark, this includes two recent state-of-the-art methods: DeFoG (ICML 2025 Oral), an atom-based discrete flow model, and SAFE-GPT (NeurIPS 2023 Workshop; *Chemical Science 2024*), a transformer-based model trained on SAFE strings, a fragment-oriented text representation of molecules. In addition, as suggested by Reviewer hbdo, we incorporated MolHF and MolGrow, a hierarchical graph normalizing flow model, as an additional baseline.
>
> To further strengthen the evaluation, we extended these comparisons to NPGen in our revision, newly adding DeFoG, SAFE-GPT, and MolHF to this benchmark as well (the official implementation of MolGrow was unavailable). This update ensures that NPGen now includes baselines spanning atom-level, fragment-level, graph-based, and sequence-based generative models, thereby enhancing the completeness and practical value of NPGen as a benchmark. The newly added results are provided below.
>
> On NPGen, training and sampling procedure of all models (DeFoG, SAFE-GPT, and MolHF) are conducted with their default molecular-generation setting. Among graph-based models, FragFM shows strong performance across the evaluated metrics. SAFE-GPT, a text-based method, attains higher distributional alignment than the graph-based baselines, while exhibiting lower novelty (73.5%). These complementary results suggest that our method performs competitively within graph-based frameworks while maintaining a balanced trade-off between structural diversity and distributional fidelity. The complete results for all models on NPGen are provided below.
>
> | NPGen | Rep. Level | Val. | Unique. | Novel | NP Score | Pathway | Superclass | Class | FCD |
> | --- | --- | --- | --- | --- | --- | --- | --- | --- | --- |
> | GraphAF | Atom (Graph) | 79.1 | 63.6 | 95.6 | 0.8546 | 0.9713 | 3.3907 | 6.6905 | 25.11 |
> | JT-VAE | Fragment (Graph) | 100.0 | 97.2 | 99.5 | 0.5437 | 0.1055 | 1.2895 | 2.5645 | 4.07 |
> | HierVAE | Fragment (Graph) | 100.0 | 81.5 | 97.7 | 0.3021 | 0.4230 | 0.5771 | 1.4073 | 8.95 |
> | **MolHF** | Fragment (Graph) | 71.0 | 59.6 | 97.6 | 0.8831 | 1.8072 | 9.1608 | 10.3760 | 31.26 |
> | **SAFE-GPT** | Fragment (Sequence) | 96.5 | 98.6 | 73.5 | 0.0024 | 0.0054 | 0.0414 | 0.1722 | 0.15 |
> | DiGress | Atom (Graph) | 85.4 | 99.7 | 99.9 | 0.1957 | 0.0229 | 0.3770 | 1.0309 | 2.05 |
> | **DeFoG** | Atom (Graph) | 85.9 | 98.4 | 99.2 | 0.1550 | 0.1252 | 0.4134 | 1.3597 | 4.46 |
> | FragFM | Fragment (Graph) | 98.0 | 99.0 | 95.4 | 0.0374 | 0.0196 | 0.1482 | 0.3570 | 1.34 |
>
> **[Q1] On training time comparison**
>
> We report the training times measured on our hardware setup. Using a single NVIDIA A100 80GB GPU, FragFM required approximately 96 hours on MOSES and 144 hours on GuacaMol (**Appendix E.2**). For comparison, DeFoG reported 46 hours on MOSES and 141 hours on GuacaMol under the same hardware configuration. On NPGen, FragFM was trained for 144 hours, whereas DiGress required 96 hours (noting that the DiGress result was obtained on a single NVIDIA H200 GPU).

---

> ### Author Response · Authors · 2025-11-27
>
> Dear Reviewer PJ7B,
>
> We would like to kindly follow up on our response to your review. In our revision, we addressed concerns regarding the natural product experiments, expanded the baseline comparisons on existing benchmarks and on NPGen, and provided the training-time comparisons, which helped improve the clarity. If you have any remaining questions or further suggestions, we would greatly appreciate the opportunity to clarify them. We sincerely appreciate your time and effort in reviewing our submission.

---

### Author Response · Authors · 2025-11-20

We would like to sincerely thank all reviewers for their thoughtful feedback and constructive suggestions, which have greatly helped us improve the quality of our manuscript. Throughout this rebuttal process, we have made several enhancements to address the concerns raised:

- **New benchmark baselines and experimental results** are highlighted in **bold** throughout our responses.
- Newly added content in the revised manuscript is marked in blue.

We hope that these revisions and clarifications adequately address the reviewers' concerns, and we remain happy to provide any further clarification or additional experiments as needed.

---

### Meta-Review · Area_Chair_fwNM · 2025-12-23

**Summary:**

The submission mainly received quite succint reviews (with the notable exception of reviewer FxE9), mostly highlighting some quite standard and unspecialized strenghts of the contribution.  Namely, reviewers agreed that the paper is well written, technically solid, and clearly motivated, addressing scalability and efficiency limitations of atom-level molecular generation. The fragment-level modeling paradigm is convincing, as it enables efficient generation of large and complex molecule while maintaining high validity and sampling efficiency (this is however well known in literature).  The coarse-to-fine reconstruction with global matching (Blossom) was viewed as a principled and effective way to ensure chemically consistent atom-level structures.  The introduction of the NPGen benchmark is also seen as a key strength of the work (though with some concerns (by PJ7B) about the niche nature of the task).

The most compelling concern raised are about missing or initially outdated baselines, particularly hierarchical and fragment-based methods. Although many of these were added in the revision, I still found that coverage of the related literature and referencing of related works is still a substantial weakness of the work which impacts my decision. Contributions to top conferences are to be expected to provide a spotless portraying of the existing literature.

Reviewer FxE9 expressed broader and deeper concerns beyond the ones shader above. In particular, they viewed the contribution as incremental, combining known components such as fragment-based autoencoders and flow matching. Questions were also posed as concerns synthesizability and the anility for macrocycles generation (answered in the rebuttal).ì

**Reviewer Concerns:**

The rebuttal successfully addressed the majority of the reviewers’ concerns.

Concerns about poor referencing of related literature, as well as missing and outdated baselines were resolved by adding a section on releted works about hierarchical models, plus additional atom-based and fragment-based models across both standard benchmarks and NPGen. The analysis of the literature and the choice of the baseline models is still not entirely satisfactory (e.g. missing both references to early works on fragment based generation, e.g. Podda et al, AISTATS 2020; missing comparison with SotA diffusion models, beyond DiGress, which has known limitations due to the classifier-based guidance).

Ablation requests were thoroughly handled, including analyses of the autoencoder, dynamic fragment vocabulary, fragment-bag size, decoder reinitialization, and alternative fragmentation rules (BRICS, RECAP, rBRICS), which demonstrated robustness of the framework.

The lack of retrosynthetic validation was convincingly resolved via AiZynthFinder analyses, showing that FragFM generates molecules with competitive or superior synthesizability relative to baselines.

Questions about macrocycle generation were addressed with additional experiments using rBRICS, demonstrating that FragFM can generate novel macrocyclic scaffolds when the fragmentation rule permits them.

A concerns that still holds is about the impact of NPGen as a benchmark: while valuable, it remains somewhat uncertain in terms of long-term adoption by the broader community.

**Reviewer Scores:**

Reviewer FxE9 was the only one providing a feedback on rebuttal, prior the incident. The comment highlights general satisfaction for how the rebuttal handled the majority of concerns, deeming this sufficient to merit a score increase (the AC can project this increase from 4 to 6).

Reviewer PJ7B and RJhm were borderline positive on the work (6) but not entirely engaged (given the limited scope and breadth of the initial review). I could expect them to contribute little to the discussion with no change of score.

Reviewer hbdo was partly negative (score 4), but their major concern about missing baselines and related works was partially addressed. I could expect them to stick to an evaluation of 4, but not to fight against acceptance.

---

### Decision · Program_Chairs · 2026-01-26

Accept (Poster)